# Wilson loops and spherical branes

**Davide Astesiano[1]⋆, Pieter Bomans[2]†, Fridrik Freyr Gautason[1,3]‡,**
**Valentina Giangreco M. Puletti[1]° and Alexia Nix[1]§**

**1** Science Institute, University of Iceland, Dunhaga 3, 107 Reykjavík, Iceland
**2** Mathematical Institute, University of Oxford, Andrew Wiles Building,
Radcliffe Observatory Quarter, Woodstock Road, Oxford, OX2 6GG, U.K.
**3** STAG Research centre & Mathematical Sciences, University of Southampton,
Highfield, Southampton SO17 1BJ, U.K

⋆ dastesiano@hi.is , † Pieter.Bomans@maths.ox.ac.uk , ‡ ffg1m24@soton.ac.uk ,
° vgmp@hi.is , § alexianix@hi.is

## Abstract

We study $\frac{1}{2}$-BPS Wilson loop operators in maximally supersymmetric Yang-Mills theory on $d$-dimensional spheres. Their vacuum expectation values can be computed at large $N$ through supersymmetric localisation. The holographic duals are given by back-reacted spherical D-branes. For $d \neq 4$, the resulting theories are non-conformal and correspondingly, the dual geometries do not possess an asymptotic AdS region. The main aim of this work is to compute the holographic Wilson loops by evaluating the partition function of a probe fundamental string and M2-brane in the dual geometry, focusing on the next-to-leading order. Along the way, we highlight a variety of issues related to the presence of a non-constant dilaton. In particular, the structure of the divergences of the one-loop partition functions takes a non-universal form in contrast to examples available in the literature. We devise a general framework to treat the divergences, successfully match the sub-leading scaling with $\lambda$ and $N$, and provide a first step towards obtaining the numerical prefactor.



# 1 Introduction

In recent years, a significant effort has been made to understand and develop gauge/string dualities beyond the conformal paradigm. In this direction, a natural example is provided by maximally supersymmetric Yang-Mills (MSYM) SU($N$) gauge theory defined on a $d$-dimensional sphere. For $2 \leq d \leq 7$ and $d \neq 4$, these (Euclidean) theories are non-conformal but are equipped with an SO($d + 1$) space-time symmetry as well as an SO($7 - d$) × SU(1, 1) R-symmetry [1]. Their holographic duals are given by the back-reaction of a stack of D($d - 1$)-branes with a spherical world-volume [2]. The branes under consideration are Euclidean, and the proper framework to treat them is in the type II* supergravity theories [3–5]. The only difference with standard type II supergravity is that the RR field strengths are purely imaginary. The global symmetries of the field theory are realised as isometries of the dual geometry as usual in holography. Due to the non-conformal nature of the dual quantum field theory (QFT), the supergravity solutions feature a running dilaton which plays an important role in

the physics of the solution.[1] Notably, MSYM on $S^d$ can be localised to a matrix model [6–8], which means that certain protected observables such as the sphere free energy can in principle be computed exactly. In practice, depending on the dimension $d$, the matrix model may only admit analytic control in the large $N$ and strong coupling limit. Nevertheless, this provides a unique framework to sharpen our understanding of string theory in a non-conformal setting characterised by a running dilaton. Strong evidence in support of the non-conformal holographic duality between MSYM theories on $S^d$ and spherical D($d-1$)-branes was provided in [6], where the on-shell action was matched at leading order in the holographic limit to the free energy of the field theory.

In this paper we focus on another observable accessible through localisation; the vacuum expectation values of the $\frac{1}{2}$-BPS Wilson loop in the fundamental representation of the gauge group. Supersymmetry constrains this loop operator to wrap the equator of the $d$-sphere and dictates a particular coupling to one of the scalars in the vector multiplet. In the large $N$ and strong coupling expansion the vacuum expectation value (vev) takes the form [6]

$$\log\langle W\rangle = 2\pi b_d + \log\left(N b_d^{(d-7)/2}\right) + \mathcal{O}\left(b_d^0 N^0\right), \tag{1}$$

where $b_d \sim \lambda^{1/(6-d)}$ denotes the endpoint of the eigenvalue distribution in the matrix model and is large in the holographic limit. The endpoint is fully specified in terms of the dimensionless 't Hooft coupling $\lambda$ (evaluating $b_6$ is more subtle, see section 2 for more details). The sub-leading corrections to this expression are not available for general $d$ as the sub-leading corrections to the eigenvalue density are not known. In holography the vev of a supersymmetric Wilson loop (in the fundamental representation) can be computed by evaluating the partition function of a single fundamental open string with appropriate boundary conditions at the asymptotic boundary [9]. In the strong coupling limit, the parameter $b_d \sim L^2/\ell_s^2$ is related to the length scale in the dual geometry so at leading order expansion, the string partition function can be evaluated using a saddle point approximation. The boundary conditions fix the leading classical configuration of the string to wrap the equator of the round $d$-sphere in the spherical D($d-1$)-brane background while trivially extending into the bulk [6]. The string partition function reduces at leading order to its regularised on-shell action which was successfully matched to the QFT prediction $2\pi b_d$ in [6].

The focus of this work is to extend this analysis to the next-to-leading order. On the gravitational side, this is provided by the one-loop partition function of the string quantised around its classical configuration. This should reproduce the log-corrections in (1) as well as the numerical constant at order $N^0 b_d^0$. In [10] it was shown that the one-loop quantisation does indeed match with the strong coupling limit of the exact QFT prediction for the Wilson loop vev for $d=5$ found in [11]. Here we will focus mostly on $\frac{1}{2}$-BPS Wilson loops of MSYM in $d=2,3,7$ but we will also briefly mention $d=6$. Before going into case specific characteristics, let us highlight some common features of the various cases. First of all, as already mentioned above, MSYM in $d \neq 4$ is non-conformal, which means that the holographic dual background is characterised by an asymptotic geometry different from the usual (asymptotically locally) AdS space. In particular, these backgrounds are characterised by a running ten-dimensional dilaton. For AdS cases, where the dilaton is constant, the contribution of the dilaton to the string partition function decouples from the quantum fluctuations of the string and can be treated separately. However, as emphasised in [10,12], this is not the case for a non-constant dilaton which is a consequence of the fact that the string world-sheet theory has non-cancelled Weyl anomaly if a running dilaton is not properly taken into account (see [13] for a review). Indeed, combining the Weyl anomaly from the string fluctuations and the dilaton is crucial to

---

[1]A general discussion of these backgrounds can be found in [2, 6], while in this work we only focus on those geometries corresponding to $d=2,3,7$.

obtain a universal result that is ultimately controlled solely by the world-sheet Euler characteristic $\chi$ and does not depend on the dilaton itself. This universal anomaly is then cancelled by the string ghosts ensuring that the string background is consistent. This demonstrates that when performing a Weyl transformation, which is frequently done in such string computations to simplify intermediate steps, one must do so simultaneously for the string fluctuations and the dilaton coupling.

The integrated Weyl anomaly of the string fluctuations also controls the UV divergences of their one-loop partition function. Even though the dilaton plays an important role in the Weyl anomaly itself, when integrated, it usually drops out leaving a universal coefficient of the UV divergence identical to the one for strings in AdS space as discussed in [14, 15]. Surprisingly, the cases studied in this paper do not seem to follow this expectation. For $d = 3, 7$ the UV divergences are different. It turns out that if the dilaton does not approach a constant value sufficiently fast in the IR, it also contributes to the UV divergences. This is exactly what happens in MSYM theory in three and seven dimensions, since the dilaton (pulled back to the classical world-sheet) diverges at the centre of the world-sheet. Importantly, the dilaton contribution to the string partition function itself diverges in the IR in a way such that at least a part of the divergences cancel against each other. Still, in order to obtain finite results we have to regularise both the UV and the IR divergences. As explained in [16], and discussed in detail in Section 3, the IR regularisation has to be carefully done in order to respect diffeomorphism invariance.

Following this logic, we propose a novel regularisation scheme that safely removes the UV divergences and generalises the ones used previously [15,17]. Furthermore, it resembles those used when computing entanglement entropy in quantum field theory. In order to describe the regularisation procedure, let us assume that the classical world-sheet metric is written in conformally flat coordinates with a conformal factor $e^{2\rho}$. The conformal factor vanishes at the centre of the world-sheet where the partition function receives a significant contribution in our regularisation procedure. Let $\sigma \to \infty$ be a local radial coordinate at the centre, then the regularised one-loop partition function of the string is

$$\log Z_{\text{1-loop}} = -(\Gamma_{\mathbb{K}})_{\text{reg}} + \lim_{\sigma \to \infty} \left( \rho - \Phi_0 - \sigma \partial_\sigma (\rho - \Phi_0) + \log \frac{\pi}{\ell_s} \right). \tag{2}$$

Here $Z_{\text{1-loop}}$ denotes the full one-loop partition function of the string and $(\Gamma_{\mathbb{K}})_{\text{reg}}$ is the quantum effective action of the quadratic fluctuations computed using the Weyl transformed metric to flat space. Here we have assumed that the quadratic fluctuations do not feature any zero-modes. If they do, these must be treated separately and combined with the above expression. The last term combines regularisation terms that involve the conformal factor $\rho$ and the coupling to the dilaton whose pull-back to the world-sheet is denoted by $\Phi_0$. Finally, we include the factor $\log \frac{\pi}{\ell_s}$ to ensure a proper match with the field theory for well known AdS cases such as the Wilson loop vev in $\mathcal{N} = 4$ SYM. This factor is regularisation scheme dependent and in this paper we use phase shifts to compute the fluctuations determinant with a cut-off on the mode momenta (see section 3). When the spectrum of the fluctuation operators can be computed explicitly it is more common to use $\zeta$-function (or heat kernel) regularisation, for which the regularisation factor is $-\log(4\pi\ell_s)$. We emphasise that the above procedure exactly matches the one in [15] when all terms are correctly evaluated for an AdS$_2$ world-sheet geometry. The benefit of the above formula is that it can be generalised to the study of string partition functions that are not necessarily dual to Wilson loops in a holographic context and not necessarily of disc topology. For general genus world-sheets we expect that the regularised partition function (for the non-zero modes) takes the form

$$\log Z_{\text{1-loop}} = -(\Gamma_{\mathbb{K}})_{\text{reg}} + \sum_{p \in P} \lim_{\sigma_p \to \infty} \left( \rho - \Phi_0 - \sigma_p \partial_{\sigma_p} (\rho - \Phi_0) + \log \frac{\pi}{\ell_s} \right), \tag{3}$$

where $P$ denotes the set of all isolated points where the conformal factor vanishes and $\sigma_p \to \infty$ denotes the local radial coordinate at each point. Note that for each such point the regularisation factor $\log \frac{\pi}{\ell_s}$ must be included as dimensional analysis would suggest. We have verified that for a closed genus $g = 0$ world-sheet the above regularisation procedure reproduces the factor $C(2)$ determined in [17].[2]

Returning to the Wilson loop in MSYM, this regularisation procedure reproduces the expected behaviour of the vev (1) for general $d$. It therefore confirms that the strong-coupling analysis of the matrix model that leads to (1) gives the correct logarithmic scaling of the Wilson loop vev. A priori, we would have expected that a sub-leading correction to the eigenvalue density of [6] is required for determining the logarithmic scaling of the vev, but surprisingly the leading order density is enough. However, in order to obtain the numerical factor of order $b_d^0$, the leading order density is in general not sufficient.

In this paper we are also interested in determining the numerical factor. To this end we take two approaches, first we compute the regularised quantum effective action $(\Gamma_{\mathbb{K}})_{\text{reg}}$ for $d = 3, 7$ and combine all factors to determine the numerical prefactor. In three dimensions the strong coupling density of eigenvalues was argued to be exact in $\lambda$ and hence the Wilson loop vev can be computed exactly in the large $N$ limit. Unfortunately, the string theory computation does not match the numerical factor predicted by the matrix model. In seven dimensions a numerical analysis of the matrix model allows us to provide concrete predictions for the numerical coefficient in the Wilson loop vev which we can then compare to the string theory computation. In addition, we provide an expression for the perturbative planar free energy, which is obtained using numerical and analytical techniques. We also compute the numerical coefficient in the Wilson loop vev using string theory, but once again it does not yield a precise match with the matrix model prediction. For $d = 2, 6$ both the matrix model and string theory computations require some extra care which we leave for a future study, although we present some preliminary results for $d = 2$.

It is tempting to speculate that the mismatch we find between the string and the matrix model is ultimately due to the divergent dilaton at the centre of the world-sheet. For this reason, in seven dimensions, we uplift the string computation to eleven dimensions and compute the one-loop partition function of an M2-brane in an orbifolded $\text{AdS}_4$ geometry, where the orbifold determines the number $N$ of D6-branes. Although we are able to compute the partition function for general $N$ the M2-brane partition function does not yield the correct scaling with $N$ and we do not find an agreement with the QFT. In the conclusion we speculate on the reason behind these mismatches.

The remainder of this paper is organised as follows. We kick off in Section 2 with a brief review of maximal super Yang-Mills theory on $S^d$ highlighting the aspects that will be important in the remainder of this work. In Section 3 we move to the gravitational side and introduce the general framework to compute holographic Wilson loops in these theories, focusing on the next-to-leading order. In this section we carefully discuss the relation between the Weyl anomaly and UV divergences as well as introduce a new regularisation scheme adapted to deal with the challenges discussed above. In addition, this section reviews the main tool used in the computation of the one-loop string and M2-brane partition functions, the phase shift method. Next, in Section 4, we extend the general discussion to the one-loop action for the M2-brane which holographically represents the Wilson loop after uplifting to M-theory. Following this general discussion we continue with a case by case discussion for $d = 2, 3$, and 7. In Section 5 we focus on the seven-dimensional theory where we present the computation using various methods both from the string and M2-brane perspective. In Section 6 we compute the string partition function dual to the Wilson loop in $d = 3$. Finally, in Section 7, we perform a partial analysis of the holographic Wilson loop for $d = 2$. We conclude with a discussion of our

---

[2]In [17] a $\zeta$-function regularisation scheme was used and so the finite regularisation factor is $-\log(4\pi\ell_s)$.

results and several future directions in Section 8. The various appendices contain technical results which were omitted from the main text. Appendix A contains our conventions used in the computations of functional determinants. Appendix B compares the heat kernel and the phase shift method for computing quantum effective actions. Next, Appendix C discusses an alternative derivation of the results of Section 5 using the Gel'fand-Yaglom method. Appendix D and E summarise the results from a WKB analysis and highlight the behaviour of the dilaton in various cases respectively.

## 2 Maximal super Yang-Mills on $S^d$

Formulating maximally supersymmetric Yang-Mills theories on a sphere is a non-trivial task. For generic dimensions $d$ with $d \neq 4$, the theory is not conformal and naively introducing minimal coupling to the curvature of the sphere breaks all supersymmetry. However, as shown in [1,7], for $d \leq 7$, one can consistently couple the MSYM theory to the sphere preserving all 16 supercharges. The curvature introduces new couplings in the MSYM action which breaks the $\mathfrak{so}(1, 9-d)$ R-symmetry of the theory in flat space to $\mathfrak{su}(1,1) \times \mathfrak{so}(7-d)$.[3] The corresponding Lagrangian in general dimensions can be obtained by dimensionally reducing the ten-dimensional MSYM theory and is then given by [1,7]

$$
\begin{aligned}
\mathcal{L} = -\frac{1}{2g_{\text{YM}}^2} \text{Tr} \Bigg( &\frac{1}{2} F_{MN} F^{MN} - \bar{\Psi} \slashed{D} \Psi + \frac{d-4}{2\mathcal{R}} \bar{\Psi} \Gamma^{089} \Psi + \frac{2(d-3)}{\mathcal{R}^2} \phi^A \phi_A \\
&+ \frac{d-2}{\mathcal{R}^2} \phi^i \phi_i + \frac{2i(d-4)}{3\mathcal{R}} [\phi^A, \phi^B] \phi^C \varepsilon_{ABC} - K_m K^m \Bigg).
\end{aligned}
\tag{4}
$$

The indices $M, N = 0, 1, \ldots, 9$ are the original ten-dimensional Lorentz indices, which after dimensional reduction split into the scalar indices $I, J = 0, d+1, \ldots, 9$ and the coordinate indices on $S^d$, $\mu, \nu = 1, 2, \ldots, d$. The additional terms containing dependence on the radius $\mathcal{R}$ of $S^d$, further split the scalar indices into the two sets, $i, j = d+1, \ldots, 7$ and $A, B = 0, 8, 9$. The spinors $\Psi$ are ten-dimensional Majorana-Weyl spinors reduced to 16 independent components by demanding that they satisfy the chirality condition $\Gamma^{11} \Psi = \Psi$. Finally, $K^m$ collectively denotes seven auxiliary fields allowing for an off-shell formulation of supersymmetry. In the remainder of this section we work in fully Euclidean signature, and hence, we analytically continue the scalar field $\phi^0 \to i\phi^0$, the auxiliary field $K^m \to iK^m$ and the Lagrangian $\mathcal{L} \to -i\mathcal{L}$.

As shown in [7,8], the theory defined by the Lagrangian (4) can be localised to a Hermitian matrix model. Introducing the dimensionless $N \times N$ Hermitian matrix $\mu = \mathcal{R}\phi^0$, for general dimension $d$, the partition function reduces to [7, 18, 19]

$$
Z = \int_{\text{Cartan}} [\mathrm{d}\mu] \exp\left( -\frac{4\pi^{\frac{d+1}{2}} N}{\lambda \, \Gamma(\frac{d-3}{2})} \text{Tr} \, \mu^2 \right) Z_{1-\text{loop}} Z_{\text{instantons}} \,,
\tag{5}
$$

where the integral is taken over the adjoint matrices in the Cartan of the gauge group, and we introduced the dimensionless 't Hooft coupling

$$
\lambda = \frac{g_{\text{YM}}^2 N}{\mathcal{R}^{d-4}} \,.
\tag{6}
$$

The term $Z_{1-\text{loop}}$ gives the contribution of the fluctuations around the localised fixed point and also includes the Vandermonde determinant [7,8]. For $d < 6$, the term $Z_{1-\text{loop}}$ is convergent, while for $d \geq 6$ it diverges and has to be regularised appropriately. We refer the reader to [6]

---

[3]Note that in the case of $d = 7$ the R-symmetry remains unbroken.

Table 1: In this table we collect the explicit expressions for the kernel $G_d$ introduced in equation (8) for various dimensions $d$. For $d = 6$ we only state the leading order term in the large $\mu$ expansion as the full kernel is quite lengthy.

| Dimension $d$ | $G_d(\mu)$ |
|:---:|:---:|
| 2 | $\frac{4}{\mu + \mu^3}$ |
| $3 + \epsilon$ | $\frac{2}{\mu} - \frac{1}{\mu + i\epsilon} - \frac{1}{\mu - i\epsilon}$ |
| 4 | $\frac{2}{\mu}$ |
| 5 | $2\pi \coth(\pi\mu)$ |
| $6 - \epsilon$ | $-6\mu \log\mu + \mathcal{O}\left(\frac{1}{\mu}\right)$ |
| 7 | $2\pi\left(1 - \mu^2\right)\coth(\pi\mu)$ |

for more details. In this work we are interested in the large $N$ regime, where Yang-Mills instanton corrections are exponentially suppressed and can be ignored. Therefore, the partition function is dominated by solutions of the following saddle point equation [6,7]

$$(d-1)(d-3)V_d N\mu_i = \lambda \sum_{j \neq i} G_d(\mu_{ij}), \tag{7}$$

where $\mu_{ij} \equiv \mu_i - \mu_j$, $V_d = 2\pi^{\frac{d+1}{2}}/\Gamma\left(\frac{d+1}{2}\right)$ is the volume of a $d$-dimensional unit sphere, and the kernel $G_d(\sigma)$, for generic dimension $d$, is given by [18]

$$\frac{i\,G_d(\mu)}{\Gamma(4-d)} = \frac{\Gamma(-i\mu)}{\Gamma(4-d-i\mu)} - \frac{\Gamma(i\mu)}{\Gamma(4-d+i\mu)} - \frac{\Gamma(d-3-i\mu)}{\Gamma(1-i\mu)} + \frac{\Gamma(d-3+i\mu)}{\Gamma(1+i\mu)}. \tag{8}$$

In Table 1 we explicitly write the expressions for the kernel $G_d$ for the various dimensions. For $d = 3$, naively we find that the left hand side of the saddle point equation vanishes, while the kernel has a pole. For this reason we set $d = 3 + \epsilon$ and expand the left and right hand side of equation (7) around $\epsilon = 0$. Higher order terms in $\epsilon$ do not contribute to $G_3(\mu)$, and its final expression, written in Table 1, is in this sense exact [6]. Similarly, in $d = 6$ the kernel diverges. As explained in more detail in [6] one should renormalise the (bare) 't Hooft coupling to obtain a finite expression for the kernel. The full expression for the kernel is rather unwieldy, but at leading order in the strong coupling, or equivalently in the large $\mu$ expansion, it reduces to the expression written in the table below.[4] To solve the saddle point equation (7) it is useful to introduce the eigenvalue density, which in the large $N$ limit becomes a smooth non-negative function of the eigenvalues,[5]

$$P(\mu) = \frac{1}{N}\sum_{i=1}^{N} \delta(\mu - \mu_i), \qquad \int_{-\mu_\star}^{\mu_\star} d\mu\, P(\mu) = 1. \tag{10}$$

In terms of the eigenvalue density $P(\mu)$, the saddle point equations can be rewritten as

$$(d-1)(d-3)V_d\,\mu = \lambda \fint_{-\mu_\star}^{\mu_\star} d\mu'\, P(\mu')G_d(\mu - \mu'), \tag{11}$$

---

[4]For completeness, the full kernel for the $d = 6$ matrix model is given by

$$G_6(\mu) = \frac{8 + 46\mu^2 + 20\mu^4}{4\mu + 5\mu^3 + \mu^5} - 3\mu\left(\psi(i\mu - 2) + \psi(-i\mu - 2)\right), \tag{9}$$

where $\psi(x) = \Gamma'(x)/\Gamma(x)$ is the polygamma function.

[5]We always assume the support of $P(\mu)$ to be compact.

where the eigenvalue distribution is supported on the interval $[-\mu_\star, \mu_\star]$ and the barred integral denotes the principal value. At weak coupling, i.e. small separation of the eigenvalues, the kernel takes the form $G_d \approx \frac{2}{\mu_{ij}}$, independently of the dimension, so in this regime the eigenvalue density reduces to the Wigner semicircle distribution. In this work we are interested in the strong coupling limit, which we define as the regime where the eigenvalues are widely separated, that is $|\mu_{ij}| \gg 1$.[6]

At leading order in the strong coupling expansion one can compute the eigenvalue density to find [6]

$$P^{\mathrm{LO}}(\mu) = \frac{2\pi^{\frac{d+1}{2}}}{\pi\lambda\Gamma(6-d)\Gamma(\frac{d-1}{2})}\left(b_d^2 - \mu^2\right)^{\frac{5-d}{2}}, \tag{12}$$

where the endpoints of the leading order distribution are located at $\pm\mu_\star = \pm b_d$, where

$$b_d = (4\pi)^{\frac{d+1}{2(d-6)}}\left(32\lambda\Gamma\left(\frac{8-d}{2}\right)\Gamma\left(\frac{6-d}{2}\right)\Gamma\left(\frac{d-1}{2}\right)\right)^{\frac{1}{6-d}}. \tag{13}$$

These results are strictly only valid in the (open) interval $d \in (3,6)$. However, the results can be analytically continued to $d \in [2,7]$, where they can be matched with the corresponding leading order supergravity result.[7]

The main focus of this work are $\frac{1}{2}$-BPS Wilson loop operators. As discussed in [6], their vacuum expectation value can be computed using supersymmetric localisation. The Wilson loop in question wraps the equator of the sphere and its expectation value can be written as [6,7]

$$\langle W\rangle = \left\langle\mathrm{Tr}\,\mathcal{P}\,e^{i\oint \mathrm{d}x^\mu A_\mu + i\oint \mathrm{d}s\,\phi_0}\right\rangle, \tag{15}$$

where $A_\mu$ is the $d$-dimensional gauge field. By choosing the Wilson loop operator to be supersymmetric with respect to the localising supercharge used to obtain (5), the vev $\langle W\rangle$ can be localised and computed using the matrix model introduced above. In particular, on the localisation locus the gauge field vanishes, and expression (15) reduces to[8]

$$\langle W\rangle^{\mathrm{LO}} = \left\langle\mathrm{Tr}\,\mathcal{P}\,e^{i\oint \mathrm{d}s\,\phi_0}\right\rangle = N\int_{-b_d}^{b_d} \mathrm{d}\mu\,P^{\mathrm{LO}}(\mu)\,e^{2\pi\mu} = N\,{}_0F_1\left(4 - \frac{d}{2}, b_d^2\pi^2\right). \tag{16}$$

Here ${}_0F_1(a,z)$ is the confluent hypergeometric function, which can be expressed in terms of modified Bessel functions. For general dimensions this expression only provides the leading order result. However, when $d = 3$ or $d = 4$ this leading order answer in fact gives the exact answer in the planar limit!

Similarly, the free energy of the MSYM on a $d$-sphere can be computed from the eigenvalue density as

$$\frac{F}{N^2} = \frac{4\pi^{\frac{d+1}{2}}}{\lambda\Gamma\left(\frac{d-3}{2}\right)}\int_{-\mu_\star}^{\mu_\star} \mathrm{d}\mu\,P(\mu)\mu^2 - \int_{-\mu_\star}^{\mu_\star} \mathrm{d}\mu\,P(\mu)\int_{-\mu_\star}^{\mu_\star} \mathrm{d}\mu'\,P(\mu')V_{\mathrm{eff}}\left(\mu - \mu'\right), \tag{17}$$

---

[6]Note that due to this somewhat unconventional definition of the strong coupling regime this does not always correspond to large $\lambda$. Indeed, for $d = 7$ and $d = 6$ (as well as some examples in five dimensions with half-maximal supersymmetry [20]) the strong coupling regime is located at $(-\lambda)^{-1} \gg 1$. However, for almost all other examples in the literature this regime coincides with the 'usual' large $\lambda$ regime.

[7]In the case $d = 6$ the expressions (12), (13) are not well-defined and the analytic continuation is more subtle. However, carefully analysing this case one finds (see Appendix C of [6]),

$$P^{\mathrm{LO}}(\mu) = \frac{1}{\pi\sqrt{b_6^2 - \mu^2}}, \qquad b_6 = 2\exp\left(-\frac{8\pi^3}{3\lambda} + c\right), \tag{14}$$

where the $\mathcal{O}(1)$ constant $c$ is scheme dependent. For all considerations in this paper we can safely set $c = 0$.

[8]In [6] the overall factor $N$ on the right hand side of equation (16) was absent due to a different normalisation convention for the Wilson loop operator.

where $V_{\text{eff}}$ is the logarithm of $Z_{1-\text{loop}}$ in (5).

Using these expressions we can compute the $\frac{1}{2}$-BPS Wilson loop vev and free energy at leading order as

$$\log \langle W \rangle^{\text{LO}} = 2\pi b_d \,, \qquad \frac{F^{\text{LO}}}{N^2} = -\frac{4(6-d)\pi^{\frac{d+1}{2}}}{\lambda(8-d)(d-4)\Gamma\left(\frac{d-3}{2}\right)} b_d^2 \,. \qquad (18)$$

The main goal of the remainder of this work is to extend this result beyond leading order and establish a next-to-leading order holographic match. In this work we focus on D1-, D2-, D5- and D6-branes, while [10] presented similar results in the context of spherical D4-branes. The D1- and D5-brane case are particularly hard to access using purely field theoretic methods, but as we shall see, the leading order kernel in these cases already proves sufficient to exhibit the correct next-to-leading order scaling while the exact numerical prefactor will receive crucial contributions from the sub-leading terms in the kernel and is currently out of our reach. On the other hand, for D2- and D6-branes the matrix models are much better behaved. The D2-brane matrix model admits an analytic large $N$ eigenvalue density while for the D6 model we can obtain precise numerical predictions.

## 3 Holographic Wilson loops

As illustrated in the introduction, the vev of a supersymmetric Wilson loop can be computed holographically by evaluating the partition function of a fundamental string with appropriate boundary conditions

$$\langle W \rangle = Z_{\text{string}} \,. \qquad (19)$$

For the vev of $\frac{1}{2}$-BPS Wilson loops we are interested in, the fundamental open string wraps the equator of the $d$-sphere in the UV region of the geometry. The string must also approach a particular location in the internal space in the UV which is dictated by the scalar coupling of the dual Wilson loop and can be read off of (15). We work in a saddle point expansion of the string partition function, that is

$$Z_{\text{string}} \approx \sum_{\text{saddles}} e^{-S_{\text{cl}}} Z_{1-\text{loop}} \,, \qquad (20)$$

where each saddle must satisfy the above boundary conditions. At strong coupling, the leading contribution to this expression is given by the regularised classical on-shell action $S_{\text{cl}}$ of a single leading saddle. The sub-leading contribution $Z_{1-\text{loop}}$ to each saddle consists of the following parts:

$$Z_{1-\text{loop}} = e^{-S_{\text{FT}}}(\text{Sdet}'\,\mathbb{K})^{-1/2}Z_{\text{zero-modes}} \,. \qquad (21)$$

The first term on the right hand side involves the Fradkin-Tseytlin action $S_{\text{FT}}$ [21, 22], which describes the coupling of the dilaton to the world-sheet, see Section 3.3.2. The second term $(\text{Sdet}'\,\mathbb{K})$ represents the functional determinant of the quadratic bosonic and fermionic operators, collectively denoted by $\mathbb{K}$, acting on the fluctuations around the classical configuration, see Section 3.3.1. The prime indicates that zero modes have been excluded in evaluating the functional determinants, and their contribution is taken into account in $Z_{\text{zero-modes}}$. In all the cases discussed in this work, zero modes are absent, hence $Z_{\text{zero-modes}} = 1$. The various quantities in this expression diverge and need to be regularised. These divergences have been discussed extensively in the past and we review them in Section 3.4. Notably, when the dilaton is constant the divergences are universal [14], depending only on the Euler characteristic $\chi$. Regularisation can then be carried out in a straightforward way, by including a regularisation

factor that similarly depends on the Euler characteristic [15]. The gravitational backgrounds we study in this paper exhibit a running dilaton. We show that for regular dilaton profiles, the UV divergences remain universal and can be treated as before. On the other hand, for a diverging dilaton we have to devise new ways to take care of the UV divergences we encounter. We propose a new regularised version of (21) in Section 3.6 below. Prior to that, we revisit the phase shift method for computing the finite part of the one-loop fluctuation determinants in Section 3.5, which also fixes our regularisation scheme.

## 3.1 The string action

The world-sheet action of the string is composed of three parts

$$S = S_{\text{bosons}} + S_{\text{fermions}} + S_{\text{FT}}. \tag{22}$$

As mentioned above, the FT-term actually contributes only at next-to-leading order and is discussed in detail in Section 3.3.2. The first term in the world-sheet action is given by the Polyakov action

$$S_{\text{bosons}} = \frac{1}{4\pi\ell_s^2} \int \left( \gamma^{ij} G_{ij} \text{vol}_2 + 2i B_2 \right). \tag{23}$$

In this expression $\gamma$ is the world-sheet metric, $\text{vol}_2$ the volume form on the world-sheet and world-sheet indices are denoted by $i, j = 1, 2$. For future reference, curved target space indices are denoted by $\mu, \nu = 1, \ldots, 10$. The fields appearing in the action are understood as the pull-back of the ten-dimensional metric $G_{\mu\nu}$ and $B_{\mu\nu}$ field to the world-sheet. The factor $i$ multiplying the two-form $B$-field originates from the fact that we are working with an Euclidean world-sheet theory. As usual, the fermions vanish for the classical solution, but we will need their action to describe the fluctuations around the classical solution below. The Green-Schwarz (GS) action describes the appropriate coupling of ten-dimensional fermions collectively denoted by $\theta$, to the background geometry. For type II superstrings, the action takes the following form [23, 24]

$$S_{\text{fermions}} = \frac{i}{2\pi\ell_s^2} \int \bar{\theta} \mathcal{P}^{ij} \left( \Gamma_i \mathcal{D}_j + \frac{1}{8} \Lambda \Gamma_i^{\mu\nu} H_{j\mu\nu} + \frac{1}{8} e^{\Phi} \Gamma_i \mathcal{F} \Gamma_j \right) \theta \, d\sigma d\tau, \tag{24}$$

where $\Gamma_i$ and $\mathcal{D}_j$ denote the ten-dimensional gamma matrices and target space covariant derivative pulled back to the world-sheet respectively, and the projector $\mathcal{P}_{ij}$ is defined as follows

$$\mathcal{P}^{ij} = \sqrt{\gamma} \gamma^{ij} + i \Lambda \epsilon^{ij}. \tag{25}$$

In these expressions $\Lambda$ is given by

$$\Lambda_{\text{IIA}} = -\Gamma_{11}, \qquad \Lambda_{\text{IIB}} = \sigma_3, \tag{26}$$

for type IIA/B respectively. The quantity $\mathcal{F}$ containing the coupling to the RR fields on the other hand is given by

$$\mathcal{F}_{\text{IIA}} = -\Gamma_{11} \slashed{F}_2 + \slashed{F}_4, \qquad \mathcal{F}_{\text{IIB}} = -i\sigma_2 \slashed{F}_1 + \sigma_1 \slashed{F}_3 - \frac{i}{2} \sigma_2 \slashed{F}_5. \tag{27}$$

In type IIA, the spinor $\theta$ is a 32-component Dirac spinor, while in type IIB, the spinor $\theta = \theta^I$ consists of a pair of 16-component Majorana-Weyl spinors subject to the constraint, $\Gamma_{11}\theta^I = \theta^I$. The Pauli matrices $(\sigma_a)^{IJ}$, with $a = 1, 2, 3$, introduced above act on the indices $I, J = 1, 2$.

## 3.2 The classical action

Before specifying the string solutions, which we discuss in the next sections, we want to underline some of their common properties. In conformal coordinates, the induced world-sheet metric takes the form

$$ds_2^2 = e^{2\rho(\sigma)}\big(d\sigma^2 + d\tau^2\big), \tag{28}$$

where the explicit form of the function $\rho(\sigma)$ varies case by case. Note that $e^{2\rho(\sigma)}$ carries a dependence on the characteristic ten-dimensional length scale $L$. The coordinate $\sigma$ plays the role of the radial direction, while $\tau$ is an angle, $\tau \in (0, 2\pi)$. Since we have assumed $\tau$ to be an isometry direction, the centre of the world-sheet is where the $\tau$-circle shrinks to zero size. Locally, around the centre, we can write the metric in polar coordinates as

$$ds_2^2 = dr^2 + r^2 d\tau^2, \tag{29}$$

where $r \to 0$. Making a coordinate transformation back to conformal coordinates we see that locally $\sigma = -\log(r/r_0) \to \infty$ and $e^\rho = r = r_0 e^{-\sigma} \to 0$.[9] The parameter $r_0$ is the characteristic length scale associated with the geometry close to the centre, and is different case by case. Later on we will introduce a cut-off at finite, but large $\sigma$, and then $r_0$ will characterise the size of the disc that is cut off. This of course assumes that the topology of the world-sheet is a disc (or a sphere). More generally, on higher genus Riemann surfaces, we expect multiple locations where the conformal factor vanishes.

The classical action (23) reduces to

$$S_{\text{cl}} = \mathcal{A} + \mathcal{B}, \qquad \mathcal{A} = \frac{1}{2\pi\ell_s^2}\int \text{vol}_2, \qquad \mathcal{B} = \frac{i}{2\pi\ell_s^2}\int B_2, \tag{30}$$

where $\text{vol}_2 = e^{2\rho}\, d\sigma \wedge d\tau$. In all cases studied here, the on-shell value of the $B$-field vanishes and hence $\mathcal{B} = 0$. Since the naive string area diverges in the above expression, $\mathcal{A}$ denotes its suitably regularised version. One way of regularising it is by performing a Legendre transform in appropriate coordinates [25]. This procedure results in a finite leading order contribution that matches with the leading order Wilson loop expectation value [6].

## 3.3 One-loop contributions

### 3.3.1 Second order fluctuations

We now turn to the quadratic expansion of the string action around the classical configuration. We work in static gauge, that is, we identify two space-time coordinates with the world-sheet directions, which in turn freeze out the two scalar fluctuations tangent to the world-sheet. Therefore, we consider only the eight scalar fluctuations transverse to the world-sheet and denote them by $\zeta^a$. The quadratic action for the fermionic degree of freedom can be worked out from the action (24), where the couplings to the background fields should be pulled back to the classical string solution. After fixing the $\kappa$-symmetry gauge, we obtain eight two-dimensional fermions $\theta^a$. Hence, we collectively write the quadratic action in the following way

$$S_{\mathbb{K}} = \frac{1}{4\pi\ell_s^2}\int \big(\zeta^a \mathcal{K}_{ab}\zeta^b + \bar{\theta}^a \mathcal{D}_{ab}\theta^b\big)\text{vol}_2, \tag{31}$$

where $a, b = 1, \ldots, 8$ and $\mathcal{K}_{ab}$ and $\mathcal{D}_{ab}$ denote the kinetic operators of the scalars and fermions respectively. Upon quantisation, the partition function reduces to a product of determinants,

---

[9]Notice however that the D6 case does not follow this general lore. Instead, its conformal factor diverges in the IR, where $\sigma \to \infty$.

which we write as (see Appendix A for our conventions)

$$\text{Sdet}\,\mathbb{K} = \frac{\det \mathcal{K}_{ab}}{\det \mathcal{D}_{ab}}\,. \tag{32}$$

In the examples studied here, all operators can be diagonalised, i.e. $\mathcal{K}_{ab} = \delta_{ab}\mathcal{K}_{a,q}$ and $\mathcal{D}_{ab} = \delta_{ab}\mathcal{D}_q$. Since we work in conformal coordinates, it is convenient to introduce flat operators, denoted by a tilde and defined by

$$\begin{aligned}
\mathcal{K}_{a,q} &= \text{e}^{-2\rho}\,\tilde{\mathcal{K}}_{a,q}\,, & \tilde{\mathcal{K}}_{a,q} &= -D^2 + E_a(\sigma)\,, \\
\mathcal{D}_q &= \text{e}^{-3/2\rho}\,\tilde{\mathcal{D}}_q\,\text{e}^{\rho/2}\,, & \tilde{\mathcal{D}}_q &= i\slashed{D} + a(\sigma)\sigma_3 + v(\sigma)\,,
\end{aligned} \tag{33}$$

where $\slashed{D} = \sigma_1 D_\sigma - \sigma_2 D_\tau$, $D_i = \partial_i - iqA_i$, $A_i$ is a (in general $\sigma$-dependent) connection on the world-sheet which will be specified case-by-case, and $q$ is the charge of the relevant mode in question. Note that for all cases considered in this paper $A_\sigma = 0$ and so $D_\sigma = \partial_\sigma$. From the expressions above, it is manifest that the two sets of operators (tilded and untilded) are related by a Weyl rescaling of the conformal factor of the two-dimensional metric (28). The world-sheet theory is Weyl invariant, and this allows us to work with the flat operators $\tilde{\mathcal{K}}_{a,q}$ and $\tilde{\mathcal{D}}_q$, which are usually easier to handle. There are some caveats to this approach which are important to take into account. Performing the Weyl rescaling as above, does indeed render the world-sheet metric flat as desired. However, after this transformation we obtain a flat metric on a cylinder, not on a disc. This is because the centre of the world-sheet (located at $\sigma \to \infty$) has been pushed infinitely far away. This means that to ensure a discrete spectrum for the operators, we have to introduce an IR cut-off $R \gg 1$ as $\sigma \to \infty$ [16].

So far our discussion has been rather general, but let us now summarise some features of the operators encountered in this paper. For the string dual to the Wilson loop in MSYM on $S^d$, the fermions split into two sets of four identical fermions, where the two sets have opposite charge $q = \pm 1$ but all have the same "mass". The mass can be expressed in general in terms of the conformal factor $\rho$:

$$a^2 - v^2 = (\partial_\sigma \rho)^2 - 1\,. \tag{34}$$

Note that even though this squared expression is identical in all our cases the explicit forms of $a$ and $v$ are not universal and more complicated. Furthermore, six of the bosonic modes are particularly simple; they have zero charge, i.e. $q = 0$,[10] and masses given by

$$E_x = \partial_\sigma^2 \rho + (\partial_\sigma \rho)^2 - 1\,, \qquad d-1 \text{ times}, \tag{35}$$

$$E_y = -\partial_\sigma^2 \rho + (\partial_\sigma \rho)^2 - 1\,, \qquad 7-d \text{ times}. \tag{36}$$

The remaining two bosonic modes are defined by two operators $\tilde{\mathcal{K}}_{z,q}$ which differ by the sign of their charge $q = \pm 2$, but have the same mass that depends on the pull-back of the dilaton $\Phi_0$

$$E_z = E_y + \partial_\sigma^2 \big[(4-d)\rho + \Phi_0\big]\,, \qquad 2 \text{ times}. \tag{37}$$

The operators $\tilde{\mathcal{K}}_x$ act on the fluctuations along the field theory $d$-sphere, while $\tilde{\mathcal{K}}_y$ act on the fluctuations along some of the transverse "compact" directions (where the classical string solution is a point). More specifically for $d \neq 7$, the classical string is located where a $(6-d)$-sphere collapses to a point, and the $y$-modes correspond to fluctuations along the $(7-d)$-dimensional space composed of the sphere and the polar angle which locally acts as a radial coordinate. For $d = 7$ there are no $y$-modes and the classical string solution is given by any fixed point on the two-sphere. Finally, the remaining two operators correspond to fluctuations along the two-sphere (where the classical string solution is a point).

---

[10]We will drop the subscript $q = 0$ on the uncharged operators $\tilde{\mathcal{K}}_x$ and $\tilde{\mathcal{K}}_y$.

Introducing the first-order operators

$$\mathcal{L} = \partial_\sigma + \partial_\sigma \rho \,, \qquad \mathcal{L}^\dagger = -\partial_\sigma + \partial_\sigma \rho \,, \tag{38}$$

we see that

$$\mathcal{L}\mathcal{L}^\dagger = -\partial_\sigma^2 + \partial_\sigma^2 \rho + (\partial_\sigma \rho)^2 \,, \qquad \mathcal{L}^\dagger \mathcal{L} = -\partial_\sigma^2 - \partial_\sigma^2 \rho + (\partial_\sigma \rho)^2 \,. \tag{39}$$

Then $\tilde{\mathcal{K}}_a$ in (33) can be rewritten as

$$\begin{aligned}
\tilde{\mathcal{K}}_x &= -\partial_\tau^2 - 1 + \mathcal{L}\mathcal{L}^\dagger \,, \\
\tilde{\mathcal{K}}_y &= -\partial_\tau^2 - 1 + \mathcal{L}^\dagger \mathcal{L} \,, \\
\tilde{\mathcal{K}}_{z,q} &= -D_\tau^2 - 1 + \partial_\sigma^2 \big[ (4-d)\rho + \Phi_0 \big] + \mathcal{L}^\dagger \mathcal{L} \,.
\end{aligned} \tag{40}$$

The operators $\mathcal{L}, \mathcal{L}^\dagger$ (38) relate the two simpler sets of bosonic operators, i.e. $\mathcal{L}\tilde{\mathcal{K}}_y = \tilde{\mathcal{K}}_x \mathcal{L}$, $\mathcal{L}^\dagger \tilde{\mathcal{K}}_x = \tilde{\mathcal{K}}_y \mathcal{L}^\dagger$, and are often helpful to determine the spectrum (with the caveat that the boundary conditions obeyed by the eigenfunctions are also mapped correctly among the two sets).

We can also formulate the fermionic operators $\mathcal{D}_q$ in (33) in terms of the first order operators (38). Since the masses $a$ and $v$ obey (34), we may write

$$a = (\partial_\sigma \rho)\cosh \xi + \sinh \xi \,, \qquad v = (\partial_\sigma \rho)\sinh \xi + \cosh \xi \,, \tag{41}$$

where $\xi \equiv \xi(\sigma)$ is some $\sigma$-dependent function that differs case-by-case. If we also define the Pauli matrix raising and lowering operators $2\sigma_\pm \equiv \sigma_3 \pm i\sigma_1$, then

$$\tilde{\mathcal{D}}_q = e^{\frac{\xi \sigma_3}{2}} \Big( \sigma_+ \mathcal{L} + \sigma_- \mathcal{L}^\dagger - i\sigma_2 \big( D_\tau + \tfrac{i}{2}\partial_\sigma \xi \big) + 1 \Big) e^{\frac{\xi \sigma_3}{2}} \,, \tag{42}$$

which can be particularly helpful whenever $\xi$ vanishes as in the case for $d = 3$ below.

### 3.3.2 Fradkin-Tseytlin action

The Fradkin-Tseytlin (FT) term [21, 22] in the world-sheet action (22) couples the string to the background dilaton and is defined in terms of its pull-back

$$e^{2\Phi_0} \equiv P\big[ e^{2\Phi} \big] \,, \tag{43}$$

namely

$$S_{\text{FT}} = \frac{1}{4\pi} \int \Phi_0 R^{(2)} \text{vol}_2 + \frac{1}{2\pi} \int_\partial \Phi_0 K \, ds \,, \tag{44}$$

where $R^{(2)}$ is the two-dimensional Ricci scalar and $K$ is the extrinsic curvature on the boundary of the string world-sheet. It is clear from expression (44), that there is no explicit factor of the string length in the above action, hence its classical evaluation contributes at the same order as the other one-loop terms.

If the dilaton $\Phi_0$ is constant, then (44) gives $S_{\text{FT}} = \chi \Phi_0$, and its contribution to the partition function (21) is simply $g_s^{-\chi}$. In our work, the pulled back dilaton $\Phi_0$ is a non-trivial function of the world-sheet coordinate $\sigma$, and thus the full integral in (44) must be evaluated.

At this point we should recall that when discussing the one-loop fluctuations, we already performed a Weyl rescaling of the world-sheet metric that renders it flat. The term (44) then vanishes trivially since $R^{(2)}$ vanishes. This is too quick, however, since at the same time as performing the Weyl rescaling we had to introduce a cut-off $R$ that effectively removes a small

disc around the centre of the world-sheet. When evaluating (44), we should include the contribution originated from the small disc which was removed. It is straightforward to see that as $\sigma = R \to \infty$, the FT action becomes

$$\tilde{S}_{\mathrm{FT}} = \lim_{\sigma \to \infty} \Phi_0\,, \tag{45}$$

where we have denoted the Weyl rescaled action with a tilde. Remember that we are assuming the topology of the world-sheet to be that of a disc. On more general world-sheets we would have to cut out multiple discs around locations where the conformal factor vanishes and (45) receives contributions from each such point.

In [10] for the five-dimensional MSYM theory, the dilaton was treated in this way and lead to an agreement between field and string theory computations. There, the dilaton $\Phi_0$ approaches a finite value at the centre of the world-sheet. As shown in Table 3 for $d = 3, 7$, the limit $\lim_{\sigma \to \infty} \Phi_0$ diverges. We anticipate that this divergence also mixes with those coming from the functional determinant Sdet $\mathbb{K}$, which is described in more detail in Section 3.6.

### 3.4 Weyl anomaly and UV divergences

In this subsection we will recall the divergent structure of the one-loop determinants. We will keep our discussion fairly general and start by focusing on the operators before Weyl rescaling. It is useful to introduce $\Gamma_{\mathbb{K}}$ defined as

$$\Gamma_{\mathbb{K}} = \frac{1}{2} \log \mathrm{Sdet}\,\mathbb{K}\,. \tag{46}$$

The "effective action" $\Gamma_{\mathbb{K}}$, has logarithmic divergences (these are the usual UV divergences of a two-dimensional quantum field theory) and needs to be regularised. We can write

$$\Gamma_{\mathbb{K}} = (\Gamma_{\mathbb{K}})_{\mathrm{reg}} + (\Gamma_{\mathbb{K}})_{\infty}\,, \qquad (\Gamma_{\mathbb{K}})_{\mathrm{reg}} = \frac{1}{2} \left( \log \mathrm{Sdet}\,\mathbb{K} \right)_{\mathrm{reg}}\,, \tag{47}$$

where $(\Gamma_{\mathbb{K}})_{\infty} = -a_2(1|\mathbb{K}) \log \Lambda$ and $\Lambda$ is a UV regulator. The coefficients in front of these UV divergences can be computed by means of the second Seeley-DeWitt coefficients $a_2(1|\mathbb{K})$ [14, 26]. In conformal gauge, these logarithmic divergences are cancelled once ghosts, longitudinal fluctuations, measure factors as well as extra Jacobian factors for a local Lorentz rotation of the GS fermions are taken into account [14]. This is equivalent to what happens in the static gauge (though perhaps the cancellation is in general less transparent, due to the mixing of some contributions), and in particular the contribution to the logarithmic divergences from the eight transverse scalar fields equates that coming from all bosonic fluctuations and ghosts [14].

Said in other words, the whole one-loop string partition function $Z_{1-\mathrm{loop}}$ is expected to be finite, but due to the limited available techniques to compute it, we are left with UV divergences that must be dealt with. The explicit expression for the Seeley-DeWitt coefficients $a_2$ for an operator $\mathcal{O}$ evaluated on a test function $f$ is

$$a_2(f|\mathcal{O}) = \frac{1}{4\pi} \int f\, b_2(\mathcal{O}) \mathrm{vol}_2 + \text{boundary terms}, \tag{48}$$

where $b_2$ is the "local" Seeley-DeWitt coefficient. For the operators $\mathcal{K}$ and $\mathcal{D}$, in our conventions, they read [26]

$$b_2(\mathcal{K}) = \frac{1}{6} R^{(2)} - \mathrm{e}^{-2\rho}\, E\,, \qquad b_2(\mathcal{D}^2) = -\frac{1}{6} R^{(2)} - 2\, \mathrm{e}^{-2\rho}(a^2 - v^2)\,, \tag{49}$$

where the masses $E, v, a$ are defined in (33). Since we are going to work with the Weyl-rescaled operators $\tilde{\mathcal{K}}, \tilde{\mathcal{D}}$, it is useful to introduce

$$\mathrm{Tr}\left( (-1)^F M^2 \right) := \mathrm{e}^{-2\rho}\, \mathrm{Tr}(E) - \mathrm{e}^{-2\rho}\, \mathrm{Tr}(a^2 - v^2) = R^{(2)} + 2\, \mathrm{e}^{-2\rho}\, \partial_\sigma^2 \Phi_0\,. \tag{50}$$

This mass sum rule is a consequence of the background geometry satisfying the ten-dimensional supergravity equations of motion. For the tilded operators, the corresponding Seeley-DeWitt coefficient is simply given by integrating the mass rule (50) (up to an overall sign), that is

$$a_2(1|\tilde{\mathbb{K}}) = -\frac{1}{4\pi} \int \left( \text{Tr}(E) - \text{Tr}(a^2 - v^2) \right) d\sigma d\tau = \frac{1}{2\pi} \int \partial_\sigma^2 (\rho - \Phi_0) d\sigma d\tau, \qquad (51)$$

where we have used that in conformal coordinates the Ricci scalar is $R^{(2)} = -2\, e^{-2\rho} \partial_\sigma^2 \rho$. Two comments are in order here. First, it is clear that the structure of the UV divergences does not appear to take a universal form due to the explicit dependence on the dilaton. We will discuss this point in more detail below. The second comment is regarding the boundary terms which we have ignored so far. Clearly the first term in (51) should give the Euler characteristic which requires a boundary term similar to the one in (44) proportional to the extrinsic curvature $K$. In conformal coordinates the extrinsic curvature is proportional to $\partial_\sigma \rho$, and the boundary term is such that the contribution of the bulk and boundary terms cancels at the boundary. By analogy it is reasonable to expect that the correct boundary term associated with the dilaton is of the same form $\sim \partial_\sigma \Phi_0$ and its role is to cancel the contribution of the bulk integral at the boundary. This can be checked against an explicit computation of the computed divergence e.g. using the WKB method and it works out. Thus, to conclude, we can perform the explicit integration in (51), using that the conformal factor $\rho$ is independent of $\tau$. We then find[11]

$$a_2(1|\tilde{\mathbb{K}}) = \lim_{\sigma \to \infty} \partial_\sigma (\rho - \Phi_0), \qquad (52)$$

which can be slightly simplified if we use that $\lim_{\sigma \to \infty} \partial_\sigma \rho = -1 = -\chi$ on the disc.

In addition to predicting the UV divergences of the determinants, the Seeley-DeWitt coefficient $a_2$ (48) also determines the contributions of the fluctuations to the Weyl anomaly of the string [14]. Indeed, a change in the length scale induces a variation on the functional determinant for $\mathbb{K}$,

$$\delta \log \det \mathcal{K} = -2a_2(\delta \rho | \mathcal{K}), \qquad \delta \log \det \mathcal{D}^2 = -2a_2(\delta \rho | \mathcal{D}^2). \qquad (53)$$

In particular, in conformal coordinates (28), the variation of the effective action $\Gamma_{\mathbb{K}}$ with respect to the conformal factor, i.e. $\langle T_i^i \rangle_{\mathbb{K}} = 2\pi\, e^{-2\rho} \frac{\delta \Gamma_{\mathbb{K}}}{\delta \rho}$, is given by [10]

$$\langle T_i^i \rangle_{\mathbb{K}} = -\frac{1}{2} R^{(2)} + e^{-2\rho} \partial_\sigma^2 \Phi_0, \qquad (54)$$

where in the last step we used (50) to simplify the result. The FT action (44) is not invariant under Weyl transformations, and thus it contributes to the anomaly, that is in conformal coordinates we have

$$\left( T_i^i \right)_{\text{FT}} = -e^{-2\rho} \partial_\sigma^2 \Phi_0. \qquad (55)$$

Hence, the total contribution to the integrated trace anomaly is

$$\frac{1}{2\pi} \int \langle T_i^i \rangle \text{vol}_2 = \frac{1}{2\pi} \int \left( \langle T_i^i \rangle_{\mathbb{K}} + \left( T_i^i \right)_{\text{FT}} \right) \text{vol}_2 = -\chi, \qquad (56)$$

where we included the standard boundary term in terms of the extrinsic curvature.[12] This expression is universal in the sense that it depends only on the world-sheet Euler characteristic $\chi$. The total Weyl anomaly vanishes, as consistency of string theory requires. This is only

---

[11]We have to be careful here as we have implicitly assumed an orientation of the world-sheet.

[12]We are using the convention $T_{ij} = -\frac{4\pi}{\sqrt{\gamma}} \frac{\delta S}{\delta \gamma_{ij}}$ for a metric with components $\gamma_{ij}$.

Table 2: In this table we summarise the results for the integrated Weyl anomaly in the second column and the total Seeley-DeWitt coefficient $a_2$ which gives us the coefficient of the logarithmic UV divergences in $\Gamma_{\mathbb{K}}$ in the third column. In seven dimensions, the opposite orientation for the world-sheet is naturally selected resulting in the opposite sign for the Seeley-DeWitt coefficient $a_2$ (see Section 5).

| Dimension | $-\frac{1}{2\pi}\int T_i^i \mathrm{vol}_2$ | $-a_2(1|\tilde{\mathbb{K}})$ |
|---|---|---|
| $d = 2$ | $\chi$ | $\chi$ |
| $d = 3$ | $\chi$ | $2\chi$ |
| $d = 5$ | $\chi$ | $\chi$ |
| $d = 6$ | $\chi$ | $\chi$ |
| $d = 7^*$ | $\chi^*$ | $-2\chi^*$ |

apparent once we have added extra contributions due to the GS fermions as well as the ghost determinant in conformal gauge, see e.g. [14].

In the next sections, we discuss the one-loop quantisation of certain string world-sheets in the ten-dimensional background of spherical D$p$-branes [2]. In all the cases encountered here, we check that relation (56) holds. The points we would like to stress are the following. When the dilaton is a constant, as in AdS$_5 \times S^5$ or AdS$_4 \times \mathbb{C}P^3$, clearly the coefficient of the UV logarithmic divergences in the effective action $\Gamma_{\mathbb{K}}$, as well as the coefficient determining the Weyl anomaly is the same, cf. (52) versus (56). It therefore appears as if the UV divergences are universal like the Weyl anomaly. To proceed, one typically computes the regularised part of $\Gamma_{\mathbb{K}}$ and drops the divergent terms. In order to get a result that matches with the QFT prediction one has to take into account a regularisation factor [15], which we will come back to and discuss in more detail below.

The case of a non-AdS background, but with a pulled back dilaton that approaches a constant at the centre of the world-sheet, is essentially equivalent to the case of a constant dilaton from the point of view of the logarithmic divergences (since only derivatives of $\Phi_0$ appear in (52)). For our example of holographic duals to SYM on $S^d$, we find that in dimensions $d = 2, 5, 6$ ($d = 5$ was already studied in [10]), the derivative of the dilaton vanishes exactly at the centre of the world-sheet, such that the UV divergence agrees with the Weyl anomaly. Of course, we would expect that any regular background has a dilaton that approaches a constant at the centre. This is however not the case for $d = 3, 7$ because the derivative of the dilaton is proportional to the derivative of the metric function $\rho$. This means that the dilaton diverges at the centre, as already mentioned in Section 3.3.2. Furthermore, the behaviour of the dilaton also leads to a different UV divergence than usual. We collect these results in Table 2.

To summarise, as expected the integrated Weyl anomaly is controlled by the Euler characteristic $\chi$ (second column in Table 2). This indicates that the full Weyl anomaly of the string vanishes, ensuring its consistency. Concerning the logarithmic UV divergences in the one-loop partition function, we see that for $d = 3, 7$ there are additional contributions to their coefficients from the dilaton, leading to the unexpected coefficients. In the remaining cases $d = 2, 5, 6$, even though the dilaton has a non-trivial profile, it is exponentially suppressed at the centre of the world-sheet and does not contribute to the UV divergence.

## 3.5 Phase shifts

In this section we give a brief review of the phase shift method used to compute the quantum effective action (46). This allows us to ultimately express the functional determinants of the quadratic operators (33) in terms of an integral of the phase shifts of each operator over the

spatial momenta.[13]

The most straightforward way to obtain the functional determinants is to first solve the spectral problem[14]

$$\tilde{\mathcal{K}}\psi(\sigma,\tau) = (-\partial_\sigma^2 - D_\tau^2 + E(\sigma))\psi(\sigma,\tau) = \tilde{\lambda}\psi(\sigma,\tau).\tag{57}$$

This two-dimensional problem can be reduced further by expanding the wave-functions in modes along $\tau$, where the frequency $\omega$ is an integer for the bosonic operators, but a half-integer for the fermionic operators. Since the potential $E(\sigma)$ vanishes and the gauge field $A_\tau$ approaches a constant in the IR, as $\sigma \to \infty$, the wave-functions take the asymptotic form

$$\psi_\omega(\sigma) \to \mathcal{C}\sin(p\sigma + \delta(p,\omega)).\tag{58}$$

Here $p$ is related to the eigenvalue via $\tilde{\lambda} = (\omega - \omega_0)^2 + p^2$ with $\omega_0 = \lim_{\sigma\to\infty} qA_\tau(\sigma)$ and $\delta$ denotes the phase shift. As we have already discussed, in order to obtain a discrete spectrum, we impose a Dirichlet boundary condition at $\sigma = R \gg 1$, where $R$ is an IR cut-off. This gives rise to a momentum quantisation condition

$$pR + \delta(p,\omega) = \pi n, \quad \text{with} \quad n \in \mathbf{Z}.\tag{59}$$

With this at hand, one can express the logarithm of the determinant as a momentum integral over the phase shifts [16],[15]

$$\log\det\tilde{\mathcal{K}} = -\int_0^\Lambda dp\big(\coth(\pi p - i\pi\omega_0)\delta(p,\omega_0 + ip) + \coth(\pi p + i\pi\omega_0)\delta(p,\omega_0 - ip)\big),\tag{60}$$

where $\Lambda$ is the UV cut-off. Importantly, we only need to know the phase shift for $\omega = \omega_0 \pm ip$, i.e. for $\tilde{\lambda} = 0$. Hence, it is clear that in order to compute the determinant of $\mathcal{K}$, we do not need to completely solve the spectral problem for $\mathcal{K}$. A similar expression can be obtained for the fermionic operators, with the only meaningful difference being the half-integer frequencies, which leads to

$$\log\det\tilde{\mathcal{D}} = -\int_0^\Lambda dp\,(\tanh(\pi p - i\pi\omega_0)\delta(p,\omega_0 + ip) + \tanh(\pi p + i\pi\omega_0)\delta(p,\omega_0 - ip)).\tag{61}$$

For Hermitian operators, the phase shifts obey $\delta(p,\omega_0 + ip) = \delta(p,\omega_0 - ip)$ and since the bosonic operators studied in this paper are Hermitian, in the remainder of the paper we will denote the bosonic phase shifts corresponding to $E_a$ with $\delta_a(p)$. The fermionic operators, on the other hand, are not Hermitian and thus we adopt the notation $\delta_{f,+} = \delta(p,\omega_0 + ip)$ and $\delta_{f,-} = \delta(p,\omega_0 - ip)$ for the fermionic phase shifts. We may further simplify these expressions by using the fact that in examples studied in this paper, holographic duals to MSYM in $d = 2,3,7$, $\omega_0$ are integers for bosons and so $\coth(\pi p \pm i\pi\omega_0) = \coth(\pi p)$. On the other hand, $\omega_0$ turns out to be half-integer for the fermions and therefore $\tanh(\pi p \pm i\pi\omega_0) = \coth(\pi p)$. This shows that the fermions effectively behave as if they are scalars due to the gauge potential that is present.[16] Combining (60) and (61) appropriately, yields the following expression for the effective action

$$\Gamma_{\mathbb{K}} = -\int_0^\Lambda dp\,\coth(\pi p)\big(\delta_{\text{bos}} - \delta_{\text{ferm}}\big),\tag{62}$$

---

[13]A more detailed analysis on this topic can be found in [12,16].

[14]We continue the analysis by considering one of the bosonic operators, but a similar analysis can be performed for the case of the fermionic operators.

[15]We have dropped the extensive part proportional to $R$ here, but we will come back to it in Section 3.6.

[16]We note that for spherical Yang-Mills in $d = 5$ considered in [10], there was no gauge potential and thus $\omega_0 = 0$.

where $\delta_{\text{bos}}$ and $\delta_{\text{ferm}}$ denote the combination of all bosonic and fermionic phase shifts respectively.

The integral above has the expected UV divergence proportional to $\log\Lambda$ as long as the difference in phase shifts $\text{Tr}\big[(-1)^F\delta\big]$ vanishes for large $p$. In most cases however this does not happen and instead for $p\to\infty$ we have

$$\text{Tr}\big[(-1)^F\delta\big]\sim\frac{n\pi}{2}+\frac{m}{p}+\mathcal{O}(p^{-2}),\tag{63}$$

for some integers $n$ and $m$. Recall that phase shifts are only defined up to an integer multiple of $\pi$, but we fix that ambiguity by assuming that all phase shifts vanish in the $p\to 0$ limit.[17] See for example Appendix B, where the phase shifts are computed for strings in AdS space. One way to deal with the constant in the asymptotic expansion is to shift the fermionic phase shifts by $-n\pi/2$, eliminating the linear divergence in $\Lambda$, leaving only the expected logarithmic divergence, which we discuss momentarily. Although following this practice with the phase shift method has yielded correct results in many cases we argue that employing this shift in $\delta_{\text{ferm}}$ results in an incomplete finite term in $(\Gamma_{\mathbb{K}})_{\text{reg}}$. Instead, one should properly take into account the finite part of the divergent integral by evaluating the integral as stated above and simply drop the linear divergence in $\Lambda$. The difference between the two regularisation schemes can be seen explicitly by computing the term which we would have otherwise dropped (here we assume $\omega_0 = 0$)

$$\frac{n\pi}{2}\int_0^\infty \mathrm{d}p\,\tanh(\pi p)=-\frac{n}{2}\log 2+\frac{n\pi}{2}\Lambda\,.\tag{64}$$

Hence the shift in the fermionic phase shifts would result in an expression that differs by a multiple of $\log 2$. We can justify that $\zeta$-function regularisation of the above integral equals its finite part as follows

$$\frac{n\pi}{2}\int_0^\infty \mathrm{d}p\,\tanh(\pi p)=n\pi\sum_{k=1}^\infty(-1)^k\int_0^\infty \mathrm{d}p\,e^{-2k\pi p}+\frac{n\pi}{2}\int_0^\infty \mathrm{d}p=-\frac{n}{2}\log 2\,,\tag{65}$$

where the latter integral evaluates to zero when taking into account the Abel-Plana formula and $\zeta$-function regularisation. But in practice it is easier to simply drop the linear divergences when evaluting the $p$-integral.

This discussion implies that some previous computations of $(\Gamma_{\mathbb{K}})_{\text{reg}}$ have been off by $\log 2$ factors. In particular we have checked that the computations in [10] did miss such factors. However, since in that paper, a ratio of partition functions was computed and the same factors of $\log 2$ were missed for both regularised effective actions, the end result is still correct. Nonetheless, the $\log 2$ factors become especially important when we are not computing ratios of partition functions. For example if we want to compare the phase shift computation of the one-loop partition function to its evaluation through the heat kernel method, cf. Appendix B. Another example is when $\Gamma_{\mathbb{K}}$ does not suffer from logarithmic divergences, in which case there is no need to compute ratios of partition functions. As we shall see, this will be the case for the M2-brane calculation of $\Gamma_{\mathbb{K}}$ for the spherical D6-branes, since 3-dimensional theories do not suffer from any logarithmic divergences [28, 29]. Without the $\log 2$ factors we do not obtain the correct expression for the finite partition function.

---

[17]A reader familiar with scattering phase shifts may attribute the constant multiple of $\pi$ in (63) with the number of bound states in the scattering problem as stated by Levinson's theorem (see e.g. [27]). However we usually do not encounter bound states in our analysis. It is also worth noting that our potentials do not satisfy the assumptions of Levinson's theorem as usually stated and so it does not apply without modification.

## 3.6 Regularisation and scaling

Let us now return to the divergent behaviour of (62). From the analysis of the Seeley-DeWitt coefficient $a_2$, we conclude that at large $p$ we find

$$\text{Tr}\Big[(-1)^F \delta\Big] \to \frac{1}{p} \lim_{\sigma \to \infty} \partial_\sigma (\rho - \Phi_0). \tag{66}$$

Inserting this into the quantum effective action, we have

$$\Gamma_\mathbb{K} \to (\log \Lambda - R) + (\log \Lambda - R) \lim_{\sigma \to \infty} \partial_\sigma \Phi_0, \tag{67}$$

where we have introduced by hand the dependence on the IR cut-off $R$ as predicted by the Seeley-DeWitt expansion and used that $\partial_\sigma \rho = -1$ as $\sigma \to \infty$. As we have discussed, in order to obtain a string partition function which we can compare with the vacuum expectation value of the Wilson loop, the method proposed by [15] is to drop the universal divergences and replace them with a universal correction factor. An equivalent method is to compute a ratio of string partition functions, where the universal divergences and correction factors cancel.

When examining the divergences in (67), there are two issues that we must first overcome. First we observe that $R$ is a cut-off defined using the coordinate $\sigma$, and so is clearly not coordinate or diffeomorphism invariant. This issue was emphasised and resolved in [16]. We simply have to relate $R$ to an invariant quantity before dropping the divergent term. One example is the area of the disc that is cut off, but equally well we can just use the conformal factor itself, since it does indeed uniquely determine the area of the disc that is being cut off. Referring to the discussion below (29), we replace $\sigma = R$ with $\log(r_0) - \rho(R)$ in the first term only and obtain

$$\Gamma_\mathbb{K} \to \rho(R) + \log \Lambda + (\log \Lambda - R) \lim_{\sigma \to \infty} \partial_\sigma \Phi_0 - \log(r_0), \tag{68}$$

where we have slightly reordered and re-expressed the terms. The last term, $\log(r_0)$, can be rewritten in a suggestive manner

$$\log(r_0) = \lim_{\sigma \to \infty} (\rho - \sigma \partial_\sigma \rho). \tag{69}$$

Recall that we got it from expressing the cut-off in terms of a diffeomorphism invariant quantity and it is crucial in order to get the scaling of the partition function correct as we will see. However, it is not entirely enough. As emphasised in [15], a complete match with QFT in simple examples such as the $\frac{1}{2}$-BPS Wilson loop in $\mathcal{N} = 4$ SYM, needs a correction factor which depends on the world-sheet topology. This could have been anticipated, since $r_0$ depends on the length scale of the world-sheet and in order to obtain a dimensionless quantity we should divide by the string length $\ell_s$. There could be extra numerical factors involved in this and we find that using the phase shift method the correction factor needed is $\log(\pi/\ell_s)$.[18]

In the cases where the dilaton is trivial at the centre, such that the third term in (68) vanishes, the first two terms constitute all of the divergences of the problem. That is, $(\Gamma_\mathbb{K})_\infty = \rho(R) + \log \Lambda$ and we drop them. If the dilaton is non-trivial, it is not enough to simply drop the above two divergent terms, since we will still have divergences stemming from the dilaton itself.[19] In fact, if we combine the quantum effective action with the Fradkin-Tseytlin term we find the terms

$$\lim_{\sigma \to \infty} (\Phi_0 - \sigma \partial_\sigma \Phi_0) + \log \Lambda \lim_{\sigma \to \infty} \partial_\sigma \Phi_0, \tag{70}$$

---

[18]When using the heat kernel method the correction factor is $-\log(4\pi\ell_s)$.

[19]Indeed, for the $d = 3, 7$ cases studied here, the dilaton is non-trivial and diverges in the IR (see Table 3).

where we have effectively taken the $R \to \infty$ limit as remarkably the divergences in the first two terms cancel. The second term diverges in the $\Lambda \to \infty$ limit and we suggest that it should be dropped leaving only the finite terms. We do not have a first principle argument for dropping the divergence, but it does pass basic tests such as carrying the correct dependence on the string coupling constant.

In summary, the string quantum effective action at one-loop order is the combination of the Fradkin-Tseytlin action and the quantum effective action of the string fluctuations. Both are computed using Weyl rescaling to a flat world-sheet. One way to compute the quantum effective action of quadratic fluctuations is to use the phase shift method as in (62). For a running dilaton, both the quantum effective action and FT action diverge and a regularisation is needed. We suggest that after regularisation the one-loop quantum effective action is

$$\log Z_{\text{1-loop}} = -(\Gamma_{\mathbb{K}})_{\text{reg}} + \lim_{\sigma \to \infty} \left( \rho - \Phi_0 - \sigma \partial_\sigma (\rho - \Phi_0) + \log \frac{\pi}{\ell_s} \right). \tag{71}$$

Throughout this discussion we have assumed that the world-sheet topology is a disc. On a higher genus world-sheet the last term should be replaced with a sum over a similar expression evaluated at all points where the conformal factor vanishes. Around each such point we must cut out a small disc to regulate various factors in the string partition function as outlined above. To each disc there is an associated regulator that should be written in terms of the local geometric quantities, such as the dilaton and the conformal factor. Importantly, the sum should include the correction factor $\log \frac{\pi}{\ell_s}$ in each term.[20] For all holographic examples that have been explored in the past and for which the dilaton is trivial, the above expression gives the correct result. Consider for example a spherical world-sheet. Here the conformal factor vanishes at two points, the north and the south pole. Therefore, when computing the string partition function, we should get two copies of the last term evaluated at each pole of the sphere. We have verified that the regularisation procedure outlined above, including the $\log \frac{\pi}{\ell_s}$ factor, reproduces the regularisation factor $C(2)$, which was found in [17]. It also reproduces the factor originally introduced in [15], which was denoted by $C(1)$ in [17].

The interesting feature about (71) is that for many backgrounds the regularised quantum effective action is a pure number, i.e. it carries no dependence on the scales in the problem. This is certainly the case for the backgrounds studied in this paper dual to maximally supersymmetric Yang-Mills on $S^d$. This means that we can predict the scaling of the partition function with relatively little effort, by just evaluating the second term in (71). In particular for $d \neq 6$, we find that

$$\rho - \sigma \partial_\sigma \rho \sim \log \ell_s + \frac{1}{2(6-d)} \log \lambda, \qquad \Phi_0 - \sigma \partial_\sigma \Phi_0 \sim -\log N + \frac{8-d}{2(6-d)} \log \lambda, \tag{72}$$

which implies that the one-loop string partition function scales as

$$Z_{\text{1-loop}} \sim N \lambda^{\frac{d-7}{2(6-d)}}, \tag{73}$$

for $d \neq 6$. This should be supplemented with the classical action of the string which was computed in [6]. For $d = 6$ a similar analysis leads to[21]

$$Z_{\text{1-loop}} \sim N \, e^{\frac{4\pi^3}{3\lambda}}. \tag{74}$$

---

[20]We remind the reader that the correction factor is sensitive to the regularisation scheme used to compute $(\Gamma_{\mathbb{K}})_{\text{reg}}$.

[21]The exponential term in this formula should not be confused with the classical action term, which in this case scales as $e^{e^{8\pi^3/3\lambda}}$ [6].

Interestingly, the sub-leading expansion of the Bessel function, cf. (16), correctly reproduces this result. In the cases where this has been computed on the QFT side ($d = 3, 4, 5$) the scaling result (73) matches precisely. For all other cases, the formula (73) constitutes a sharp prediction for the sub-leading structure of the localisation results.

Of course this answer is incomplete, as we have not computed the numerical prefactor. The goal of the remainder of this paper is to attempt to verify the string theory prediction by analysing the matrix model beyond leading order in $\lambda$ for the case of $d = 7$, and to initialise the computation of the numerical prefactor that is missed by the scaling argument. The latter computation is done for $d = 2, 3, 7$ and will present significant challenges. This indicates that we may not have a complete understanding of the regularisation procedure of the divergences, which are associated to the running dilaton. To this end, it may be useful to have the M-theory point of view, when the dual string is of type IIA, and can be uplifted to an M2-brane.

## 4 M2-branes dual to Wilson loops

In the cases originating from type IIA string theory, it can be useful to uplift the background to eleven-dimensional supergravity and compute the Wilson loop expectation value by evaluating the partition function of a probe M2-brane which the string in IIA uplifts to. That is, the M2-brane wraps the string world-sheet, as well as the M-theory circle. The M2-brane action is given by

$$\hat{S} = \hat{S}_{\text{bosons}} + \hat{S}_{\text{fermions}}, \tag{75}$$

where the Polyakov action for M2-branes, as well as its coupling to the three-form $A_3$ takes the form

$$\hat{S}_{\text{bosons}} = \frac{1}{2(2\pi)^2 \ell_s^3} \int \left( \hat{\gamma}^{ij} \hat{G}_{ij} - 1 \right) \text{vol}_3 - \frac{i}{(2\pi)^2 \ell_s^3} \int A_3. \tag{76}$$

In order to avoid overloading the symbols we have introduced hats on the eleven-dimensional metric and the three-dimensional M2-brane quantities. However, we will use the indices $i, j = 1, 2, 3$ for world-volume indices here and hope not to cause confusion with the two-dimensional world-sheet indices. They should never appear explicitly in the same context. Finally, we identify the eleven-dimensional Planck length with the string length $\ell_s$. The fermionic action will be discussed momentarily.

As for the string, we are interested in the action of quadratic fluctuations of all modes of the M2-brane around a given classical configuration that satisfies the classical probe brane equations of motion. In static gauge, the world-volume metric of the M2-brane is given by the induced metric for the classical configuration $\hat{\gamma}_{ij} = \hat{G}_{ij}$, where we are using the same conventions as before, i.e. the world-volume indices appearing on higher-dimensional objects implies their pull-back to the world-volume of the brane.

In order to work out the action of quadratic fluctuations of the brane we split up the eleven-dimensional indices into tangent-space indices and normal bundle indices $a, b = 1, \cdots, 8$, using two orthogonal bases of orthonormal frame fields along the world-volume and the normal bundle. We consider only the transverse fluctuations of the M2-brane, which give rise to eight scalar fields living on the M2-brane world-volume defined by its classical leading order configuration. Performing the quadratic expansion of the bosonic M2-brane action we find

$$\hat{S}_{\text{bosons}} \approx \frac{1}{(2\pi)^2 \ell_s^3} \int \left( \text{vol}_3 - iA_3 - \hat{\mathcal{L}}_{\text{bosons}} \text{vol}_3 \right), \tag{77}$$

where the Lagrangian for the eight scalar fields takes the form[22]

$$\hat{\mathcal{L}}_{\text{bosons}} = -\frac{1}{2}(\hat{D}^a{}_b\phi^b)^2 + \frac{1}{2}\Big(\hat{R}^i{}_{aib} + K_a{}^{ij}K_{bij} - \frac{i}{3!}\epsilon^{ijk}\nabla_a G_{bijk} + \hat{A}_{Gica}\hat{A}_G{}^{ic}{}_b\Big)\phi^a\phi^b. \tag{78}$$

The scalars $\phi^i$ are in general charged with respect to a gauge field that is inherited from the embedding of the world-volume in the eleven-dimensional background. This is captured by the covariant derivative $\hat{D}^i{}_j$ which is a matrix that acts on the scalar fields as

$$\hat{D}^a{}_b\phi^b = \nabla\phi^a + \hat{A}^a{}_b\phi^b, \qquad \hat{A}_i{}^{ab} = \hat{\Omega}_i{}^{ab} + \hat{A}_{Gi}{}^{ab}, \qquad \hat{A}_{Gi}{}^{ab} = -\frac{i}{4}\epsilon_{ijk}G_4^{abjk}. \tag{79}$$

We also need the quadratic expansion of the fermionic action. To this end, we start from the fermionic action in [33], which has been reduced to quadratic order in [34,35]. The action at this order in Euclidean signature takes the form

$$\hat{S}_{\text{fermions}} = \frac{i}{(2\pi)^2\ell_s^3}\int \text{vol}_3\,\bar{\theta}\Big[\hat{\gamma}^{ij}\Gamma_i\hat{D}_j - \frac{i}{2}\varepsilon^{ijk}\Gamma_{ij}\hat{D}_k\Big]\theta, \tag{80}$$

where $\hat{\Gamma}_i = \partial_i X^M\hat{\Gamma}_M$ are the eleven-dimensional gamma matrices pulled back to the world volume, $\theta$ is a 32-component spinor in eleven dimensions and

$$\hat{D}_i = \partial_i + \frac{1}{4}\partial_i X^M\hat{\Omega}_M{}^{AB}\Gamma_{AB} - \frac{\partial_i X^M}{288}\Big(\Gamma^{PNKL}{}_M + 8\Gamma^{PNK}\delta_M^L\Big)G_{PNKL}. \tag{81}$$

Note that $\varepsilon^{ijk} = \epsilon^{ijk}/\sqrt{\hat{\gamma}}$ is the Levi-Civita tensor where $\epsilon^{ijk} = \{0, \pm 1\}$ is the standard Levi-Civita symbol.

This action can be simplified to a form that is similar to the string action discussed above. Let us start by defining

$$\mathcal{P}_{\pm}^{ij} = \frac{1}{3!}\Big(\hat{\gamma}^{ij} \pm \frac{i}{2}\varepsilon^{ijk}\Gamma_k\Big), \qquad \mathcal{P}_{\pm} = \frac{1}{2}(1 \pm i\Gamma_{(3)}), \qquad \Gamma_{(3)} = \frac{1}{3!}\varepsilon^{ijk}\Gamma_{ijk}. \tag{82}$$

Since $(\Gamma_{(3)})^2 = -1$, the operators $\mathcal{P}_{\pm}$ are projectors onto orthogonal subspaces, and $\mathcal{P}_+\mathcal{P}_- = 0$. We can then show that

$$\mathcal{P}_{\pm}^{ij}\Gamma_i = \frac{1}{3}\Gamma^j\mathcal{P}_{\pm}, \qquad \Gamma^j\mathcal{P}_{\pm} = \mathcal{P}_{\pm}\Gamma^j, \qquad \mathcal{P}_{\pm}^{ij}\Gamma_i\Gamma_j = \mathcal{P}_{\pm}. \tag{83}$$

Using this, the fermionic action can now be rewritten as

$$\hat{S}_{\text{fermions}} = \frac{2i}{(2\pi)^2\ell_s^3}\int \text{vol}_3\,\bar{\theta}_-\Gamma^j\hat{D}_j\theta_-, \tag{84}$$

where we have defined

$$\theta_- = \mathcal{P}_-\theta. \tag{85}$$

The natural $\kappa$-symmetry gauge is therefore $\theta_- = \theta$, and this is the gauge we will use in this paper. The next step is to deal with the modified fermionic derivative $\hat{D}_M$. It is a straightforward exercise to verify that

$$\frac{1}{288}\Big(\Gamma^{PNKL}{}_M + 8\Gamma^{[PNK}\delta_M^{L]}\Big)G_{PNKL} = \frac{1}{8}\slashed{G}_4\Gamma_M - \frac{1}{24}\Gamma_M\slashed{G}_4, \tag{86}$$

---

[22]See [30,31] for a similar formula for the quadratic expansion of the bosonic string action and [32] for the quadratic expansion of D3-branes. Note that the anti-M2-brane has opposite signs in front of the $A_3$ and $G_4$ terms in these formulae.

where we used that $\not{G}_4 = \Gamma^{PNKL} G_{PNKL}/4!$. Using this we have

$$\Gamma^i \hat{D}_i = \Gamma^i \partial_i + \frac{1}{4} \Gamma^i \hat{\Omega}_i{}^{AB} \Gamma_{AB} + \frac{1}{8}\big(\not{G}_4 - \Gamma^i \not{G}_4 \Gamma_i\big). \tag{87}$$

The second term gives rise to both the world-volume spin connection and possibly a gauge connection where the $AB$-indices are in the normal direction. The four-form could also contribute to a connection piece when it has legs along the world-volume. In fact, this could be made apparent by further processing this formula, but instead of doing this we will work directly with the above expression which is simple enough.

# 5  MSYM on $S^7$ and spherical D6-branes

In this section we analyse in detail seven-dimensional MSYM and its holographic dual. We start in Section 5.1 by obtaining the sub-leading corrections at strong coupling of the planar free energy and $\frac{1}{2}$-BPS Wilson loop from the matrix model. In Section 5.2, we focus on the dual theory, where we compute the one-loop partition function for the string and M2-brane dual to the $\frac{1}{2}$-BPS circular Wilson loop.

## 5.1  Field theory

Before diving into the holographic computation, let us exhibit in more detail the computation of the free energy and Wilson loop expectation value in the seven-dimensional MSYM theory. At leading order we recover the results of [6] and subsequently extend this to the first sub-leading order in the strong coupling expansion.

The matrix model obtained by localising the seven-dimensional MSYM was introduced in Section 3.3.2, which for the reader's convenience we repeat here. The matrix model partition function (5) reduces to [6, 36]

$$Z = \int_{\text{Cartan}} [\mathrm{d}\mu] \exp\left(-\frac{4\pi^4 N}{\lambda}\operatorname{Tr}\mu^2\right) Z_{1\text{-loop}}(\mu) Z_{\text{instantons}}(\mu),$$

$$Z_{1\text{-loop}} = \exp\left(\sum_{i=1}^N \sum_{j\neq i}^N \big(\log|\sinh\pi\mu_{ij}| + f(\mu_{ij})\big)\right), \tag{88}$$

where the function $f(\mu_{ij})$ is given by [37]

$$f(\mu_{ij}) = \frac{\pi\mu_{ij}^3}{3} - \mu_{ij}^2 \log\big(1 - e^{2\pi\mu_{ij}}\big) - \frac{\mu_{ij}\operatorname{Li}_2(e^{2\pi\mu_{ij}})}{\pi} + \frac{\operatorname{Li}_3(e^{2\pi\mu_{ij}}) - \zeta(3)}{2\pi^2}. \tag{89}$$

Since we are working in the large $N$ regime, Yang-Mills instanton corrections are exponentially suppressed and will be ignored from now on. The instantons were recently studied in [36] and we refer the reader to that work for further details. In order to obtain the one-loop partition function in the above form we had to carefully renormalise the bare 't Hooft coupling constant [6, 36].[23] The result of this procedure is that, unlike the bare coupling constant, the renormalised 't Hooft coupling $\lambda$ can take negative values. After renormalisation, the strong coupling regime which is characterised by large separation of the eigenvalues is found at small negative coupling, $\lambda^{-1} \to -\infty$. This might seem odd but is corroborated by the dual supergravity analysis.

---

[23]See also [20] for a similar discussion in the context of five-dimensional SYM.

From the above matrix model we can derive the saddle point equation (7), which in this case becomes

$$-\frac{8\pi^4 N}{|\lambda|}\mu_i = \sum_{j\neq i} 2\pi\left(1-\mu_{ij}^2\right)\coth(\pi\mu_{ij})\,. \tag{90}$$

Note that the left hand side of this equation provides a repulsive force, pushing the eigenvalues far apart in the regime where $\lambda^{-1}$ is large and negative. The right hand side provides a short distance attractive potential, forcing the eigenvalues to clump together in two peaks near the end-points of the eigenvalue distribution at $\mu = \pm\mu_\star$.

At leading order in the strong coupling expansion we can approximate $(1-\mu^2)\coth(\pi\mu)\approx -\mu^2$. Doing so, the large $N$ saddle point equation can be solved exactly, producing the leading order eigenvalue density at strong coupling

$$P^{\mathrm{LO}}(\mu) = \frac{1}{2}\left(\delta(\mu+b_7)+\delta(\mu-b_7)\right)\,, \tag{91}$$

where at leading order the end-points are located at

$$\mu_\star^{\mathrm{LO}} = b_7 = \frac{2\pi^3}{|\lambda|}\,. \tag{92}$$

At this order the strong coupling results are in perfect agreement with the supergravity results [6]

$$\langle W\rangle^{\mathrm{LO}} = e^{4\pi^4/|\lambda|}\,, \qquad F^{\mathrm{LO}} = -\frac{16\pi^{10}N^2}{3|\lambda|^3}\,, \tag{93}$$

where the superscript LO denotes that we are working at leading order in the strong coupling expansion $|\lambda| \to 0$. The goal of this section is to extend these results beyond leading order in the strong coupling expansion. To attain this goal we use a hybrid analytic/numerical approach. In Figure 1, we show the eigenvalue densities for $N = 200$ points and various values of $\lambda$. As $|\lambda|$ becomes smaller, the peaks move further and further away from the origin. In addition, the numerical eigenvalue densities show us that beyond leading order the peaks get a finite width. As we zoom in on the peaks, we see that this width is largely independent of both $\lambda$ and $N$.

The shape of the two peaks can in fact be studied using analytic methods. First it was proven in [20] that to leading order in large $N$ and up to exponentially suppressed corrections in $1/|\lambda|$ (but otherwise perturbatively exact in $\lambda$), the eigenvalues can be expressed as

$$\mu_I = \frac{2\pi^3}{|\lambda|} + \delta_I\,, \quad \mu_{N/2+I} = -\frac{2\pi^3}{|\lambda|} - \delta_I\,, \qquad \text{with} \qquad \sum_I \delta_I = 0\,, \quad \sum_I \delta_I^2 = \frac{N}{4}\,, \tag{94}$$

where $I = 1, \cdots, N/2$ and we have assumed that $N$ is even. This result allows us to find the first two moments of the full distribution

$$\langle\mu\rangle = 0\,, \qquad \langle\mu^2\rangle = \frac{4\pi^6}{\lambda^2} + \frac{1}{2}\,. \tag{95}$$

Higher moments seem difficult to determine reliably using analytic techniques. One could e.g. approximate the kernel by [20, 36]

$$G(\mu) = 2\pi(1-\mu^2)\operatorname{sgn}(\mu) + \mathcal{O}(e^{-\pi\mu})\,. \tag{96}$$

Doing so includes a small repulsive force pushing the eigenvalues slightly apart. Taking the continuum limit of the saddle point equation results in

$$\frac{4\pi^3}{\lambda}\mu = \int_{-\mu_\star}^{\mu} d\mu' P(\mu')\left(1-(\mu-\mu')^2\right) - \int_{\mu}^{\mu_\star} d\mu' P(\mu')\left(1-(\mu-\mu')^2\right)\,. \tag{97}$$

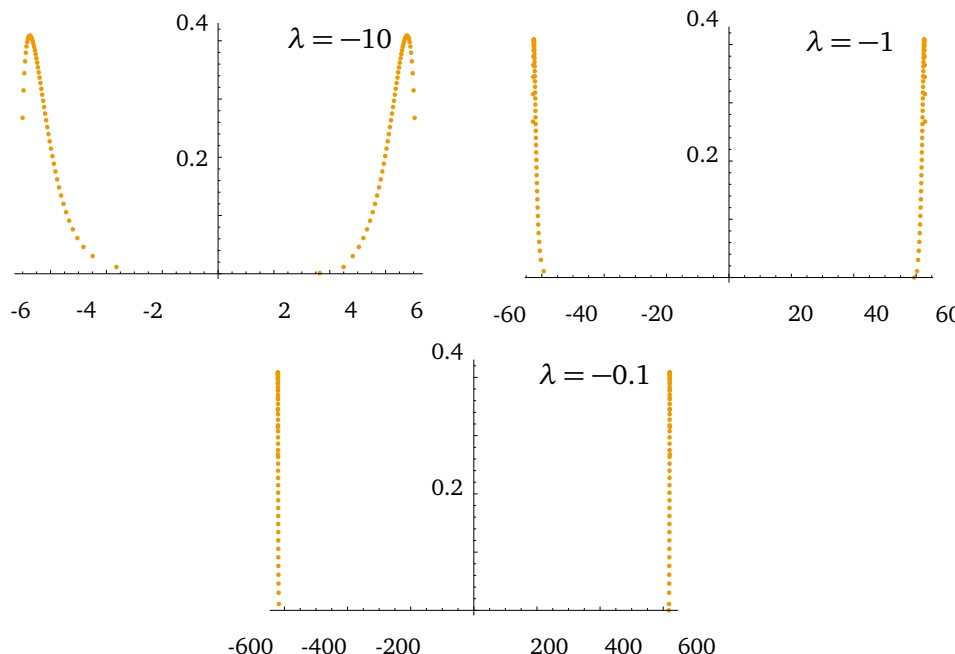

Figure 1: We plot the eigenvalue density for the leading saddle point satisfying the saddle point equation (90) for various values of $\lambda$ and for $N = 200$. The eigenvalues get pushed away from the origin as $|\lambda|$ becomes smaller. Note however that the shape of the peaks is largely independent of $N$ and $\lambda$. This is perhaps not clear from the plots as the peaks themselves are pushed further and further from each other.

This equation is solved by

$$P^{\mathrm{NLO}}(\mu) = \frac{k}{\sqrt{2}} \cosh \sqrt{2}\mu \,, \tag{98}$$

where $k$ is determined as the solution to the equation

$$\sinh\left( \sqrt{1+k^2} + \frac{1}{t} \right) = \frac{1}{k} \,, \tag{99}$$

and we defined $t = -\frac{\lambda}{2\sqrt{2}\pi^3}$. The end-points of the distribution $\mu_\star$ are then found by demanding the proper normalisation of the eigenvalue density. We can now solve these equations to get an estimate of the spread of the eigenvalue distribution. In order to solve equation (99), we need $k = \mathcal{O}\left( e^{-\frac{1}{t}} \right)$, so it is useful to define $k_0 = -t \log k$. We can then solve for $k_0$ and $\mu_\star$ perturbatively in $\frac{1}{t}$, resulting in the following expressions

$$k_0 = 1 + (1 - \log 2)t + \mathcal{O}\left( e^{-\frac{1}{t}} \right) \,, \qquad \mu_\star = \frac{1}{\sqrt{2}}\left( \log 2 + \frac{k_0}{t} \right) + \mathcal{O}\left( e^{-\frac{1}{t}} \right) \,. \tag{100}$$

The approximate distribution found by this method can be used to reproduce the perturbatively exact result (95), however higher moments are not accurately predicted. In order to compute the Wilson loop vacuum expectation value we do need access to all higher moments. Using the approximated density of eigenvalues is therefore bound to be imprecise. Nevertheless we will present the result of the computation for later comparison. Using (16), the Wilson loop vev is found to be

$$\log \frac{\langle W \rangle}{N} = \frac{4\pi^4}{|\lambda|} + \sqrt{2}\pi - \log\left( 2 + 2\sqrt{2}\pi \right) \approx \frac{4\pi^4}{|\lambda|} + 2.05543\ldots \tag{101}$$

As already mentioned, this result is quite far away from the numerical evaluation of the Wilson loop expectation value, which we turn to momentarily. However, it does get one feature correct; the sub-leading correction does not depend on the coupling constant $\lambda$. This is a rather peculiar feature that we also found evidence for from the string theory analysis. The main result of this analysis can therefore be expressed as

$$\langle W \rangle = wN\, e^{4\pi^4/|\lambda|}, \tag{102}$$

where $w$ is an undetermined numerical constant.

It turns out we can do substantially more when it comes to the free energy. Indeed, equipped only with the second moment we can compute the free energy through the large $N$ relation

$$\langle \mu^2 \rangle = -\frac{\lambda^2}{4\pi^4 N^2} \frac{\partial F}{\partial \lambda}, \tag{103}$$

where $F = -\log Z$ and the partition function $Z$ is defined through (88). Using (95) and integrating, results in

$$F = N^2 \left( -\frac{16\pi^{10}}{3|\lambda|^3} - \frac{2\pi^4}{|\lambda|} + f \right), \tag{104}$$

where $f$ is a constant that is yet to be determined. This expression is perturbatively exact in $\lambda$ to leading order in $N$! Once again we could evaluate $f$ using the approximate eigenvalue density discussed above, but since the computation of $f$ is sensitive to all higher moments just like the Wilson loop vev we do not expect an accurate result.

We therefore proceed with a careful numerical analysis of the matrix model, in order to check the expressions obtained above and add/correct the finite pieces in the strong coupling expansion. The numerical analysis proceeds as follows, for a given coupling constant $\lambda$, and rank $N$, we numerically solve equation (90) in terms of the eigenvalues $\mu_i$. We use a standard numerical solving algorithm (such as Newton's method) to find the numerical solution, provided a seed solution, which we take to consist of two peaks as given by the leading order solution in (91). From the numerical solution for the eigenvalues, we can compute the Wilson loop as well as the free energy using discrete forms of (16) and (17).

It is straightforward to extract the infinite contributions in the small negative $\lambda$ expansions and these match precisely with the predictions from our analytic results. Extracting the finite pieces requires a bit more care as our numerical results necessarily include $\frac{1}{N}$ corrections. To distinguish the leading and sub-leading parts in the $\frac{1}{N}$ expansion we compute the finite parts of the free energy and Wilson loop vev for a variety of values of $N$ and use this to fit to an assumed functional behaviour in $N$ and then extrapolate to $N \to \infty$. In Figure 2 we plot the results and find the following asymptotic values

$$w = 12 + \mathcal{O}\left(\frac{1}{N}\right), \tag{105}$$

$$f = 0.14\ldots + \mathcal{O}\left(\frac{1}{N}\right), \tag{106}$$

where the constants $w$ and $f$ are defined in (102) and (104) respectively.

In the case of the Wilson loop, the precision is noticeably better than for the free energy and we can obtain the constant $\log 12$ up to 10 digits. For the free energy on the other hand, the numerical errors become significant at large $N$, prohibiting us from pushing the precision and confidently identifying it with some known constant. Note that the resulting value $\log 12 \approx 2.4849\ldots$ deviates significantly from our analytic guess in equation (101) above. As already mentioned there, this is not entirely surprising, since our analytic approximation is not guaranteed to accurately describe the higher moments, which contribute significantly to

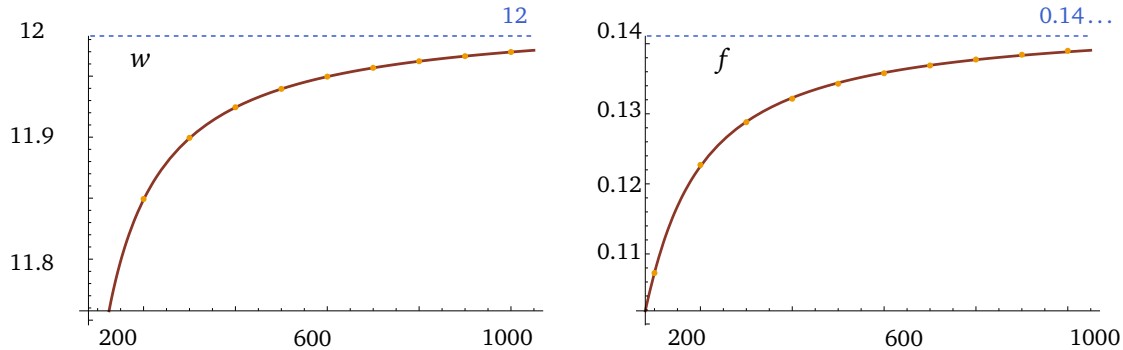

Figure 2: The free energy and Wilson loop vev for $\lambda = -0.1$ and various values of $N$ from 100 to 1000. The blue dashed line is the limiting value obtained after extrapolation. The red line is the fit with a large $N$ expansion of the form $X = X_{\text{finite}} + \frac{a_1}{N} + \frac{a_2}{N^2}$. For $\langle W \rangle$ and $F$ respectively the fitted values are given by $(w, a_1, a_2) = (12, -30.25, 25.52)$ and $(f, a_1, a_2) = (0.143, -4.68, 107.34)$.

this result. As the numerical result is more trustworthy, this will provide the target for our subsequent holographic computation.

We end this section by remarking that our results in (105) are derived using a single saddle for the eigenvalue distribution. As discussed in [6], other saddles are possible where (to leading order) some eigenvalues are zero, while the remaining ones are distributed in two peaks as described above. Our analysis of these saddles indicates that they lead to highly suppressed corrections to our result (105) in the large $N$ and small $|\lambda|$ limit.

## 5.2 M-theory and string theory

Let us now turn to the holographic solution dual to the seven-dimensional MSYM theory discussed above. As shown in [2], the eleven-dimensional solution is obtained as the analytic continuation of $\text{AdS}_4/\mathbf{Z}_N \times S^7$ to Euclidean signature

$$\text{d}s^2 = L^2 \left( \text{d}s_4^2 + 4\text{d}\Omega_7^2 \right), \tag{107}$$

where $\text{d}\Omega_7^2$ is the round metric on $S^7$ with unit radius. The four-dimensional part of the metric is given by

$$\text{d}s_4^2 = 4\text{d}\sigma^2 + \frac{1}{4}\sinh^2(2\sigma)\left( \text{d}\theta^2 + \sin^2\theta\,\text{d}\phi^2 + (\text{d}\omega + \cos\theta\,\text{d}\phi)^2 \right). \tag{108}$$

Here $\sigma \in (0, \infty)$ is the radial direction on $\text{AdS}_4$ (or $\mathbf{H}_4$), which we have rescaled by a factor 2 for later convenience. The angles $\theta$ and $\phi$ parametrise $\mathbf{C}P^1$ using spherical coordinates, and in addition to $\omega \in (0, \beta)$, the three angles parametrise the Lens space $S^3/\mathbf{Z}_N$ with $\beta = 4\pi/N$. Here the integer $N$ denotes the number of D6-branes in the ten-dimensional reduction, as we will see momentarily. The 11-dimensional four-form is given in terms of the volume form of the above four-dimensional metric as follows

$$G_4 = 3iL^3\text{vol}_4. \tag{109}$$

Notice that the length scale of $\text{AdS}_4/\mathbf{Z}_N$, which in usual ABJM holography would determine the number of M2-branes, is here related to the coupling constant of the seven-dimensional theory[24]

$$L^3 = \frac{2\pi^4 N}{|\lambda|}\ell_s^3. \tag{110}$$

---

[24]We identify the eleven-dimensional Planck length $\ell_p$ with the ten-dimensional string length $\ell_s$.

As we already discussed, the strong coupling in the QFT side is obtained when $|\lambda|^{-1} \gg 1$. In order to have positive definite metric we therefore express $L$ in terms of the absolute value of $\lambda$. As seen from this expression, observables computed in holography, using the eleven-dimensional solution, correspond to observables in 7D MSYM in the large $N/\lambda$ limit, but at finite $N$. The string theory limit of this background is more directly related to the matrix model computation discussed in previous sections as there the $\alpha'$ and $g_s$ expansions are directly related to the large $N$ and strong coupling limit $|\lambda|^{-1} \gg 1$. In ABJM holography, we would perform a dimensional reduction along the great circle of $S^7$ down to ten dimensions. This reduction can only be justified when the radius of that circle is small, which occurs when the Chern-Simons level $k$ is taken to be large in the ABJM theory. For our setup the great circle of $S^7$ is not small, however the circle parametrised by $\omega$ is small for large $N$ and so we perform the reduction over that angle. The dimensional reduction of (107) to ten dimensions takes the form

$$\mathrm{d}s_{10}^2 = \frac{8\pi^4 \ell_s^2}{|\lambda|} \sinh(2\sigma)\left( \mathrm{d}\sigma^2 + \mathrm{d}\Omega_7^2 + \frac{1}{16}\sinh^2(2\sigma)\mathrm{d}\Omega_2^2 \right), \tag{111}$$

where $\mathrm{d}\Omega_2^2$ is the round metric on the two-sphere with unit radius. The following fluxes are turned on

$$H_3 = \mathrm{d}B_2 = \frac{3i\pi^4 \sinh^3(2\sigma)\ell_s^2}{|\lambda|}\mathrm{d}\sigma \wedge \mathrm{vol}_2, \tag{112}$$

$$F_2 = \mathrm{d}C_1 = \frac{N\ell_s}{2}\mathrm{vol}_2, \tag{113}$$

where $\mathrm{vol}_2$ is the volume form on $S^2$, and the dilaton is given by

$$\mathrm{e}^{2\Phi} = \frac{2\pi^4}{|\lambda|N^2}\sinh^3(2\sigma). \tag{114}$$

We note that the presented supergravity solutions exhibit the same symmetries as the field theory dual, thereby passing the first test as the right dual geometry. In [6] a more detailed test was performed. Using holographic renormalisation, the on-shell supergravity action was computed for the above solution resulting in a match with the leading order term in (93).

Before discussing string and M2-brane holography beyond leading order, we should point out that the UV region of the geometry should be assigned to the region where the metric takes the form of D6-branes in flat space, which is at $\sigma \to 0$. This is however not the usual UV region of $\mathrm{AdS}_4$ in eleven dimensions, which is located at $\sigma \to \infty$. Indeed, the D6-branes are positioned at the centre of $\mathrm{AdS}_4$ where the orbifold singularity is located.[25]

### 5.2.1 String and M2-brane dual to the Wilson loop

Our main task in this section is to compute the expectation value of a Wilson loop operator in the fundamental representation, which wraps the equator of $S^7$ in 7D MSYM. The string configuration dual to this operator wraps the equator of the seven-sphere and extends along the radial direction $\sigma$. The complete embedding of the string is determined by selecting any fixed position on $S^2$. It is straightforward to verify that any position chosen minimises the string action, which is not surprising as the isometries of $S^2$ can be used to rotate the string around. We identify the two world-sheet coordinates with the ten-dimensional coordinates

---

[25]Related to this reversal of UV and IR is the fact that when computing the Euler characteristic of the string world-sheet one must integrate from $\sigma \to \infty$ to $\sigma = 0$ to get the correct sign. That is to say the natural orientation of the string should be used.

$(\sigma, \tau)$, where $\tau$ is the angle that parametrises a great circle on $S^7$. The resulting world-sheet metric in conformal coordinates and the pull-back of dilaton read

$$\mathrm{d}s_2^2 = \mathrm{e}^{2\rho}(\mathrm{d}\sigma^2 + \mathrm{d}\tau^2), \qquad \mathrm{e}^{2\rho} = \frac{8\pi^4 \ell_s^2}{|\lambda|}\sinh(2\sigma), \qquad \mathrm{e}^{2\Phi_0} = \frac{2\pi^4}{|\lambda|N^2}\sinh^3(2\sigma), \quad (115)$$

and we note that $\partial_\sigma \Phi_0 = 3\partial_\sigma \rho$.

We can also consider the M2-brane dual to the Wilson loop by uplifting the string embedding to eleven dimensions. The M2-brane then wraps the angle $\omega$ in addition to the string directions. The world-volume metric on the M2-brane is given by

$$\mathrm{d}s_3^2 = L^2\left(4\mathrm{d}\sigma^2 + \frac{1}{4}\sinh^2(2\sigma)\mathrm{d}\omega^2 + 4\mathrm{d}\tau^2\right), \qquad (116)$$

which is just the metric on $\mathrm{AdS}_2/\mathbf{Z}_N \times S^1$ where the radius of $\mathrm{AdS}_2$ is $L$ and the radius of $S^1$ is $2L$. It is important to note here that this configuration of M2-branes is exactly the same as the one used to describe the holographic dual of $\frac{1}{2}$-BPS Wilson-loop in the ABJM theory. In that case we should take $N = 1$, but possibly allow for modding of the seven-sphere to account for the difference in Chern-Simons level $k$. The one-loop quantisation of the corresponding string (in the large $k$ limit) was considered in [38–40], but recently the one-loop quantisation of the M2-brane for any $k$ was worked out in [41]. Unfortunately we are not able to borrow their results because the D6-brane number $N$ was trivial in their computation. We will however compare our results with theirs for the case of $N = 1$.

The on-shell string action can be computed using (30) and as usual it diverges and therefore has to be renormalised. This was already done in [6] and will not be repeated here. Instead we can compute the on-shell action of the M2-brane, and the answer we get should equal the string on-shell action. Using (76) we find

$$S_{\mathrm{M2}} = \frac{1}{(2\pi)^2 \ell_s^3}(4\pi L)\left(-\frac{2\pi L^2}{N}\right) = -\frac{4\pi^4}{|\lambda|}, \qquad (117)$$

where we have used that the volume of $\mathrm{AdS}_2/\mathbf{Z}_N$ is $-2\pi/N$, the relation (110), and that $\lambda$ is negative. This is indeed the same answer as obtained for fundamental strings in [6] and matches the exponential scaling of the fundamental Wilson loop in 7D MSYM (93).

### 5.2.2 One-loop string action

Now, our goal is to calculate the next-to-leading order contribution to the string partition function, namely $Z_{\text{1-loop}}$, given by equation (21). We start this discussion by expanding the string Lagrangian to quadratic order in the fields around the classical background. As summarised in Section 3.3.1, the second order theory consists of 8 scalar modes and 8 fermionic modes living on the string world-sheet. The dynamics of these modes are dictated by their masses and charge with respect to a background field living on the string. In this case the background field is pure gauge and takes the form

$$A = A_\tau \mathrm{d}\tau = \frac{3}{2}\mathrm{d}\tau, \qquad (118)$$

and descends from the $B$-field present in the ten-dimensional background.

As discussed in Section 3.3.1, six of the scalars are uncharged and all have the same mass $E_x$ defined in equation (35). The remaining two scalars have opposite charge $q = \pm 2$, but equal mass

$$E_z = -\partial_\sigma^2 \rho + (\partial_\sigma \rho)^2 - 1, \qquad (119)$$

where we have used (37) together with the fact that $\partial_\sigma^2(\Phi_0 - 3\rho) = 0$. For completeness we give the explicit form of all bosonic operators in the current context, namely

$$\tilde{\mathcal{K}}_x = -\partial_\sigma^2 - \partial_\tau^2 - \frac{1}{\sinh^2(2\sigma)}\,, \qquad \tilde{\mathcal{K}}_{z,q} = -\partial_\sigma^2 - (\partial_\tau - iqA_\tau)^2 + \frac{3}{\sinh^2(2\sigma)}\,. \tag{120}$$

Notice that we have performed the Weyl rescaling that removes the metric function entirely, and these operators are defined on a flat world-sheet. Since the combination $qA_\tau$ is integer for the two charged bosons, we can perform a simple $\tau$-dependent field redefinition that absorbs the gauge field, leaving a pair of uncharged modes with the same mass. As for the fermions, we find that they are all charged with respect to the background gauge field with charge $q = \pm 1$, each of which appear with degeneracy 4. Explicitly the fermionic operators are

$$\tilde{\mathcal{D}}_q = i(\slashed{\partial} - iq\slashed{A}) - \frac{1}{\sinh(2\sigma)}\sigma_3\,. \tag{121}$$

By combining the above results, the contribution of the string fluctuations around the classical solution to the one-loop effective action is given by

$$\Gamma_{\mathbb{K}} = \log \frac{\left(\det\tilde{\mathcal{K}}_x\right)^3 \left(\det\tilde{\mathcal{K}}_{z,+}\right)^{1/2} \left(\det\tilde{\mathcal{K}}_{z,-}\right)^{1/2}}{\left(\det\tilde{\mathcal{D}}_+\right)^2 \left(\det\tilde{\mathcal{D}}_-\right)^2}\,. \tag{122}$$

As explained in Section 3, in order to assemble the one-loop string partition function we also need to include the contribution of the dilaton. Evaluating the FT term directly, we encounter a problem, namely the dilaton (114) diverges in the IR $\sigma \to \infty$, which directly leads to a divergence in the FT term itself. This divergence also shows up in the one-loop determinant (122) as explained in Section 3.6. We can follow the regularisation procedure suggested in Section 3.6 to obtain a finite answer. To this end we need to evaluate

$$\rho - \Phi_0 - \sigma\partial_\sigma(\rho - \Phi_0) + \log\frac{\pi}{\ell_s}\,, \tag{123}$$

in the IR and subtract the regularised value of $\Gamma_{\mathbb{K}}$. The factor (123) evaluates in the IR to $\log(4\pi N)$, which means that the regularised 1-loop partition function of the string is

$$Z_{\text{1-loop}} = 4\pi N\, e^{-(\Gamma_{\mathbb{K}})_{\text{reg}}}\,. \tag{124}$$

Clearly the scaling of the partition function agrees with the one obtained from the QFT side, which is already a good sign. In order to get a perfect match with the QFT the regularised partition function should evaluate to $\log(\pi/3)$. To check this, we compute the regularised partition function using the phase shifts as discussed in Section 3.5. Solving the scattering problem for our operators leads to the explicit expressions

$$\begin{aligned}
\delta_{0,0}^{\text{bos}}(p) &= \text{Im}\log\Gamma(1 + \tfrac{ip}{2}) + \text{Im}\log\Gamma(\tfrac{1}{2} - \tfrac{ip}{2})\,, \\
\delta_{0,\pm 2}^{\text{bos}}(p) &= \text{Im}\log\Gamma(1 + \tfrac{ip}{2}) + \text{Im}\log\Gamma(\tfrac{3}{2} - \tfrac{ip}{2})\,, \\
\delta_{0,\pm 1}^{\text{ferm}}(p) &= \text{Im}\log\Gamma(\tfrac{1}{2} + \tfrac{ip}{2}) + \text{Im}\log\Gamma(1 - \tfrac{ip}{2})\,,
\end{aligned} \tag{125}$$

where the second index on the phase shifts $\delta$'s denotes the charge of the operators in (120) and (121), but the first index is introduced to be consistent with phase shifts for the M2-brane considered in the next subsection.

Assembling the phase shifts into the quantum effective action (62) according to (122), we must remember that due to the 1/2-integer gauge potential (118), the fermions are effectively quantised as if they are scalars. In the language of Section 3.5, we find that $\omega_0 = 0$ for the

uncharged scalars, $\omega_0 = \pm 3$ for the two charged scalars, and $\omega_0 = \pm 3/2$ for the fermions. Using this we find the regularised value $(\Gamma_{\mathbb{K}})_{\mathrm{reg}} = 3\log(2\pi)$.[26] This result is obtained by performing a numerical integration to high accuracy and matching this to the analytic answer up to 10 digits. Unfortunately, this is far from the value $\log(\pi/3)$ expected from the QFT and so at the level of the numerical constant we do not find a perfect match.

One possible explanation is that the dilaton is indeed diverging, as explained above. One way to circumvent the diverging dilaton is to uplift the string background to M-theory where the dilaton becomes a metric function. The Wilson loop expectation value can also be computed in eleven dimensions as the partition function of an M2-brane. In the next subsection we will therefore consider the M2-brane uplift of the string studied so far.

### 5.2.3 One-loop M2 action

Carrying out the expansion of the M2 action around its classical configuration is analogous to the corresponding computation for the string. Before presenting the results for the M2 expansion we recall that since the eleven-dimensional geometry dual to the spherical D6-branes is just $\mathrm{AdS}_4/\mathbf{Z}_N \times S^7$, and the M2 configuration wraps $\mathrm{AdS}_2/\mathbf{Z}_N$ times the equator of $S^7$, the M2-brane and its fluctuations are the same as for the M2-brane dual to Wilson loops in ABJM theory which were considered in [41, 42]. The difference is that here we have $\mathrm{AdS}_4/\mathbf{Z}_N \times S^7$ and not $\mathrm{AdS}_4 \times S^7/\mathbf{Z}_k$. This will affect the one-loop partition function of the M2-brane, but not the spectrum of fluctuations. We could therefore recycle the results of [41, 42] for the fluctuation operators. We find it useful however to use slightly different coordinates in eleven dimensions and derive the operators using equations from Section 4. Our operators will show minor differences from the ones in [41, 42], which can however be recovered by a coordinate transformation and a field redefinition.

For the M2 expansion we use static gauge with the world-volume metric given by (116). We find that the fermionic spectrum is particularly simple, given by 8 massless fermions in three dimensions. These fermions are however charged with respect to a background gauge field, which as for the string, is now also pure gauge but gains one extra component compared to (118)

$$\hat{A} = \frac{3}{2}\mathrm{d}\tau + \frac{1}{4}\mathrm{d}\omega \, . \tag{126}$$

Here the hat denotes three-dimensional quantities relevant for the M2-brane, and un-hatted quantities will refer to the corresponding string quantities. The operator that acts on the fermionic modes is

$$\hat{\mathcal{D}}_{(q)} = i\hat{\slashed{D}}_{(q)} = i(\hat{\slashed{\nabla}} - iq\hat{\slashed{A}}) \, , \tag{127}$$

where $\hat{\slashed{\nabla}}$ is the standard Dirac operator on $\mathrm{AdS}_2/\mathbf{Z}_N \times S^1$ written in the coordinates (116). We find that the charges of the fermions are $q = \pm 1$ each coming with degeneracy 4. The slashes here are performed using the three-dimensional gamma matrices, which are obtained as the pull-back of ten-dimensional gamma matrices to the world-volume.

Just like the string, the fluctuations of M2-branes involve 8 scalar modes. Performing the quadratic expansion of the M2-brane action, we compute the kinetic operators of the scalar fields, which take the form

$$\hat{\mathcal{K}}_{(q)} = -\hat{D}^2_{(q)} + \hat{M}^2 \, , \qquad \hat{M}^2 L^2 = -\frac{1}{4} \, . \tag{128}$$

Here $\hat{D}$ is the same gauge covariant differential operator that appeared in the fermionic operators (127). In particular, it also involves the background gauge connection $\hat{A}$. Notice that

---

[26]If we ignore the effect of the constant gauge potential we obtain $(\Gamma_{\mathbb{K}})_{\mathrm{reg}} \approx 6.176843$.

even though all scalar fields have the same mass, six of the bosonic modes have charge $q = 0$, but two remaining modes have charge $q = \pm 2$.

In order to make a connection to the string operators for the D6-case we perform a KK reduction along the $\omega$-coordinate. The three-dimensional metric is already flat along the string directions parametrised by the coordinates $\tau$ and $\sigma$, and so we only need to Weyl rescale the metric by a constant factor $L^2/4$ to end up with (a generalisation of) the Weyl rescaled operators $\tilde{\mathcal{K}}$ and $\tilde{\mathcal{D}}$ in (120) and (121). In order to complete the explicit map to the two-dimensional operators we must also rescale both the bosonic, and fermionic wave-functions by a factor

$$h(\sigma) \equiv \frac{1}{\sqrt{\sinh(2\sigma)}} \,. \tag{129}$$

We then find the operators

$$
\begin{aligned}
\tilde{\mathcal{K}}_{k,q} &\equiv 4L^2 \frac{1}{h} \hat{\mathcal{K}}_{(q)} h = -\partial_\sigma^2 - (\partial_\tau - iqA_\tau)^2 - \frac{1 - (4i\partial_\omega - q)^2}{\sinh^2(2\sigma)} \,, \\
\tilde{\mathcal{D}}_{k,q} &\equiv \frac{2Li}{h} \hat{\mathcal{D}}_{(q)} h = i(\slashed{\partial} - iq\slashed{A}) + \sigma_3 \frac{(4i\partial_\omega - q)}{\sinh(2\sigma)} \,.
\end{aligned}
\tag{130}
$$

If we momentarily assume that the scalar and fermionic wave-functions are independent of $\omega$, we effectively set $\partial_\omega$ to zero in (130) and recover the two-dimensional kinetic operators for a string in the D6-brane geometry.

We would like to consider the full three-dimensional fluctuations of the M2-brane modes. To this end we Fourier expand the wave-functions along the $\omega$-direction. We then recover a tower of two-dimensional operators (labelled by $k$), which are obtained by replacing

$$i\partial_\omega \mapsto \frac{Nk}{2} \,, \tag{131}$$

in (130). We can then summarise the operators at each level as taking the form

$$
\begin{aligned}
\tilde{\mathcal{K}}_{k,q} &= -\partial_\sigma^2 - (\partial_\tau - iqA_\tau)^2 + E_{k,q} \,, & E_{k,q} &= -\frac{1 - (2kN - q)^2}{\sinh^2(2\sigma)} \,, \\
\tilde{\mathcal{D}}_{k,q} &= i(\slashed{\partial} - iq\slashed{A}) + a_{k,q}\sigma_3 \,, & a_{k,q} &= \frac{(2kN - q)}{\sinh(2\sigma)} \,,
\end{aligned}
\tag{132}
$$

where the charges $q$ and associated degeneracies are

$$q_{\text{bosons}} = \{0_{\times 6}, -2_{\times 1}, 2_{\times 1}\} \,, \qquad q_{\text{fermions}} = \{-1_{\times 4}, 1_{\times 4}\} \,. \tag{133}$$

It is interesting to note that at each level $k$, we can view these as two-dimensional operators living on a world-sheet defined by the metric $e^{2\rho}$. This is just an extension of the string operators, which we recover at level $k = 0$. Here we find that the anomaly contributed at each level is

$$e^{-2\rho}(\text{Tr}\, E - \text{Tr}\, a^2) = -2R^{(2)} \,, \tag{134}$$

and is independent of the level $k$. Summing over all modes (assuming $k$ is an integer for both bosons and fermions) and regularising using standard $\zeta$-function regularisation, the total anomaly of the entire tower exactly vanishes. This is as expected, since the three-dimensional theory living on the M2-brane is anomaly-free. Furthermore, this also gives us hope that we can assign an unambiguous value to the M2-brane partition function even though the string partition function is somewhat problematic as discussed above.

### 5.2.4 M2-brane partition function using phase shift method

In this section, we compute the M2-brane partition function using the phase shift method as described in Section 3.5. In addition, in Appendix C, we present an alternative derivation that reaches the same conclusion using the Gel'fand-Yaglom method. We start by finding the homogenous wave-functions satisfying

$$\tilde{\mathcal{K}}_{k,q}\psi_B = 0\,, \qquad \tilde{\mathcal{D}}_{k,q}\psi_F = 0\,. \tag{135}$$

The solutions are subject to regularity conditions at $\sigma \to 0$, where the corresponding potentials blow up. It is a simple matter to verify that the operators $\tilde{\mathcal{K}}_{k,q}$ and $\tilde{\mathcal{D}}_{k,q}$ become free in the limit $\sigma \to \infty$ and the wave-functions approach plane waves. The wave-functions can be written explicitly in terms of hypergeometric functions, but we refrain from writing their explicit form here (see however Appendix C). Yet, once the wave-functions are found, it is a simple matter to read off the phase shifts, which take the form

$$\delta_{k,q}^{\text{bos}}(p) = \operatorname{Im}\log\Gamma(1+\tfrac{ip}{2}) + \operatorname{Im}\log\Gamma(\tfrac{1}{2}+\tfrac{|2kN+q|}{2}-\tfrac{ip}{2})\,,$$
$$\delta_{k,q}^{\text{ferm}}(p) = \operatorname{Im}\log\Gamma(\tfrac{1}{2}+\tfrac{ip}{2}) + \operatorname{Im}\log\Gamma(\tfrac{1}{2}+\tfrac{|2kN+q|}{2}-\tfrac{ip}{2})\,. \tag{136}$$

These phase shifts match with the string phase shifts in (125) for $k = 0$ as they should. We verify that the asymptotic behaviour of the phase shifts, when summed over all modes at a given level $k$ is

$$\lim_{p\to\infty}\left(\sum_q \delta_{k,q}^{\text{bos}}(p) - \sum_q \delta_{k,q}^{\text{ferm}}(p)\right) \sim (2+\delta_{k0})\pi - \frac{2}{p}\,, \tag{137}$$

where the multiple of $\pi$ gives rise to linear divergences and ultimately contribute important factors of $\log 2$ as discussed in Section 3.5 and should not be disregarded. Using (60) and (61) we conclude that, at each level $k$, the two-dimensional partition function diverges as $-2\log\Lambda$ as we found for the string. When summing over all modes of the M2-brane these divergences will however drop out as we will see momentarily. The same happens to almost all of the linear divergences. One factor of $\pi$ is left because the level $k = 0$ has one extra.

The full M2-brane effective action is given by a combination of a sum over $k$ and an integral over $p$

$$\Gamma_{\mathbb{K}} = -\sum_{k=-\infty}^{\infty}\int_0^\Lambda dp\,\coth(\pi p)\Big[\delta_{k,\text{bos}}(p) - \delta_{k,\text{ferm}}(p)\Big]\,, \tag{138}$$

where we have used the fact that for each level $k$ we find the same $\omega_0$ (in the language of Section 3.5) as for the string, meaning that fermions are effectively treated as if they satisfy periodic boundary conditions along the $\tau$ direction just like the scalars. We briefly mention in Appendix C how the result changes if we ignore the constant frequency shift $\omega_0$. In (138) we have also explicitly summed over all modes

$$\delta_{k,\text{bos}}(p) \equiv 6\delta_{k,0}^{\text{bos}}(p) + \delta_{k,2}^{\text{bos}}(p) + \delta_{k,-2}^{\text{bos}}(p)\,,$$
$$\delta_{k,\text{ferm}}(p) \equiv 4\delta_{k,1}^{\text{ferm}}(p) + 4\delta_{k,-1}^{\text{ferm}}(p)\,. \tag{139}$$

In order to evaluate the partition function in (138), we first note that any term which is independent of $k$ that is summed over can be dropped when regulating the infinite sum using standard $\zeta$-function regularisation

$$\sum_{k=-\infty}^{\infty} 1 = 1 + 2\zeta(0) = 0\,. \tag{140}$$

This means that we can significantly simplify the phase shifts as the first terms in (136) are all $k$-independent. This has the added benefit of eliminating the logarithmic divergences at each level. Next, we proceed by switching the order of the sum and the integral. But before doing so, it is convenient to first integrate (138) by parts, leaving

$$\Gamma_{\mathbb{K}} = \sum_{k=-\infty}^{\infty} \int_0^{\infty} \frac{\mathrm{d}p}{\pi} \log \sinh(\pi p) \Big[ \delta'_{k,\text{bos}}(p) - \delta'_{k,\text{ferm}}(p) \Big] + \log 2 , \tag{141}$$

where the $\log 2$ arises along a divergent boundary term when integrating by parts. Most boundary terms drop out when we sum over $k$ as they are $k$-independent, but due to the extra factor of $\pi$ at the level where $k = 0$ in the asymptotic expansion (137), we are left with the linear divergence and the $\log 2$ above. We have dropped the linear divergence as discussed in Section 3.5. As we will see, this means that the remaining integral is convergent, which allows us to take $\Lambda \to \infty$. Combining modes with positive and negative charge and evaluating the derivatives of the simplified phase shifts we are interested in evaluating

$$S_N(p;q) \equiv - \sum_{k=-\infty}^{\infty} \frac{1}{2} \text{Re} \left( \psi(\tfrac{1}{2} + \tfrac{|2kN+q|}{2} - \tfrac{ip}{2}) + \psi(\tfrac{1}{2} + \tfrac{|2kN-q|}{2} - \tfrac{ip}{2}) \right), \tag{142}$$

where $\psi(x) = \Gamma'(x)/\Gamma(x)$ is the polygamma function and the sums of the derivative of the phase shifts are

$$\sum_{k=-\infty}^{\infty} \delta'_{k,\text{bos}}(p) = 3 S_N(p;0) + S_N(p;2) ,$$
$$\sum_{k=-\infty}^{\infty} \delta'_{k,\text{ferm}}(p) = 4 S_N(p;1) . \tag{143}$$

The effective action is now given by

$$\Gamma_{\mathbb{K}} = \int_0^{\infty} \frac{\mathrm{d}p}{\pi} \log \sinh(\pi p) \Big[ 3 S_N(p;0) + S_N(p;2) - 4 S_N(p;1) \Big] + \log 2 . \tag{144}$$

We can slightly simplify $S_N(p;q)$, by noting that $|q| \le 2$ and $N \in \mathbf{N}, k \in \mathbf{Z}$, therefore

$$S_N(p;q) = - \text{Re}\, \psi(\tfrac{1}{2} + \tfrac{|q|}{2} - \tfrac{ip}{2}) - \sum_{k=1}^{\infty} \text{Re} \left( \psi(\tfrac{1}{2} + \tfrac{2kN+|q|}{2} - \tfrac{ip}{2}) + \psi(\tfrac{1}{2} + \tfrac{2kN-|q|}{2} - \tfrac{ip}{2}) \right). \tag{145}$$

The last sum in this expression diverges, but its third derivative with respect to $p$ is convergent. We will use this fact to evaluate the $\zeta$-function regularised version of $S_N(p;q)$. Consider the function

$$s_N(x) \equiv \sum_{k=1}^{\infty} \left[ \psi(kN + x) - \log(kN) - \frac{2x-1}{2kN} \right], \tag{146}$$

which is clearly inspired by the form of $S_N(p;q)$, but is however defined in terms of a convergent sum. By a formal manipulation of the $S_N(p;q)$, which is allowed for its regularised version, we can show that

$$S_N(p;q) + \text{Re}\, \psi(\tfrac{1}{2} + \tfrac{|q|}{2} - \tfrac{ip}{2}) + \text{Re} \left( s_N(\tfrac{1+|q|-ip}{2}) + s_N(\tfrac{1-|q|-ip}{2}) \right) = -2 \sum_{k=1}^{\infty} \log(kN) = \log \frac{N}{2\pi} . \tag{147}$$

This implies that in order to determine the $\zeta$-function regularised value for $S_N(p;q)$ we only have to simplify and evaluate $s_N(x)$. As it turns out, the infinite sum in $s_N(x)$ can be simplified to a finite one

$$s_N(x) = 1 - \frac{1}{2} \log(2\pi) + \frac{1-\gamma_E}{2N}(2x-1) - \frac{1}{N^2} \sum_{l=1}^{N} (N-l+x) \psi(\tfrac{2N-l+x}{N}) , \tag{148}$$

where $\gamma_E \approx 0.577$ is the Euler–Mascheroni constant. The appearance of $\gamma_E$ is a clear signal of the regularisation scheme being employed and it should not appear in our final expression, as it would indicate that the observable is scheme dependent. For the combination of $s_N$-functions that appears in $S_N(p;q)$, the term with $1-\gamma_E$ drops out as expected for a scheme independent observable.

We are mostly interested in the large $N$ limit of the partition function, which can be worked out by using the Euler–Maclaurin approximation for the sum in (148). This yields

$$s_N(x) = -\frac{\pi^2(6x^2 - 6x + 1)}{72N^2} + \mathcal{O}(N^{-3}). \tag{149}$$

Carrying out this expansion to higher powers in $N$ seems to yield regularisation scheme dependent terms multiplying odd powers of $1/N$, but it also always comes with $(2x-1)$, which vanishes in the final expression when all modes are combined. Therefore, it seems that the expansion of $S_N(p;q)$ at large $N$ only contains even powers of $1/N$. There is a slight problem with this expansion however, since at order $1/N^4$ the integrals diverge and it is not clear how to handle them. It should be noted that (148) is the exact expression for $s_N(x)$ and so in principle finite $N$ answers can be worked out. Indeed, we will present a numerical evaluation of the final integral (144) using the result (148) below. In this case all integrals are finite and no issue is encountered. We therefore do not attempt to work out sub-leading corrections in the large $N$ approximation below, but instead find a fit to the numerical finite $N$ computation from which we can deduce the $1/N$ corrections.

Focusing on the leading large $N$ expansion, we can now assemble the partition function. First we have

$$S_N(p;q) = -\mathrm{Re}\ \psi(\tfrac{1+|q|-ip}{2}) + \log\frac{N}{2\pi} + \mathcal{O}(N^{-2}). \tag{150}$$

The first term in (150) involves the polygamma function and the resulting integral (144) can only be performed numerically. The integral can however be evaluated to high accuracy which shows that

$$\int_0^\infty \frac{\mathrm{d}p}{\pi} \log\sinh(\pi p)\,\mathrm{Re}\left[-3\psi(\tfrac{1+ip}{2}) - \psi(\tfrac{3+ip}{2}) + 4\psi(\tfrac{2+ip}{2})\right] = \log\pi. \tag{151}$$

The remaining integrals can be computed analytically, and combining all factors we find

$$\Gamma_{\mathbb{K}} \approx \log(2\pi) + \mathcal{O}(N^{-2}), \tag{152}$$

which corresponds to the following partition function

$$Z_{\mathrm{M2}} \approx \frac{1}{2\pi}\,e^{\frac{4\pi^4}{|\lambda|}}\left(1 + \mathcal{O}(N^{-2})\right). \tag{153}$$

It is clear that this result does not agree with the QFT prediction in (105). In fact the $N$-dependence does not even match. Most likely this mismatch is a result of the fact that we have not taken careful account of the orbifold on the M2-brane world-volume. Indeed the M2-brane world-volume is $\mathrm{AdS}_2/\mathbf{Z}_N \times S^1$, and we have not taken any special care of this orbifolding. This is very similar to earlier studies of the string partition function of a string that is multi-wound around $\mathrm{AdS}_2$. This setup has been analysed extensively, notably for the case of strings in $\mathrm{AdS}_5$ dual to $\mathcal{N} = 4$ SYM [43–45]. These analyses have not yielded a precise match with the QFT prediction and a prescription for dealing with the multi-wound string remains an open problem (see however [17]). We expect that resolving the problem for multi-wound string in AdS will also lead to a solution to the mismatch we encounter here. It is interesting that the string partition function does give the right $N$-scaling even though the M2-brane partition function does not. The two partition functions are expected to match exactly in the type IIA limit which

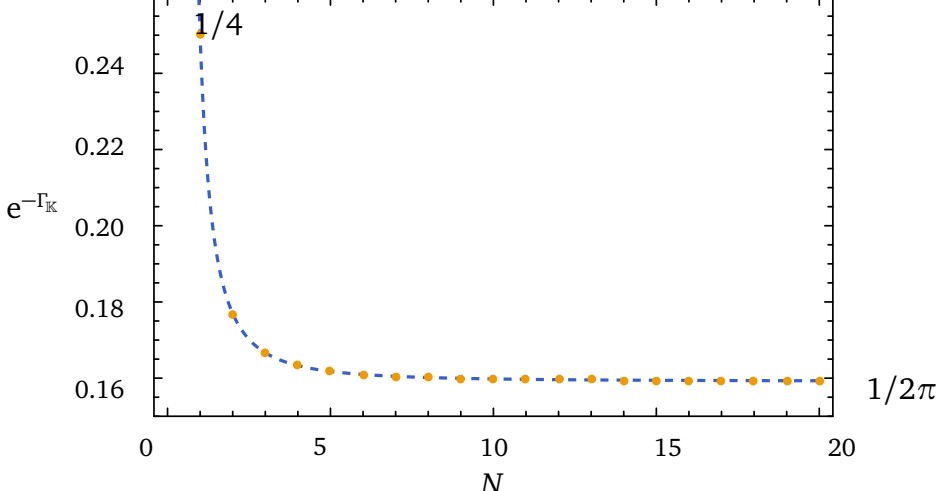

Figure 3: The one-loop M2-brane partition function as a function of $N$. For $N = 1$ we recover the answer obtained for $k = 1$ in [41]. As $N$ is increased the partition function quickly approaches the large $N$ answer of $1/2\pi$. The blue dashed line shows the fit (154) to the numerical data.

in the present case is for large $N$. This further indicates that a more complete understanding on the relation between M2-brane partition functions and string partition functions is required, especially on (mildly) singular backgrounds such as we have here.

Our approach allows for an evaluation of the partition function for general $N$, i.e. without taking the large $N$ limit. This is done by plugging the exact expression (148) into (144) and evaluating the integral numerically. Doing so yields the plot in Figure 3. The figure has two notable features, first for $N = 1$ we recover the M2-brane partition function on non-orbifolded $\mathrm{AdS}_4 \times S^7$ which was first computed in [41]. In that paper the M2-brane partition function was computed for $\mathrm{AdS}_4 \times S^7/\mathbf{Z}_k$ and matched with the vev of the $\frac{1}{2}$-BPS Wilson loop in ABJM theory. The $k = 1$ limit had to be treated separately where it was shown that the M2-brane one-loop partition function is $1/4$. The figure also shows that the large $N$ limit is reached for relatively low $N$. Already for $N = 10$ the final answer is very close to $1/2\pi$, which we found in our large $N$ analysis. This can be made more precise by finding a best fit to the numerical data. We find that the function

$$\Gamma_{\text{fit}}(N) = \log(2\pi) - \frac{1}{c_1 N^2 + c_2 N + c_3}, \tag{154}$$

with $\{c_1, c_2, c_3\} \approx \{2.42608, 0.03030, -0.24195\}$ is within $10^{-5}$ of the numerical data for $\Gamma_{\mathbb{K}}$ throughout the range and its negative exponential is plotted in Figure 3.

It is interesting that this setup allows us in principle to compute the Wilson loop vev for finite $N$ using holography. Unfortunately the mismatch at large $N$, due to the conical singularity, is preventing us from fully utilizing this fact. We expect however that resolving this mismatch will still allow for an evaluation of the WL vev at finite $N$ in a similar manner as how the computation of [41] allowed for a match with QFT for the ABJM WL at a finite Chern-Simons level $k$.

# 6  MSYM on $S^3$ and spherical D2-branes

After the extensive discussion of the holographic Wilson loop in seven-dimensional MSYM, we continue in this and the next section with a discussion of the analogous story in three- and two-dimensional MSYM respectively. These cases are similarly complicated due to the running behaviour of the dilaton. In this section we start by reviewing the holographic dual to MSYM on $S^3$, i.e. spherical D2-branes [2]. We review the string configuration dual to the $\frac{1}{2}$-BPS Wilson loop in the field theory, which was studied at leading order in [6]. Our main task in the remainder of this section then consists of computing the quadratic action of the fluctuations around the classical string configuration and evaluating the corresponding partition function. Before doing so, let us quickly review the prediction for the observable computed using supersymmetric localisation of the QFT.

The localised matrix model is somewhat subtle due to the fact that the MSYM action in three dimensions is $Q$-exact. However, a careful analysis that was performed in [6], demonstrates that even though the free energy vanishes as expected for a $Q$-exact action, the Wilson loop vacuum expectation value is non-trivial and reads

$$\langle W \rangle = \frac{3N}{6\pi^2\lambda}\left((6\pi^2\lambda)^{1/3}\cosh\left(6\pi^2\lambda\right)^{1/3} - \sinh\left(6\pi^2\lambda\right)^{1/3}\right). \tag{155}$$

Moreover, note that this result is exact in $\lambda$. The strong coupling expansion $\lambda \gg 1$ then takes the form

$$\langle W \rangle = \frac{3N}{2(6\pi^2\lambda)^{2/3}}\, e^{(6\pi^2\lambda)^{1/3}}\left(1 + \mathcal{O}(\lambda^{-1})\right), \tag{156}$$

which is the target we hope to reproduce using holography.

The gravity dual to MSYM on a three-sphere is realised by a stack of $N$ D2-branes wrapping a spherical world-volume. The ten-dimensional metric can be written as [2][27]

$$ds_{10}^2 = \ell_s^2(6\pi^2\lambda)^{1/3}h^{1/2}\left(\frac{du^2 + d\theta^2}{\cosh u} + \frac{\cos^2\theta}{f}d\Omega_2{}^2 + \frac{\sin^2\theta}{h}d\widetilde{\Omega}_3^2 + \sinh^2 u\, d\Omega_3^2\right), \tag{157}$$

where $u \in (0,\infty)$ and $d\Omega_n^2$, $d\widetilde{\Omega}_n^2$ both denote the metric on a round $n$-sphere with volume form $\mathrm{vol}_n$ and $\widetilde{\mathrm{vol}}_n$ respectively. In addition, we defined the functions $f$ and $h$ as

$$f = \frac{\sin^2\theta + \sinh^2 u}{\cosh u}, \qquad h = \cosh u + \frac{\sin^2\theta}{1 + \cosh u}. \tag{158}$$

The dilaton in this case is given by

$$e^{2\Phi} = \frac{\lambda^{5/3}}{(6\pi^2)^{1/3}N^2}\frac{\sqrt{h}}{f\cosh^2 u}, \tag{159}$$

while the (external) RR- and NS-potentials supporting this background are given by

$$B_2 = -i\ell_s^2(6\pi^2\lambda)^{1/3}\frac{\cos^3\theta}{f\cosh u}\mathrm{vol}_2, \tag{160}$$

$$C_3 = i\ell_s^3 N\frac{(6\pi^2)^{2/3}}{\lambda^{1/3}}\frac{\sin^4\theta}{h}\widetilde{\mathrm{vol}}_3, \tag{161}$$

$$C_5 = i\frac{N\pi^2\ell_s^6}{2}\left(9\cos\theta - \cos 3\theta - \frac{12\cosh u\cos\theta\sin^4\theta}{f}\right)\mathrm{vol}_3 \wedge \mathrm{vol}_2. \tag{162}$$

---

[27]Note that we have changed coordinates compared to [2].

From these expressions we can obtain the field strengths $F_4$ and $H_3$ as

$$F_4 = \mathrm{d}C_3 - \star_{10}(\mathrm{d}C_5 - H_3 \wedge C_3), \qquad H_3 = \mathrm{d}B_2. \tag{163}$$

Expanding this solution in the UV ($u \to \infty$) we recover the flat space D2-brane background, while in the IR ($u \to 0$) the metric smoothly caps off. It is important to note though that the dilaton is singular in the IR when $\theta = 0$. This singularity will play an important role in the one-loop string computation below.

## 6.1 Holographic Wilson loop

The $\frac{1}{2}$-BPS Wilson loop lies along the equator of the three-sphere. Holographically this translates to a string wrapping the equator of the three-sphere with metric $\mathrm{d}\Omega_3^2$ and extending into the bulk along the $u$-coordinate. The focus here is on the next-to-leading order string partition function in the large $\lambda$ expansion, but let us start by briefly recalling the leading order contribution.

As in the previous section, we work in static gauge where we can identify the world-sheet coordinates with the space-time coordinates $(u, \tau)$, where $\tau$ parametrises the coordinate along the equator of the three-sphere. The classical solution is then obtained by minimising the string action and can be described by the string lying at $\theta = 0$, along the equator of the sphere and at a generic point on $S^2$. The world-sheet metric is obtained as the pull-back of (157) and reads

$$\mathrm{d}s_2^2 = \ell_s^2 (6\pi^2\lambda)^{1/3} \cosh^{1/2}u \left( \frac{\mathrm{d}u^2}{\cosh u} + \sinh^2 u \, \mathrm{d}\tau^2 \right) = \mathrm{e}^{2\rho} \left( \mathrm{d}\sigma^2 + \mathrm{d}\tau^2 \right), \tag{164}$$

where in the second step we change the coordinates to conformal coordinates, defined through

$$\mathrm{d}\sigma = -\frac{\mathrm{d}u}{\sinh u \cosh^{1/2} u}, \qquad \mathrm{e}^{2\rho} = \ell_s^2 (6\pi^2\lambda)^{1/3} \sinh^2 u \cosh^{1/2} u. \tag{165}$$

Although it is straight-forward to express $\sigma$ analytically in terms of $u$, it is not possible to find an analytic inverse and so we will often present our results below using the original $u$ coordinates. Asymptotically, the IR and UV behaviour of the coordinates is given by $u \sim \mathrm{e}^{-\sigma}$ for $\sigma \gg 1$ and $u \sim -\frac{2}{3}\log\sigma$ for $\sigma \ll 1$ respectively. We also give the pull-back of the dilaton which reads

$$\mathrm{e}^{2\Phi_0} = \frac{\lambda^{5/3}}{(6\pi^2)^{1/3}N^2} \frac{1}{\sinh^2 u \cosh^{1/2} u}, \tag{166}$$

and we note that $\partial_\sigma \Phi_0 = -\partial_\sigma \rho$.

The classical string action can be obtained from (23) by pulling back the metric and $B$-field to the world-volume of the string. The pull-back of the $B$-field vanishes, and after regularising the area by means of e.g. the Legendre transform,[28] we obtain the following classical on-shell action

$$S_{\mathrm{cl}} = -(6\pi^2\lambda)^{1/3}, \tag{167}$$

resulting in a match at leading order with the field theory result (156).

Following the recipe outlined in Section 3, we can proceed to compute the first quantum correction. In line with the discussion above, expanding the fluctuation Lagrangian to quadratic order we find that the fluctuations are described by eight scalar and eight fermionic modes living on the string world-sheet. The dynamics of these modes is fully determined by their mass and charge with respect to the background gauge fields living on the string.

---

[28]Note that [6] uses a different regularisation scheme.

The background field, obtained from the pull-back of the ten-dimensional $B$-field (160), is given by

$$A = A_\tau \mathrm{d}\tau = \frac{1}{2}\mathrm{d}\tau \,. \tag{168}$$

As outlined in Section 3.3.1, six of the scalars are uncharged, while the remaining two have opposite charge $q = \pm 2$. Of the uncharged bosons, two have mass $E_x$ and four of them have mass $E_y$, while the charged bosons have mass $E_{z,\pm} = E_y$. Hence, we find the following bosonic fluctuation operators,

$$
\begin{aligned}
\tilde{\mathcal{K}}_x &= -\partial_\sigma^2 - \partial_\tau^2 + E_x \,, \\
\tilde{\mathcal{K}}_y &= -\partial_\sigma^2 - \partial_\tau^2 + E_y \,, \\
\tilde{\mathcal{K}}_{z,q} &= -\partial_\sigma^2 - (\partial_\tau - iqA_\tau)^2 + E_y \,,
\end{aligned} \tag{169}
$$

where $E_x$ and $E_y$ are given by (40) and the tilde refers to the usual Weyl rescaled operators. Note that the difference between $\tilde{\mathcal{K}}_y$ and $\tilde{\mathcal{K}}_{z,q}$ is given solely by the coupling of the latter operator to the gauge field $A$. However since the charge is $\pm 2$, after expanding in integer modes along the $\tau$ direction, the difference can be absorbed in the sum over frequencies and so we may replace the operator $\tilde{\mathcal{K}}_{z,\pm 2}$ with $\tilde{\mathcal{K}}_y$.

The fermions on the other hand come in two kinds, distinguished by their charge $q = \pm 1$, where half of them are positively charged and the other half negatively charged. The fermionic fluctuation operators can be written as

$$\tilde{\mathcal{D}}_q = i\left(\slashed{\partial} - iq\slashed{A}\right) + a\sigma_3 + v \,, \tag{170}$$

where we defined the functions

$$a = -\partial_\sigma \rho = \frac{3 + 5\cosh 2u}{8\sqrt{\cosh u}} \,, \qquad v = 1 \,. \tag{171}$$

Notice that in the language of Section 3.3.1, the rotation angle is trivial $\xi = 0$ and the fermionic operators are simply related to the bosonic operators. To be more specific, we write

$$\tilde{\mathcal{D}}_q = \sigma_+ \mathcal{L} + \sigma_- \mathcal{L}^\dagger - i\sigma_2 D_\tau + 1 \,. \tag{172}$$

A conjugate operator can be defined using charge conjugation, 'time'-reversal and the second Pauli matrix

$$\tilde{\mathcal{D}}_q^\dagger = -\sigma_2 \mathbf{T}\tilde{\mathcal{D}}_{-q}\sigma_2 = \sigma_+ \mathcal{L} + \sigma_- \mathcal{L}^\dagger - i\sigma_2 D_\tau - 1 \,. \tag{173}$$

Now it is easy to verify that

$$\tilde{\mathcal{D}}_q^\dagger \tilde{\mathcal{D}}_q = \sigma_+ \sigma_- \tilde{\mathcal{K}}_{x,q} + \sigma_- \sigma_+ \tilde{\mathcal{K}}_{y,q} \,, \tag{174}$$

where we have introduced the operators

$$
\begin{aligned}
\tilde{\mathcal{K}}_{x,q} &= -\partial_\sigma^2 - (\partial_\tau - iqA_\tau)^2 + E_x \,, \\
\tilde{\mathcal{K}}_{y,q} &= -\partial_\sigma^2 - (\partial_\tau - iqA_\tau)^2 + E_y \,.
\end{aligned} \tag{175}
$$

Due to the half-integer valued gauge field, and the integer charge of the fermions, effectively the fermions can be treated as bosons which means that using (174) we have

$$\det\left(\tilde{\mathcal{D}}_{\pm 1}^\dagger \tilde{\mathcal{D}}_{\pm 1}\right) = \left(\det\tilde{\mathcal{K}}_x\right)\left(\det\tilde{\mathcal{K}}_y\right) \,, \tag{176}$$

where on the left-hand side we are dealing with a fermionic determinant but on the right-hand side we have an honest bosonic determinant. Putting things together we find that the effective action $\Gamma_{\mathbb{K}}$ can be written as

$$\Gamma_{\mathbb{K}} = \log \frac{\left(\det\tilde{\mathcal{K}}_x\right)\left(\det\tilde{\mathcal{K}}_y\right)^3}{\left(\det\tilde{\mathcal{K}}_x\right)^2\left(\det\tilde{\mathcal{K}}_y\right)^2} = \log \frac{\det\tilde{\mathcal{K}}_y}{\det\tilde{\mathcal{K}}_x} \,. \tag{177}$$

The above expression diverges as discussed in Section 3.4 (see Table 2). We also have to combine this with the contribution from the dilaton, which just like for seven-dimensional MSYM is divergent in this case. We refer to the regularisation procedure suggested in Section 3.6 to obtain a finite answer. To this end we must evaluate (71), which requires

$$\lim_{\sigma \to \infty} \left( \rho - \Phi_0 - \sigma \partial_\sigma (\rho - \Phi_0) + \log \frac{\pi}{\ell_s} \right) = \log \frac{6 N \pi^3}{(6\pi^2 \lambda)^{2/3}} \,, \qquad (178)$$

where we have used that $u \sim e^{-\sigma}$ asymptotically.

To obtain the regularised value of the quantum effective action $\Gamma_{\mathbb{K}}$, we compute the difference of phase shifts for the two bosonic operators. It turns out that for any two operators that can be written in terms of first order operators $\mathcal{L}$ and $\mathcal{L}^\dagger$, as is the case for $\tilde{\mathcal{K}}_x$ and $\tilde{\mathcal{K}}_y$, the phase shifts are simply related as

$$\delta_y = \delta_x + \arctan p \,, \qquad (179)$$

which is corroborated by the WKB approximation in Appendix D. Inserting this expression into (62) we find that $(\Gamma_{\mathbb{K}})_{\text{reg}} = \log(2\pi)$, which results in the 1-loop partition function

$$Z_{\text{1-loop}} = \frac{3N\pi^2}{(6\pi^2 \lambda)^{2/3}} \,. \qquad (180)$$

The scaling of the partition function with $N$ and $\lambda$ is consistent with the QFT result (156). However, the numerical factor is off by $2\pi^2$. This could be due to the divergence in the dilaton, which we regulated as we explained previously but did not assign any additional factors to. At this stage we leave the numerical mismatch to further future study.

Finally, note that similar to the D6-brane above, we can uplift the type IIA supergravity solution to eleven dimensions. However, as discussed in [2], the uplifted geometry has a singularity in the metric at the location of the probe string. This singularity is worse than the conical singularity on the M2-brane world-volume encountered in the previous section hence in this case we will refrain from analysing the one-loop determinant in the M2-brane picture.

# 7  MSYM on $S^2$ and spherical D1-branes

In this section we study the $\frac{1}{2}$-BPS Wilson loop in MSYM on $S^2$ using holography. The dual supergravity background, as well as the leading order holographic Wilson loop were previously studied in [6], and here we will initiate the analysis of the next-to-leading order correction. The dual ten-dimensional geometry is given by[29]

$$ds_{10}^2 = 2\ell_s^2 (8\pi^3 \lambda)^{1/4} f_1^{1/2} \left( \frac{du^2}{4u(u+2)} + d\theta^2 + \frac{1}{2} u(1+u) d\Omega_2^2 + \frac{\cos^2 \theta}{f_2} d\tilde{\Omega}_2^2 + \frac{\sin^2 \theta}{f_3} d\Omega_4^2 \right), \quad (181)$$

where we introduced the functions

$$f_1 = \frac{1 + u + 2\sin^2 \theta}{(1+u)^2} \,, \qquad f_2 = 1 - \frac{2(3+u)\cos^2 \theta}{(1+u)^2} \,, \qquad f_3 = 1 + \frac{2\sin^2 \theta}{1+u} \,. \qquad (182)$$

As above, $d\Omega_n^2$ as well as their tilded analogues represent the unit radius round metrics on $n$-spheres with volume forms $\text{vol}_n$. In the above metric, the field theory is located on the two-sphere $d\Omega_2^2$ and $u$ is the holographic coordinate while the rest of the space is the geometric realisation of the R-symmetry of the dual QFT.

---

[29]As in Section 6, we have changed coordinates compared to [2].

In addition, the full supergravity background contains a non-trivial dilaton

$$e^{2\Phi} = \frac{\sqrt{2\pi}\lambda^{3/2}}{N^2(1+u)((1+u)^2 - 2(3+u)\cos^2\theta)}, \tag{183}$$

as well as the following type IIB form potentials

$$B_2 = i(2\pi^3\lambda)^{1/4}\ell_s^2 \frac{8\sqrt{2+u}\cos^3\theta}{(1+u)^2 - 2(3+u)\cos^2\theta}\,\text{vol}_2, \tag{184}$$

$$C_4 = iN\ell_s^4 16\pi\left(\frac{8\pi}{\lambda}\right)^{1/4}\frac{\sqrt{2+u}\sin^5\theta}{(1+u+2\sin^2\theta)}\text{vol}_4, \tag{185}$$

$$C_6 = iN\pi^2\ell_s^6\left(12\,\theta - 8\sin(2\theta) + \sin(4\theta) + \frac{32(1+u)^2\sin^5\theta\cos\theta}{(1+u)^2 - 2\cos^2\theta(3+u)}\right)\text{vol}_2\wedge\text{vol}_4. \tag{186}$$

In terms of these potentials, the field strengths are defined as follows

$$F_7 = dC_6 - H_3\wedge C_4, \qquad F_3 = -\star_{10}F_7, \qquad F_5 = (1 + \star_{10})dC_4. \tag{187}$$

## 7.1 Holographic Wilson loop

The $\frac{1}{2}$-BPS Wilson loop lies along the equator of $S^2$ and is dual to a string that extends into the bulk along the $u$ coordinate introduced above and sits on the equator of $S^2$ throughout. We work in static gauge and identify the space-time coordinates $(u, \tau)$ with the world-sheet coordinates of the string. Here $\tau$ is the coordinate parametrising the equator of $d\Omega_2^2$. The classical solution of the string is localised at the point $\theta = 0$ and an arbitrary fixed point on $\tilde{S}^2$, and the classical world-sheet metric takes the following form

$$ds_2^2 = \frac{\ell_s^2(8\pi^3\lambda)^{1/4}}{\sqrt{u+1}}\left(\frac{du^2}{2u(u+2)} + u(u+1)d\tau^2\right). \tag{188}$$

Similarly as before, it proves useful to define conformal coordinates via

$$du = -u\sqrt{2(1+u)(2+u)}d\sigma, \tag{189}$$

such that the world-sheet metric becomes

$$ds_2^2 = e^{2\rho}\left(d\sigma^2 + d\tau^2\right), \qquad e^{2\rho} = \ell_s^2(8\pi^3\lambda)^{1/4}u(1+u)^{1/2}. \tag{190}$$

Although it is straight-forward to integrate (189) in order to express $\sigma$ as a function of $u$, the inverse can not be expressed analytically, which means that we will use a hybrid form of the two coordinate systems. In these coordinates, the IR and UV are located at $\sigma \to \infty$ (or $u \to 0$) and $\sigma \to 0$ (or $u \to \infty$) respectively.

The leading order contribution to the Wilson loop expectation value, corresponding to the string on-shell action, was computed in [6]. The details of the computation can be found there and will not be reproduced here. We have verified that the answer given in [6] can be obtained by the regularisation procedure which uses an appropriate Legendre transform [25]. The regularised classical action is given by

$$S_{\text{cl}} = -2\left(8\pi^3\lambda\right)^{1/4}, \tag{191}$$

and hence we reproduce the leading order contribution to the Wilson loop vev

$$\langle W\rangle^{\text{LO}} = e^{2\left(8\pi^3\lambda\right)^{1/4}}, \tag{192}$$

in line with the matrix model expectation [6]. It should be noted here that the matrix model has only been solved to leading order in the strong coupling expansion, and so the expectation value of the $\frac{1}{2}$-BPS Wilson loop is only known to that order.

The goal here is to go beyond the leading order using string theory and give a prediction for the Wilson loop expectation value to next-to-leading order. We proceed in a similar manner as above, following the general discussion of Section 3 we expand the string action to quadratic order in the fluctuating field. The resulting theory is described by eight bosonic and eight fermionic modes living on the string world-sheet, and are characterised by their masses and charge. In the case at hand, the $B$-field induces a non-trivial background connection

$$A = A_\tau \mathrm{d}\tau = \frac{(5+3u)u}{2(u^2-5)}\mathrm{d}\tau\,. \tag{193}$$

When expanding the string action we find, in line with the discussion in Section 3.3.1, that two of the scalar operators are oppositely charged with charge $q = \pm 2$ and have mass $E_z$ (which we present below), while the remaining six operators are uncharged. These six scalars are divided into five with mass $E_y$ and a remaining one with mass $E_x$. We thus find the following bosonic operators[30]

$$\begin{aligned}
\tilde{\mathcal{K}}_x &= -\partial_\sigma^2 - \partial_\tau^2 + E_x\,, \\
\tilde{\mathcal{K}}_y &= -\partial_\sigma^2 - \partial_\tau^2 + E_y\,, \\
\tilde{\mathcal{K}}_{z,q} &= -\partial_\sigma^2 - (\partial_\tau - iqA_\tau)^2 + E_z\,,
\end{aligned} \tag{195}$$

where the masses are explicitly

$$\begin{aligned}
E_x &= \frac{u(40+60u+21u^2)}{8(u+1)}\,, \\
E_y &= \frac{-3u^3}{8(u+1)}\,, \\
E_z &= E_x + \frac{5u(15+18u+9u^2+2u^3)}{(u^2-5)^2}\,.
\end{aligned} \tag{196}$$

When expanding the fermionic action we find that the fermions split into four positively charged ones and four negatively ones, with kinetic operators

$$\tilde{\mathcal{D}}_q = i\left(\slashed{\partial} - qi\slashed{A}\right) + a\sigma_3 + v\,. \tag{197}$$

Here the fermionic potentials are

$$a = -\frac{5iq}{2}\sqrt{\frac{u(u+2)}{u^2-5}}\,, \qquad v = -\frac{iu^{3/2}(5+3u)}{2\sqrt{2(u+1)(u^2-5)}}\,, \tag{198}$$

and $q = \pm 1$. Putting all these contributions together, we obtain the following quantum effective action

$$\Gamma_{\mathbb{K}} = \log \frac{\left(\det \tilde{\mathcal{K}}_x\right)^{1/2}\left(\det \tilde{\mathcal{K}}_y\right)^{5/2}\left(\det \tilde{\mathcal{K}}_{z,+2}\right)^{1/2}\left(\det \tilde{\mathcal{K}}_{z,-2}\right)^{1/2}}{\left(\det \tilde{\mathcal{D}}_{+1}\right)^2\left(\det \tilde{\mathcal{D}}_{-1}\right)^2}\,. \tag{199}$$

---

[30]Note that naively we encounter two coupled operators for the bosonic fluctuations along the transverse two-sphere $\mathrm{d}\tilde{\Omega}_2^2$. However, these are diagonalised with the following similarity transformation

$$U^\dagger \tilde{\mathcal{K}}_z U = \begin{pmatrix} \tilde{\mathcal{K}}_{z,+} & 0 \\ 0 & \tilde{\mathcal{K}}_{z,-} \end{pmatrix}\,, \qquad U = \frac{1}{\sqrt{2}}\begin{pmatrix} 1 & i \\ i & 1 \end{pmatrix}\,. \tag{194}$$

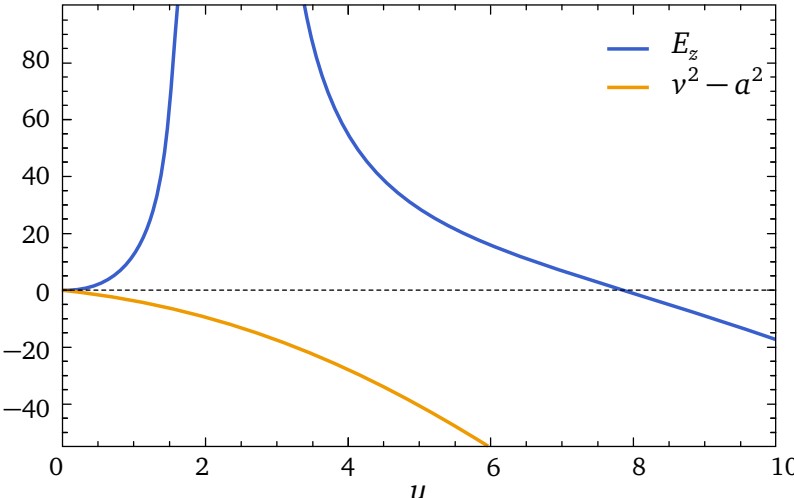

Figure 4: The behaviour of the masses $E_z$ and $v^2 - a^2$ of the charged fields in terms of the $u$-coordinate. Notice that the total fermionic mass $v^2 - a^2$ is not divergent for $u = \sqrt{5}$, while each term in this expression separately does diverge.

We have verified that the operators satisfy the mass-rule (50), where we use the pull-back of the dilaton

$$e^{2\Phi_0} = \frac{\sqrt{2\pi}\lambda^{3/2}}{N^2(1+u)(u^2-5)}.$$  (200)

We notice that the mass of the charged bosons $E_z$ in (196) diverges in the limit where $u \to \sqrt{5}$. The potentials appearing in the fermionic operators also diverge but the physical mass of the fermions, given by $v^2 - a^2$, is regular (see Figure 4). This divergence is directly related to the fact that the dilaton (200) diverges at this point. Furthermore, the problematic behaviour of the masses prevents us from proceeding with the computation of the quantum effective action via the phase shift method without modification. Indeed, we should expect bound states in the quantum mechanical problem for $\tilde{\mathcal{K}}_z$, which have to be appropriately taken into account. We will therefore leave the computation of the regularised quantum effective action for the future. However, we can still characterise the structure of divergences by employing a WKB approximation for the phase shifts of the bosonic operators in the large $p$ limit. This is carried out in Appendix D and here we quote the main result. That is, the difference between the bosonic and fermionic phase shifts has the asymptotic behaviour

$$\delta_{\text{bos,D1}} - \delta_{\text{ferm,D1}} = -\frac{1}{p} + \mathcal{O}(p^{-3}),$$  (201)

up to a constant term which is not captured by the WKB method. This means that we should expect the same logarithmic divergence in the partition function, as we encounter for the string in AdS space. This could have already been anticipated from the results in Table 2. The situation here is therefore improved when compared to the $d = 7$ and $d = 3$ cases discussed in the previous two sections, since the dilaton is regular in the IR of the geometry, and is therefore regular at the centre of the world-sheet. Recall that the divergence of the dilaton and the associated divergences in the one-loop quantum effective action was the source of all our troubles in the previous two cases, as discussed in detail in Section 3.

Even without computing the regularised quantum effective action, we can provide a prediction for the scaling of the one-loop partition function. This will provide a prediction for the scaling of the Wilson loop expectation value in the dual theory to next-to-leading order. An answer that can hopefully be checked by analysing the D1 matrix model in more detail.

To this end we use the regularisation procedure outlined in Section 3.6

$$\log Z_{\text{1-loop}} = -(\Gamma_{\mathbb{K}})_{\text{reg}} + \lim_{\sigma \to \infty} \left( \rho - \Phi_0 - \sigma(\partial_\sigma \rho) + \log \frac{\pi}{\ell_s} \right), \tag{202}$$

where we have dropped the $\partial_\sigma \Phi_0$ term as it vanishes for a dilaton regular in the IR. Evaluating this and combining it with the leading order term at one-loop level we find

$$Z_{\text{string}} = -i \frac{\sqrt{5}\, 4\pi^3 N}{(8\pi^3 \lambda)^{5/8}} \, e^{-(\Gamma_{\mathbb{K}})_{\text{reg}}} \, e^{2(8\pi^3 \lambda)^{1/4}}. \tag{203}$$

Note that naively this suggests an imaginary vacuum expectation value for the Wilson loop, originating from the imaginary IR value of the dilaton. Obviously this does not make much sense and we expect the imaginary unit to be compensated by the regularised quantum effective action. Indeed, this expectation is corroborated by the WKB analysis in Appendix D, where the large $p$ expansion of the phase shift results in an imaginary expression, in contrast to the familiar real expression for the other cases.

## 8 Summary and conclusion

In this paper we conduct a study of sub-leading, strong coupling corrections to the 1/2 BPS Wilson loop vev in MSYM on a $d$-sphere. More precisely, we compute the one-loop corrections to the partition function of a probe string and M2-brane in the relevant holographic background. In all cases where independent QFT results exist, we find an agreement with the scaling of the sub-leading contributions both in $N$ and $\lambda$. Matching the numerical coefficient of the sub-leading terms proved more challenging and we only partially succeed, leaving some puzzles for future work.

In more detail, we determine the one-loop contributions to the $\frac{1}{2}$-BPS Wilson loop both holographically and by direct analysis of the matrix model. We construct the relevant string and M2-brane fluctuation operators and evaluate their one-loop determinants using the phase shift method adapting the available prescription to our non-conformal setup. In addition, for the seven-dimensional case, we verify our results using the Gel'fand-Yaglom method in Appendix C. Our holographic results are summarised as follows:

$$
\begin{aligned}
\text{D6:} \quad & Z_{\text{string}} = \frac{N}{2\pi^2} \, e^{\frac{4\pi^4}{|\lambda|}}, \\
\text{D5:} \quad & Z_{\text{string}} \sim N \, e^{\frac{4\pi^3}{3\lambda}} \, e^{4\pi \, e^{-\frac{8\pi^3}{3\lambda}}}, \\
\text{D2:} \quad & Z_{\text{string}} = \frac{3N\pi^2}{(6\pi^2 \lambda)^{2/3}} \, e^{(6\pi^2 \lambda)^{1/3}}, \\
\text{D1:} \quad & Z_{\text{string}} = -i \frac{4\sqrt{5}\pi^3 N}{(8\pi^3 \lambda)^{5/8}} \, e^{-(\Gamma_{\mathbb{K}})_{\text{reg}}} \, e^{2(8\pi^3 \lambda)^{1/4}}.
\end{aligned} \tag{204}
$$

In addition, we uplift the D6-brane solution to eleven-dimensional supergravity and analyse the M2-brane one-loop partition function in the resulting $\text{AdS}_4/\mathbf{Z}_N \times S^7$ background. When $N = 1$ this is precisely the dual to (abelian) ABJM and we correctly reproduce the one-loop results of [41]. In the large $N$ limit we find that the M2-brane computation to one-loop order results in

$$Z_{\text{M2}} = \frac{1}{2\pi} \, e^{\frac{4\pi^4}{|\lambda|}}. \tag{205}$$

This result should match with the string theory result obtained above, but unfortunately we find a $1/\pi$ discrepancy in the numerical prefactor with respect to the probe string computation. More importantly, in the M-theory calculation the scaling with $N$ is incorrect, for which we suggest a possible explanation below.

In [7, 18] the partition function of MSYM on $S^d$ was localised to a matrix integral. At leading order in the strong coupling and large $N$ expansion, the matrix model can be solved for any $d$ yielding a prediction for the $\frac{1}{2}$-BPS Wilson loop vev. At leading order this prediction matches the classical area of the dual string world-sheet and, surprisingly, also matches the sub-leading scaling of the string partition function. In the case $d = 3, 4, 5$ the matrix model can be solved exactly in the coupling constant and the scaling agrees with the leading order strong coupling prediction. For $d = 7$, we perform a numerical analysis of the full matrix model and obtain an agreement with the scaling and a prediction for the numerical constant. When $d = 2, 6$, the matrix model is more subtle and results beyond leading order have not yet been found. Summarising, the one-loop predictions from field theory are given by

$$
\begin{aligned}
\text{MSYM in } d = 7: \quad &\langle W \rangle_{1-\text{loop}} = 12N \, e^{\frac{4\pi^4}{|\lambda|}} \,, \\[4pt]
\text{MSYM in } d = 6: \quad &\langle W \rangle_{1-\text{loop}} \sim N \, e^{\frac{4\pi^3}{3\lambda}} \, e^{4\pi \, e^{-\frac{8\pi^3}{3\lambda}}} \,, \\[4pt]
\text{MSYM in } d = 3: \quad &\langle W \rangle_{1-\text{loop}} = \frac{3N}{2(6\pi^2\lambda)^{2/3}} \, e^{(6\pi^2\lambda)^{1/3}} \,, \\[4pt]
\text{MSYM in } d = 2: \quad &\langle W \rangle_{1-\text{loop}} \sim \frac{N}{(8\pi^3\lambda)^{5/8}} \, e^{2(8\pi^3\lambda)^{1/4}} \,.
\end{aligned}
\tag{206}
$$

Finally, we also compute the perturbatively exact planar free energy of MSYM on $S^7$,

$$
F = N^2 \left( -\frac{16\pi^{10}}{3|\lambda|^3} - \frac{2\pi^4}{|\lambda|} + f \right),
\tag{207}
$$

where $f \simeq 0.14$ is a constant fixed through numerical analysis, while (at leading order in $N$) the first two terms are obtained analytically.

We have therefore provided a match for the scaling of the Wilson loop vev across a wide range of dimensions, as well as provided new targets for matrix model computations that could shine a new light on the physics of little strings and MSYM on $S^2$. The numerical coefficient of the Wilson loop vev on $S^7$ or $S^3$ does not match our holographic computation, leaving us somewhat puzzled. At present we do not have an adequate explanation for this discrepancy, but a possibility could be that the measure factor ambiguity plaguing the GS string partition function is sensitive to the divergences in the supergravity background we encounter. More pressing in some sense is the absence of $N$ in the M2-brane one-loop result considered in Section 5. The probe M2-brane wraps an orbifolded $\text{AdS}_2 / \mathbf{Z}_N \times S^1$ where subtle effects due to the orbifold singularity may have been missed by our analysis.[31]

Our work opens a window towards obtaining a more fine-grained understanding of holography in a non-conformal setting. In particular we provide novel tools to analyse such setups, which are bound to be useful in more general, less supersymmetric cases. The absence of conformal symmetry introduces a variety of challenges which were partly addressed in this work. However, opening this non-conformal Pandora's box inevitably laid bare more questions and puzzles that could not all be addressed here. Below we list a few directions for future research.

- First of all, we are currently lacking a proper understanding of the matrix model corresponding to the spherical D1-brane background. Developing a better understanding of this case would allow us to match the scaling, as well as predict the sub-leading contributions to the free energy. In particular, the relevant matrix model is an analytic continuation of the matrix model considered in [46] to negative coupling constant. However, before reaching the relevant strong coupling regime this matrix model goes through a

---

[31]In [17] a proposal for taking into account the contribution coming from an orbifold singularity in the string partition function was put forward. It is not obvious to us that the same proposal is valid here in the case of an M2-brane partition function.

phase transition.[32] It would be very interesting to get better control over this model and understand the implications for MSYM on $S^2$. In addition it would be interesting to complete the computation of the one-loop string partition function initiated in Section 7. We hope to come back to this in future work.

- Our results show a uniform scaling behaviour of the Wilson loop vev across dimensions of the form

$$Z_{\text{1-loop}} \sim \begin{cases} N\lambda^{\frac{d-7}{2(6-d)}}, & d \neq 6, \\ N\,e^{\frac{4\pi^3}{3\lambda}}, & d = 6. \end{cases} \tag{208}$$

This uniform scaling behaviour begs the question whether there is a underlying explanation along the lines of the scaling similarity of [47] extended to our one-loop results.

- Evaluating the one-loop determinants we find that the one-loop contribution can be decomposed into a term coming from the smooth part of the string world-sheet with additional contributions from the special loci of the world-sheet, see for example equation (3). This decomposition strongly suggests that there might be an underlying fixed point argument at play. It is appealing to speculate that such localisation properties hold more general and it would be very interesting to derive this from first principles.

- The techniques introduced in this work open a window to accessing next-to-leading order results in a variety of other theories. In particular, in $d \leq 5$ MSYM allows for a universal deformation by giving a mass to the adjoint scalar in the maximally supersymmetric vector multiplet.

- In addition to computing Wilson loop vevs, the techniques used in this work can be used to compute $(p,q)$-string instanton contributions to the partition function. In [17] such contributions were studied for ABJM theory providing valuable holographic information, which in general is challenging to obtain using field theory methods. Doing so in four-dimensional $\mathcal{N} = 2^*$ or five-dimensional $\mathcal{N} = 1^*$ allows us to access holographic data probing integrated correlators in the related MSYM theories.

# Acknowledgments

We are grateful to Francisco Correa, Aldo Cotrone, Luca Griguolo, Charlotte Kristjansen, Daniel Medina-Rincon, Domenico Seminara, Valentin Reys, Watse Sybesma, Lárus Thorlacius, Evangelos Tsolakidis, Jesse van Muiden, Konstantin Zarembo and especially Joe Minahan and Arkady Tseytlin for useful discussions.

**Funding information**  DA, FFG, VGMP and AN are supported by the Icelandic Research Fund under grant 228952-053. FFG and VGMP are partially supported by grants from the University of Iceland Research Fund. PB and VGMP would like to acknowledge the Mainz Institute for Theoretical Physics (MITP) of the Cluster of Excellence PRISMA+ (Project ID 39083149) for its hospitality and partial support during the completion of this work. The contributions of PB were made possible through the support of grant No. 494786 from the Simons Foundation (Simons Collaboration on the Non-perturbative Bootstrap) and the ERC Consolidator Grant No. 864828, titled "Algebraic Foundations of Supersymmetric Quantum Field Theory" (SCFTAlg).

---

[32]We thank Joe Minahan for discussions on this topic.

# A    Functional determinants

In this appendix we collect our convention for the path integral over the fluctuations. The action of the quadratic scalar and fermionic fluctuations in conformal coordinates (28) is defined as

$$S_{\mathbb{K}} = \frac{1}{4\pi\ell_s^2} \int \left( \zeta^a \mathcal{K}_{ab} \zeta^b + \bar{\theta}^a \mathcal{D}_{ab} \theta^b \right) \mathrm{vol}_2 \,, \tag{A.1}$$

where $\zeta_a$ and $\theta_a$ (with $a = 1, \ldots, 8$) denote the eight bosonic and eight fermionic modes respectively of the string, cf. Section 3.3.1. In order to evaluate the one-loop string partition function (21), we need to compute the functional determinants of the second order operators $\mathbb{K}$, that is

$$\mathrm{Sdet}\,\mathbb{K} = \frac{\displaystyle\prod_{\text{bosons}} \det \mathcal{K}}{\displaystyle\prod_{\text{fermions}} \det \mathcal{D}} \,. \tag{A.2}$$

The path integral measure satisfies

$$\int [D\zeta] \mathrm{e}^{-\frac{1}{4\pi\ell_s^2}\|\zeta\|^2} = 1 = \int \left[ D\theta D\bar{\theta} \right] \mathrm{e}^{-\frac{1}{4\pi\ell_s^2}\|\theta\|^2} \,, \tag{A.3}$$

where the norms are defined as follows

$$\|\zeta\|^2 = \int \zeta^a \zeta^b \delta_{ab}\, \mathrm{vol}_2 \,, \qquad \|\theta\|^2 = \int \bar{\theta}^a \theta^b \delta_{ab}\, \mathrm{vol}_2 \,. \tag{A.4}$$

This results in the following Gaussian path integral over the one-loop action:

$$\int \left[ D\zeta D\theta D\bar{\theta} \right] \mathrm{e}^{-S_{\mathbb{K}}} = (\mathrm{Sdet}\,\mathbb{K})^{-1/2} \,. \tag{A.5}$$

# B    Comparing the heat kernel and the phase shift method

In this appendix we contrast the standard heat kernel method, that is often used to compute string partition functions, with the one used in this paper, which amounts to Weyl rescale the world-sheet metric to a flat metric and then use the phase shift method to evaluate the partition function. We will focus on strings in $\mathrm{AdS}_n$ and our goal is to compare the quantum effective action $\Gamma_{\mathbb{K}}$ computed using the two methods.

Consider a string with the world-sheet metric of $\mathrm{AdS}_2$ written in conformally flat coordinates

$$\mathrm{d}s_2^2 = \mathrm{e}^{2\rho}(\mathrm{d}\sigma^2 + \mathrm{d}\tau^2), \qquad \mathrm{e}^{2\rho} = \frac{L^2}{\sinh^2 \sigma} \,, \tag{B.1}$$

where $L$ is the length scale of the string. This setup is encountered for the string in $\mathrm{AdS}_5$, which is dual to a $\frac{1}{2}$-BPS Wilson loop in $\mathcal{N} = 4$ SYM where the parameter is $L^2 = \ell_s^2\sqrt{\lambda}$, and likewise for the $\frac{1}{2}$-BPS Wilson loop in ABJM where the dual $\mathrm{AdS}_4$ string has $L^2 = \pi\ell_s^2\sqrt{2\lambda}$.

On the string we assume there are eight scalar modes, and eight fermionic modes. The kinetic operators for these modes are

$$\begin{aligned} \mathcal{K} &= -\nabla^2 + M_b^2 \,, & &\text{for scalars,} \\ \mathcal{D} &= i\slashed{\nabla} + M_f \sigma_3 \,, & &\text{for fermions.} \end{aligned} \tag{B.2}$$

The consistent spectrum for the numbers $M_b$ and $M_f$ is given by (see e.g. [15] and references therein)

$$M_b^2 L^2 \in \{2_{\times n-2}, 0_{\times 10-n}\} \,, \qquad M_f^2 L^2 \in \{1_{\times 2n-2}, 0_{\times 10-2n}\} \,, \tag{B.3}$$

where the subscript denotes the degeneracy of each mass and the mass rule works out as expected

$$\text{Tr}\left[(-1)^F M^2\right] = (n-2)\frac{2}{L^2} - (2n-2)\frac{1}{L^2} = -\frac{2}{L^2} = R^{(2)}. \tag{B.4}$$

Indeed for $n = 5$, (B.3) gives the spectrum of the $\frac{1}{2}$-BPS string in $\text{AdS}_5$ and for $n = 4$ it corresponds to the spectrum in $\text{AdS}_4$. Now, the goal is to compute the quantum effective action

$$\Gamma_{\mathbb{K}} = \frac{1}{2}\log\text{Sdet}\mathbb{K}, \tag{B.5}$$

using two methods, the heat kernel method and the phase shift method. In both cases we expect a divergent expression and so a suitable regularisation has to be employed. It is not guaranteed that the finite value obtained with the two methods is the same, since this depends on the regularisation scheme that is being used. However, due to the universal nature of the divergences we will show that it is possible to find a map between the UV and IR regulators.

Let us start by reviewing the heat kernel result. The calculation was recently summarised in [15] and we quote their results. It is important to recall here that we do not perform any Weyl rescaling when computing $\Gamma_{\mathbb{K}}$ using the heat kernel method. We also typically do not rescale our operators to make them dimensionless, this means that the UV cut-off, which we denote by $\Lambda_{\text{HK}}$, is defined in terms of the $\text{AdS}_2$ length scale (see [15] for more in-depth discussions). The result for the quantum effective action computed using heat kernel is

$$\Gamma_{\mathbb{K}}^{\text{HK}} = \frac{n-4}{2}\log(2\pi) - \log\Lambda_{\text{HK}}. \tag{B.6}$$

When comparing this result to the QFT prediction of a Wilson loop expectation value we have to replace the UV regulator $\Lambda_{\text{HK}}$ with a universal factor that depends on the length scale $L$ as discussed in Section 3.4.

Now let us turn to the similar computation using the phase shift method. Following the discussion in Section 3.5 we start by Weyl rescaling the operators $\tilde{\mathcal{K}} = e^{2\rho}\mathcal{K}$, and $\tilde{\mathcal{D}} = e^{3\rho/2}\mathcal{D}e^{-\rho/2}$, and we compute their phase shifts. The massless bosonic and fermionic operators have trivial phase shifts and the total contribution is given by the contribution of the massive fields only, i.e.

$$\delta_{\text{bos}} = (2-n)\arctan p, \qquad \delta_{\text{ferm}} = 2(1-n)\arctan(2p). \tag{B.7}$$

We note that there is some ambiguity in how to define the phase shifts for fermions, since it is sensitive to the quantisation condition imposed in the IR. The quantisation condition we choose is such that the phase shifts for both bosons and fermions vanish in the $p \to 0$ limit. As a consequence, the appropriate combination of the phase shifts seems to have a linear divergence produced by a constant factor in the UV (for $p$ large), that is

$$\lim_{p\to\infty}(\delta_{\text{bos}} - \delta_{\text{ferm}}) = \frac{n\pi}{2} - \frac{1}{p}. \tag{B.8}$$

In fact this will be our main point and is imperative to obtain a match between the two methods.

In order to deal with the constant $n\pi/2$ factor at large $p$ we can subtract this factor from the fermionic phase shift, as long as we remember to include the correction factor $n/2\log 2$ to the quantum effective action, as discussed in Section 3.5. Doing this and integrating the non-trivial phase shifts over $p$, results in

$$\Gamma_{\mathbb{K}}^{\text{PS}} = \frac{n}{2}\log(2\pi) + \log(\Lambda_{\text{PS}}e^{-R}). \tag{B.9}$$

Here $R$ is an IR cut-off and $\Lambda_{\text{PS}}$ is an UV cut-off on the momentum variable $p$. The IR cut-off is there to keep track of the $L$-dependence of the quantum effective action which would otherwise

not show up at all, since we have Weyl rescaled the operators. Notice also that, without the correction factor mentioned above, we would have had $\pi$ and not $2\pi$ as the argument of the log.

Let us now take a look at the explicit comparison of the quantum effective actions which are computed by the two methods, that is

$$\Gamma_{\mathbb{K}}^{\text{HK}} - \Gamma_{\mathbb{K}}^{\text{PS}} = -2\log(2\pi) - \log\Lambda_{\text{HK}} - \log(\Lambda_{\text{PS}}\, e^{-R})\,. \tag{B.10}$$

In order to obtain a match between the two methods we must set the right hand side to zero, yielding

$$\Lambda_{\text{HK}} = \frac{e^R}{4\pi^2\Lambda_{\text{PS}}}\,. \tag{B.11}$$

It is not surprising that the two methods do not yield exactly the same finite answer, resulting in the apparent mismatch of $-2\log(2\pi)$, since the two methods effectively imply different regularisation schemes. However, it is important that the regulators can be identified in a universal way which does not depend on the particular string analysed, as long as their world-sheet Euler characteristic is the same. This means that in the current case the map should not depend on the AdS dimension $n$. We highlight that this only happens because we correctly included the $\log 2$ factors when computing the phase shift quantum effective action.

## C  Alternative derivation of the M2-brane partition function

In this appendix we reproduce the result for the one-loop string partition function of the M2-brane, cf. (153) at leading order, by making use of the Gel'fand-Yaglom method [48,49]. We start by giving the general analysis for obtaining the effective actions $\Gamma_{\mathbb{K}}$, and finally use these results to extract the leading large $N$ behaviour of $Z_{\text{M2}}$.

We recall the scalar and fermionic operators that were derived in Section 5.2, namely:

$$\begin{aligned}
\tilde{\mathcal{K}}_{k,m,q} &= -\partial_\sigma^2 + m^2 + E_{k,q}\,, & E_{k,q} &= -\frac{1-(2kN-q)^2}{\sinh^2(2\sigma)}\,, \\
\tilde{\mathcal{D}}_{k,m,q} &= i\sigma_1\partial_\sigma + m\sigma_2 + a_{k,q}\sigma_3\,, & a_{k,q} &= \frac{(2kN-q)}{\sinh(2\sigma)}\,.
\end{aligned} \tag{C.1}$$

Notice that we made the replacement $\partial_\tau - iqA_\tau \to im$, where the quantum number $m$ associated to the $\tau$-direction is an integer for both scalars and fermions, since we take into account the constant shift due to the connection $qA_\tau$ as explained in Section 3.5. One could imagine that the quantum number of the fermionic operators should be fixed to have half-integer value after the connection has been absorbed into it. This means that $m$ is a half-integer in the end. To address this option, in this appendix we will perform the analysis for both cases by dividing each one in a separate subsection.

By making use of the Gel'fand-Yaglom method, the functional determinant for the bosonic operators is given by[33]

$$\log\frac{\det\mathcal{K}_{k,m,q}}{\det\mathcal{K}_{0,m,q}} = \lim_{\sigma\to\infty}\log\frac{\psi_{k,m,q}(\sigma)}{\psi_{0,m,q}(\sigma)}\,, \tag{C.2}$$

where the wave-functions $\psi_{k,m,q}$ satisfy the boundary problem

$$\mathcal{K}_{k,m,q}\psi_{k,m,q}(\sigma) = 0\,, \tag{C.3}$$

$$\lim_{\sigma\to 0}\psi_{k,m,q}(\sigma) \to \sigma^{|2kN-q|+1/2}\,. \tag{C.4}$$

---

[33]Note that exactly the same expression is satisfied by the squared fermionic operators. We choose to omit some of the details for the solutions of the fermionic operators and we refer the interested reader to [43,50,51] for a more detailed analysis.

Then the corresponding (log of the) functional determinant is given by

$$\Gamma_{\text{bos}}^{(q)} = \frac{1}{2} \sum_{k,m \in \mathbf{Z}} \log \det \mathcal{K}_{k,m,q} = \frac{1}{2} \sum_{k,m \in \mathbf{Z}} \log \frac{\det \mathcal{K}_{k,m,q}}{\det \mathcal{K}_{0,m,q}}, \tag{C.5}$$

since $\zeta$-function regularisation implies $\sum_{k \in \mathbf{Z}} 1 = 0$, and the effective action of the M2-brane, according to the corresponding multiplicity of each operator cf. (133), reads

$$\Gamma_{\text{M2}} = 6\Gamma_{\text{bos}}^{(0)} + \Gamma_{\text{bos}}^{(-2)} + \Gamma_{\text{bos}}^{(2)} - 4\Gamma_{\text{ferm}}^{(-1)} - 4\Gamma_{\text{ferm}}^{(1)}. \tag{C.6}$$

In turn, solving for (C.3) and imposing (C.4) yields

$$\psi_{k,m,q}(\sigma) = \frac{\tanh^{\frac{\left|kN - \frac{q}{2}\right| + 1}{2}} \sigma}{\cosh^m \sigma} \, {}_2F_1\left(\frac{m+1}{2}, \frac{1}{2}\left(m + \left|kN - \frac{q}{2}\right| + 1\right); \frac{\left|kN - \frac{q}{2}\right| + 2}{2}; \tanh^2 \sigma\right), \tag{C.7}$$

which gives rise to the logarithm of the functional determinant for a single bosonic operator

$$\Gamma_{\text{bos}}^{(q)} = \frac{1}{2} \sum_{k,m \in \mathbf{Z}} \log \frac{\Gamma\left(\frac{1+|m|}{2}\right) \Gamma\left(1 + \left|kN - \frac{q}{2}\right|\right)}{\Gamma\left(\frac{1}{2} + \frac{|m|}{2} + \left|kN - \frac{q}{2}\right|\right)}, \tag{C.8}$$

and after imposing $\zeta$-function regularisation, reads

$$\Gamma_{\text{bos}}^{(q)} = -\frac{1}{2} \sum_{k,m \in \mathbf{Z}} \log \Gamma\left(\frac{1}{2} + \frac{|m|}{2} + \left|kN - \frac{q}{2}\right|\right). \tag{C.9}$$

Now, we may use the definition of the gamma function[34] and combine the resulting triple sum over $k, n, m$ to a double sum over $l, k$ as

$$\Gamma_{\text{bos}}^{(q)} = \frac{1}{2} \sum_{k \in \mathbf{Z}} \sum_{l=0}^{\infty} (l+1) \log\left(1 + l + |2kN - q|\right). \tag{C.11}$$

This expression can be simplified even further by specifying the values of $q$, that are $q = 0, \pm 2$ and allow for the evaluation of the sum over $k$, by making use of the $\zeta$-regularised identity [52]

$$\prod_{n=0}^{\infty} (n + x) = \frac{\sqrt{2\pi}}{\Gamma(x)}. \tag{C.12}$$

We may then summarise our results for $\Gamma_{\text{bos}}^{(q)}$ as

$$\Gamma_{\text{bos}}^{(0)} = \sum_{l=0}^{\infty} (l+1)\left(\log \Gamma\left(\frac{l+1}{2N}\right) - \log \sqrt{2\pi} + \frac{1}{2} \log \frac{l+1}{2N}\right), \tag{C.13}$$

$$\Gamma_{\text{bos}}^{(2)} + \Gamma_{\text{bos}}^{(-2)} = \sum_{l=0}^{\infty} (l+1)\left\{\log\left(\Gamma\left(\frac{l+3}{2N}\right) \Gamma\left(\frac{l-1+2N}{2N}\right)\right) - \log(2\pi)\right\}. \tag{C.14}$$

---

[34]The definition of the gamma function as an infinite product reads

$$\Gamma(z) = \frac{e^{\gamma z}}{z} \prod_{n=1}^{\infty} \left(1 + \frac{z}{n}\right)^{-1} e^{z/n}, \tag{C.10}$$

where $z$ is a complex number excluding all non-positive integers.

## C.1 Fermionic operators with integer quantum number

Following a similar procedure as above for the fermionic contributions in (C.6), for the case where $m$ is an integer, yields

$$\Gamma_{\text{ferm}}^{(q)} = -\frac{1}{2} \sum_{k,m \in \mathbf{Z}} \log \Gamma \left( \frac{1}{2} + \frac{|m|}{2} + \left| kN - \frac{q}{2} \right| \right). \tag{C.15}$$

In this case the values of $q$ are given by $q = \pm 1$. We proceed by using (C.10) and (C.12) in similar fashion as above. Subsequently, we arrive at

$$4\Gamma_{\text{ferm}}^{(1)} + 4\Gamma_{\text{ferm}}^{(-1)} = -4 \sum_{l=0}^{\infty} (l+1) \left\{ \log \left( \Gamma \left( \frac{l+2}{2N} \right) \Gamma \left( \frac{l+2N}{2N} \right) \right) - \log 2\pi \right\}. \tag{C.16}$$

Combining everything appropriately in (C.6) gives rise to the M2-brane effective action

$$\Gamma_{\text{M2}} = \sum_{l=0}^{\infty} (l+1) \log \left( \frac{8N^3 \Gamma \left( \frac{l+2}{2N} \right)^4 \Gamma \left( \frac{l+2N}{2N} \right)^4}{(l+1)^3 \Gamma \left( \frac{l+1}{2N} \right)^6 \Gamma \left( \frac{l+3}{2N} \right) \Gamma \left( \frac{l-1+2N}{2N} \right)} \right). \tag{C.17}$$

Extrapolating this sum to $N \to \infty$ yields

$$\Gamma_{\text{M2}} = \log(2\pi) + \mathcal{O}(N^{-2}), \tag{C.18}$$

and agrees with (152) to that order. Notice that the large $N$ scaling of the one-loop partition function is again incorrect, as was already observed in Section 5.2.4. This is believed to be attributed to the orbifold and thus we are unable to match (C.18) with the field theory result (105).

It should be highlighted that for $N = 1$ in (C.18) we reproduce the result of the corresponding one-loop M2-brane partition function of [41], i.e. when their quantum number satisfies $k = 1$. That is, for $N = 1$ (C.18) reads

$$\Gamma_{\text{M2}}^{N=1} = \log 4, \tag{C.19}$$

which was also obtained through the phase shift method in Section 5.2.4.

## C.2 Fermionic operators with half-integer quantum number

Following a similar procedure for the fermionic contributions in (C.6), but now with $m$ being a half-integer, we obtain

$$\Gamma_{\text{ferm}}^{(q)} = -\frac{1}{2} \sum_{k \in \mathbf{Z}} \sum_{m \in \mathbf{Z}+\frac{1}{2}} \log \Gamma \left( \frac{1}{2} + \frac{|m|}{2} + \left| kN - \frac{q}{2} \right| \right). \tag{C.20}$$

We continue by performing the shift of the half-integers to integers in the supersymmetric way of [53], namely

$$\Gamma_{\text{ferm}}^{(q)} = -\frac{1}{4} \sum_{k \in \mathbf{Z}} \left\{ \sum_{m \geq 1} \log \Gamma \left( \frac{1}{2} + \frac{m-1/2}{2} + \left| kN - \frac{q}{2} \right| \right) + \sum_{m \geq 0} \log \Gamma \left( \frac{1}{2} + \frac{m+1/2}{2} + \left| kN - \frac{q}{2} \right| \right) \right.$$

$$\left. + \sum_{m \leq 0} \log \Gamma \left( \frac{1}{2} - \frac{m-1/2}{2} + \left| kN - \frac{q}{2} \right| \right) + \sum_{m \leq -1} \log \Gamma \left( \frac{1}{2} - \frac{m+1/2}{2} + \left| kN - \frac{q}{2} \right| \right) \right\}, \tag{C.21}$$

which in turn yields

$$\Gamma_{\text{ferm}}^{(q)} = -\frac{1}{4}\sum_{k\in\mathbf{Z}}\left\{\sum_{m\in\mathbf{Z}}\log\Gamma\left(\frac{3}{4}+\frac{|m|}{2}+\left|kN-\frac{q}{2}\right|\right)+\sum_{m\in\mathbf{Z}}\log\Gamma\left(\frac{1}{4}+\frac{|m|}{2}+\left|kN-\frac{q}{2}\right|\right)\right.$$
$$\left.+\log\Gamma\left(\frac{3}{4}+\left|kN-\frac{q}{2}\right|\right)-\log\Gamma\left(\frac{1}{4}+\left|kN-\frac{q}{2}\right|\right)\right\}. \tag{C.22}$$

Again, we may proceed by fixing the values of $q$ in order to simplify (C.22) further. In this case we have $q = \pm 1$, then using (C.10), (C.12) and following the above analysis, we arrive at

$$4\Gamma_{\text{ferm}}^{(1)}+4\Gamma_{\text{ferm}}^{(-1)}=-2\sum_{l=0}^{\infty}(l+1)\left\{\log\left(\Gamma\left(\frac{1/2+l+2N}{2N}\right)\Gamma\left(\frac{-1/2+l+2N}{2N}\right)\right)-4\log\sqrt{2\pi}\right.$$
$$\left.+\log\left(\Gamma\left(\frac{5/2+l}{2N}\right)\Gamma\left(\frac{3/2+l}{2N}\right)\right)\right\}+2\sum_{k=1}^{\infty}\log\frac{\Gamma\left(-\frac{1}{4}+kN\right)\Gamma\left(\frac{3}{4}+kN\right)}{\Gamma\left(\frac{1}{4}+kN\right)\Gamma\left(\frac{5}{4}+kN\right)}$$
$$+2\log\frac{\Gamma(3/4)}{\Gamma(5/4)}. \tag{C.23}$$

Combining everything appropriately in (C.6) gives rise to

$$\Gamma_{\text{M2}}=\sum_{l=0}^{\infty}(l+1)\log\left(\frac{8N^3\Gamma\left(\frac{l+1/2+2N}{2N}\right)^2\Gamma\left(\frac{l+5/2}{2N}\right)^2\Gamma\left(\frac{l-1/2+2N}{2N}\right)^2\Gamma\left(\frac{l+3/2}{2N}\right)^2}{(l+1)^3\Gamma\left(\frac{l+1}{2N}\right)^6\Gamma\left(\frac{l+3}{2N}\right)\Gamma\left(\frac{l-1+2N}{2N}\right)}\right)$$
$$+2\sum_{k=1}^{\infty}\log\frac{\Gamma\left(\frac{1}{4}+kN\right)\Gamma\left(\frac{5}{4}+kN\right)}{\Gamma\left(-\frac{1}{4}+kN\right)\Gamma\left(\frac{3}{4}+kN\right)}+2\log\frac{\Gamma(5/4)}{\Gamma(3/4)}. \tag{C.24}$$

At this point we should note that the latter sum in (C.24) prohibits us from expanding this expression around large $N$. In order to correctly extract the large $N$ behaviour of (C.24), we numerically evaluate this expression for various values of $N$. This procedure leads to the following result for the effective action of the M2-brane at leading order in $N$

$$\Gamma_{\text{M2}}\approx\log\left(\frac{11.87}{\sqrt{N}}\right)+\mathcal{O}(N^{-2}), \tag{C.25}$$

where the large $N$ scaling of the one-loop partition function is again incorrect. At this point it is clear that neither boundary condition for the fermions yields a match with the field theory result (105), not even at the level of the scaling of the partition function with $N$. Nonetheless, both (C.18) and (C.25) match with the respective phase shift calculations for the one-loop partition function, see e.g. (152).

## D  WKB approximation

In this section we verify our results from the phase shift method with the WKB approximation, in the large $p$ momentum limit. Since we are interested in the asymptotic limit $p \to \infty$, the WKB approximation instructs us that the solution to the differential equation will be of the following form

$$\psi(\sigma)=\exp ip\sum_{i=0}^{n}p^{-i}S_i(\sigma). \tag{D.1}$$

Once we plug this into our differential equations we may expand our result in powers of $p$ and collect the equations that we obtain at each order $\mathcal{O}(p)$. These equations can then be solved

recursively, order by order for each $S_i'(\sigma)$, and in turn yield an expression for $S_i(\sigma)$. Since we have a second order equation for $S_0'$, we obtain two possible solutions for $S_0$ that differ by an overall sign, corresponding to the incoming and outgoing plane wave solution. Because we approximated the plane wave solution of the eigenfunctions to the sine function, cf. (58), the outgoing wave solution yields the correct phase shift, while for the incoming wave solution we will have to take into account an overall minus sign for the resulting phase shift, cf. (D.4) below.

Furthermore, since the asymptotic solution to the differential equations are plane waves, each $S_i'$ will contain an imaginary and a real part. Since we are working with a ratio of determinants, the imaginary part is of no use for us, as it corresponds to the amplitude of the wave. Therefore only the real part of the solution for $S_i'$ is relevant to the phase shifts. Moreover, if we define

$$S(\sigma) = p \sum_{i=0}^{n} p^{-i} S_i(\sigma), \qquad (D.2)$$

then, at large $\sigma$, the real part of $S(\sigma)$ should be compared with

$$\mathrm{Re}\,[S(\sigma)]_{\mathrm{out}} \simeq p\sigma + \delta(p), \qquad (D.3)$$

for the outgoing wave solution, while the incoming wave solution reads

$$\mathrm{Re}\,[S(\sigma)]_{\mathrm{in}} \simeq -p\sigma - \delta(p), \qquad (D.4)$$

according to equation (58). This is all valid in the IR of our geometry, where the eigenfunctions can be approximated by plane waves, and thus, choosing the outgoing solution, the phase shifts will be given by the following expression in terms of $S_i(\sigma)$ expanded around $\sigma \gg 1$:

$$\delta(p) = p\left(\mathrm{Re}\,[S_0(\sigma)] - \sigma\right) + \mathrm{Re}\,[S_1(\sigma)] + \frac{1}{p}\,\mathrm{Re}\,[S_2(\sigma)] + \frac{1}{p^2}\,\mathrm{Re}\,[S_3(\sigma)] + \mathcal{O}(p^{-3}), \quad (D.5)$$

in the large $p$ limit. We use this relation when evaluating the various phase shifts through the WKB approximation. We conclude this section by collecting our results for the various $d$-dimensional cases. In all the examples discussed in this work, both the bosonic as well as the fermionic fluctuations yield $S_1'(\sigma) = 0$, which gives rise to an undetermined constant in (D.5), and we set to zero.

**Spherical D6-branes**

The $d = 7$ case is actually the simplest case from a computational perspective, and thus yields the following nice relations for the large momentum expansion of the bosonic and fermionic phase shifts:

$$\delta_{x,\mathrm{D6}}(p) = -\frac{1}{4p} - \frac{1}{24p^3} + \mathcal{O}(p^{-5}),$$

$$\delta_{z,\mathrm{D6}}(p) = \frac{3}{4p} - \frac{3}{8p^3} + \mathcal{O}(p^{-5}), \qquad (D.6)$$

$$\delta_{f,+,\mathrm{D6}}(p) = \delta_{f,-,\mathrm{D6}}(p) = \frac{1}{4p} + \frac{1}{24p^3} + \mathcal{O}(p^{-5}).$$

We recall that $\delta_{x,\mathrm{D6}}(p), \delta_{z,\mathrm{D6}}(p)$ correspond to the uncharged and the charged bosonic operators respectively, while $\delta_{f,\pm,\mathrm{D6}}(p)$ to the fermionic ones, cf. (125). Combining these results, the asymptotic behaviour of the integrand of (62) is

$$\delta_{\mathrm{bos},\mathrm{D6}}(p) - \delta_{\mathrm{ferm},\mathrm{D6}}(p) = 6\delta_{x,\mathrm{D6}}(p) + 2\delta_{z,\mathrm{D6}}(p) - 8\delta_{f,+,\mathrm{D6}}(p) = -\frac{2}{p} - \frac{4}{3p^3} + \mathcal{O}(p^{-5}), \quad (D.7)$$

which exactly agrees with (137) at $k = 0$, which is the mode corresponding to the string, up to a constant.

**Spherical D2-branes**

For the $d = 3$ case we found the following results for the first few orders in the large $p$ expansion of the phase shifts

$$\delta_{x,\text{D2}} = \delta_{f,-,\text{D2}} = \frac{13 + 3\pi}{12p} - \frac{19 + 9\pi}{72p^3} + \mathcal{O}(p^{-5}),$$
$$\delta_{y,\text{D2}} = \delta_{z,\text{D2}} = \delta_{f,+,\text{D2}} = \frac{1 + 3\pi}{12p} + \frac{5 - 9\pi}{72p^3} + \mathcal{O}(p^{-5}),$$

(D.8)

and thus the total contribution of the phase shifts becomes

$$\delta_{\text{bos,D2}} - \delta_{\text{ferm,D2}} = 2\delta_{x,\text{D2}} + 6\delta_{y,\text{D2}} - 4\delta_{f,+,\text{D2}} - 4\delta_{f,-,\text{D2}} = -\frac{2}{p} + \frac{2}{3p^3} + \mathcal{O}(p^{-5}),$$

(D.9)

which matches the asymptotic expansion of the appropriate combination of D2 phase shifts when using (179), up to the undetermined constant.

**Spherical D1-branes**

For the $d = 2$ case the phase shifts of the bosonic operators read

$$\delta_{x,\text{D1}} = \frac{38 + 15\sqrt{2}\log(1 + \sqrt{2})}{32p} - \frac{7706 + 7425\sqrt{2}\log(1 + \sqrt{2})}{24576p^3} + \mathcal{O}(p^{-5}),$$
$$\delta_{y,\text{D1}} = 3\frac{2 + 5\sqrt{2}\log(1 + \sqrt{2})}{32p} + 9\frac{18 - 275\sqrt{2}\log(1 + \sqrt{2})}{8192p^3} + \mathcal{O}(p^{-5}),$$
$$\delta_{z,\text{D1}} = \delta_{z,+,\text{D1}} + \delta_{z,-,\text{D1}}$$
$$= \frac{38 + 15\sqrt{2}\log(1 + \sqrt{2})}{16p} - \frac{12314 + 18945\sqrt{2}\log(1 + \sqrt{2})}{12288p^3} + \mathcal{O}(p^{-5}),$$

(D.10)

and the fermionic phase shifts are given by

$$\delta_{f,+,\text{D1}} = \delta_{f,-,\text{D1}} = \frac{22 + 15\sqrt{2}\log(1 + \sqrt{2})}{p} - \frac{6266 - 3825\sqrt{2}\log(1 + \sqrt{2})}{24576p^3} + \mathcal{O}(p^{-5}). \quad \text{(D.11)}$$

We then find that the total phase shift asymptotically behaves as

$$\delta_{\text{bos,D1}} - \delta_{\text{ferm,D1}} = \delta_{x,\text{D1}} + 5\delta_{y,\text{D1}} + \delta_{z,\text{D1}} - 4\delta_{f,+,\text{D1}} - 4\delta_{f,-,\text{D1}}$$
$$= -\frac{1}{p} - \frac{158 - 405\sqrt{2}\log(1 + \sqrt{2})}{192p^3} + \mathcal{O}(p^{-5}).$$

(D.12)

This expression verifies the (universal) logarithmic UV divergence in (201). Notice that this analysis only looks at the real part of the solutions and thus the imaginary contribution we conjecture to appear in $\Gamma_{\mathbb{K}}$ is not obvious here. However, performing the WKB analysis gave rise to complex values both for the fermionic as well as the bosonic phase shifts in odd powers of $p$, in contrast to all previously studied cases which only contain these imaginary values at even powers in $p$. In particular we found that the imaginary solution for odd powers in $p$ reads

$$\text{Im}[\delta_{x,\text{D1}}] = \text{Im}[\delta_{y,\text{D1}}] = -\frac{15\pi}{32\sqrt{2}p} + \frac{2475\pi}{8192\sqrt{2}p^3} + \mathcal{O}(p^{-5}),$$
$$\text{Im}[\delta_{z,\text{D1}}] = -\frac{15\pi}{32\sqrt{2}p} + \frac{6315\pi}{4096\sqrt{2}p^3} + \mathcal{O}(p^{-5}),$$
$$\text{Im}[\delta_{f,+,\text{D1}}] = \text{Im}[\delta_{f,-,\text{D1}}] = -\frac{15\pi}{32\sqrt{2}p} + \frac{1275\pi}{8192\sqrt{2}p^3} + \mathcal{O}(p^{-5}),$$

(D.13)

Table 3: Here we summarise the behaviour of the world-sheet conformal factor (28), as well as the behaviour of the pulled-back dilaton $\Phi_0$ at the centre (IR) of the world-sheet. Once again we note that $\sigma \to \infty$ in the IR.

| Dimension | $\left. e^{2\rho}\,\ell_s^{-2}\right|_{\text{IR}}$ | $\left. N\,e^{\Phi_0}\right|_{\text{IR}}$ |
|:---:|:---:|:---:|
| $d = 2$ | $\left(8\pi^3\lambda\right)^{1/4} e^{-2\sigma}$ | $-i\dfrac{\left(2\pi\lambda^3\right)^{1/4}}{5^{1/2}}$ |
| $d = 3$ | $(6\pi^2\lambda)^{1/3} e^{-2\sigma}$ | $\dfrac{\lambda^{5/6}}{(6\pi^2)^{1/6}} e^{\sigma}$ |
| $d = 5$ | $\dfrac{8\lambda}{\pi} e^{-2\sigma}$ | $\sqrt{\dfrac{\lambda^3}{8\pi^5}}$ |
| $d = 6$ | $21\pi\, e^{-\frac{8\pi^3}{3\lambda}} e^{-2\sigma}$ | $4\pi\sqrt{\dfrac{5}{3}}\, e^{-\frac{8\pi^3}{3\lambda}}$ |
| $d = 7$ | $\dfrac{4\pi^4}{\lambda} e^{2\sigma}$ | $\dfrac{\pi^2}{2\sqrt{\lambda}} e^{3\sigma}$ |

for the bosonic and fermionic phase shifts respectively. Combining these appropriately yields

$$\text{Im}\left[\delta_{\text{bos,D1}} - \delta_{\text{ferm,D1}}\right] = \frac{135\pi}{64\sqrt{2}p^3} + \mathcal{O}(p^{-5}), \tag{D.14}$$

and thus the imaginary contribution cancels at leading order and therefore the term involving the UV divergence is not imaginary. However, the next-to-leading order does not cancel, therefore providing some evidence for our conjecture regarding the imaginary value of $(\Gamma_{\mathbb{K}})_{\text{reg}}$ proposed in Section 7.1.

# E  Behaviour of dilaton across different cases

In this section we collect the behaviour of the pull-back of the dilaton $\Phi_0$ at the centre of the world-sheet, that is at large $\sigma$ in the conformal coordinates (28), for the cases $d = 2, 3, 5, 6, 7$. In two cases ($d = 2, 3$) the computations are worked out in non-conformal coordinates, and the relation among the two sets of coordinates is only implicit.

For $d = 2$ the pulled-back dilaton is expressed in terms of the $u$ coordinate as

$$e^{\Phi_0} = \frac{\left(2\pi\lambda^3\right)^{1/4}}{N\sqrt{(u+1)(u^2-5)}}. \tag{E.1}$$

However, recall that $u$ is not conformal and is related to the conformal coordinates $(\sigma, \tau)$ by (189). In this case we can relate the two coordinates in the IR by[35] $u_{\text{IR}} = e^{-2\sigma}$, when $\sigma \to \infty$.

For $d = 3$ the pulled back dilaton in terms of $u$ reads

$$e^{\Phi_0} = \frac{\lambda^{5/6}}{6^{1/6}\pi^{1/3}N\sinh u\cosh^{1/4}u}, \tag{E.2}$$

and the two coordinate patches $(u, \tau)$ and $(\sigma, \tau)$ are related by equation (165). Also in this case,[36] the IR regime is reached by setting $u_{\text{IR}} = e^{-\sigma}$, when $\sigma \to \infty$.

When the number of dimensions $d$ of the sphere is equal to 3 and 7, the dilaton is divergent in the IR, i.e. it exhibits a non-trivial dependence on $\sigma$. These are also the only two cases in

---

[35]In particular, from (189) we have $\sigma = \text{arccoth}\left(\sqrt{\frac{2(u+1)}{u+2}}\right) - \text{arccoth}\sqrt{2}$.

[36]In this case the conformal coordinate is given by $\sigma = \frac{2}{3\sinh^{3/2}u}\,{}_2F_1\left(\frac{3}{4}, \frac{3}{4}, \frac{7}{4}, -\text{sech}^2u\right)$, cf. (165).

which the pull-backed dilaton $\Phi_0$ is proportional to the conformal factor (up to constants), as was already highlighted in Section 3.4, where the discussion on the logarithmic divergences of the effective action took place. This is also shown in more detail in Table 3, where we also recap the asymptotic behaviour of the conformal world-sheet factor in the IR, in the different dimensions $d$.

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
