# Peer review of "Wilson Loops and Spherical Branes"

_SciPost Physics, doi:SciPost Phys. 18, 050 (2025)_

## Round 1 · Referee Report · Anonymous (Referee 1) · 2024-10-11

Strengths

  1. Non-trivial test of holography in the non-conformal case, and non-trivial related analysis of quantization of strings and branes with proposal of regularization scheme.
  2. Thorough analysis with many details, for sure of reference for future work on non-conformal precision holography

Weaknesses

There are no weaknesses.

Report

This paper contains results, obtained via holography, on Wilson loops expectation values at strong coupling, for Yang-Mills theories with maximal supersymmetry and defined on a d-dimensional sphere, for d=2,3,7. For such vevs results exact in the coupling are evailable in the literature, obtained at large N via supersymmetric localization.
The holographic calculation of this paper consists in the evaluation of the one-loop partition function for the string and M2-brane in the relevant dual geometry.
This case goes beyond the realm of the usual holographic dictionary, because the relevant field theory is not conformal. On the gravity side, this implies the presence of a running dilaton, something which plays a relevant role when discussing the UV finiteness of the partition function. The question of how to quantize strings and branes in backgrounds relevant in holography is definitively a relevant one, and it is also full of subtleties - mostly related to the evaluation of the relevant functional determinants - already in the conformal case. The thorough analysis of this paper addresses them in details (also with useful appendices containing standard and less standard lore on determinants calculations) and proposes a regularization scheme, which is also tested on previously treated cases. The agreement (or disagreement) with the results obtained from localization is also discussed in details, and the directions for future research are an interesting part of the manuscript.
For all these reasons, this work is certainly going to be of future reference for works on (precision) holography in the non-conformal case, and I am happy to recommend its publication on Scipost.

Requested changes

Very minor, on the Conclusions: in the first sentence, “we conduct a study into sub-leading corrections to the 1/2 BPS Wilson loop vev [...]” should better be “we conduct a study, via holography, of sub-leading, strong coupling corrections to the 1/2 BPS Wilson loop vev […]”. Also, to faciliate the comparison between localization and gravity results, (8.1) and (8.3), it would be perhaps useful to have them much closer in the text or on a Table.

Recommendation

Publish (meets expectations and criteria for this Journal)

---

## Round 1 · Referee Report · Anonymous (Referee 2) · 2024-11-26

# Report

This paper is relevant for the community working on supersymmetric Wilson loops in holography. The authors investigate the vacuum expectation values of 1/2-BPS Wilson loops in maximally supersymmetric Yang-Mills (MSYM) theory defined on d-dimensional spheres. They focus on the holographic duals of these theories, particularly back-reacted spherical D-branes, evaluating the holographic Wilson loops to next-to-leading order. The paper highlights novel treatments of divergences due to non-constant dilaton profiles and presents a framework to match the subleading scaling of the Wilson loops with string theory predictions.

The manuscript is generally well-structured with clear sections dedicated to the theoretical setup, technical computation, and specific cases for d=2,3,7. The introduction comprehensively motivates the work and situates it within the broader context of gauge/gravity duality. Technical details, though extensive, are provided systematically, such that the paper can be used as reference for further developments in the field.

One tiny improvement would be to move the comment about instantons being exponentially suppressed at large N from page 25 to page 5, where such fact is first mentioned. Apart from that, the computations appear robust and I could not find mistakes or typos. The authors were clear regarding what they did and did not manage to answer, also providing a list of open questions and problems to address.

I recommend the paper for publication in its current form. The work is a valuable contribution to the study of holographic Wilson loops and non-conformal gauge theories.

---

## Editorial Decision

published